# The role of land cover on the climate of glacial Europe

Patricio Velasquez[1,2], Jed O. Kaplan[3], Martina Messmer[1,2,4], Patrick Ludwig[5], and Christoph C. Raible[1,2]

[1]Climate and Environmental Physics, Physics Institute, University of Bern, Bern, Switzerland
[2]Oeschger Center for Climate Change Research, University of Bern, Bern, Switzerland
[3]Department of Earth Sciences, The University of Hong Kong, Hong Kong
[4]School of Earth Sciences, The University of Melbourne, Melbourne, Victoria, Australia
[5]Institute of Meteorology and Climate Research, Karlsruhe Institute of Technology, Karlsruhe, Germany

**Correspondence:** Patricio Velasquez (patricio.velasquez@climate.unibe.ch)

**Abstract.** Earth system models show wide disagreement when simulating the climate of the continents at the Last Glacial Maximum (LGM). This disagreement may be related to a variety of factors, including model resolution and an incomplete representation of Earth system processes. To assess the importance of resolution and land-atmosphere feedbacks on the climate of Europe, we performed an iterative asynchronously coupled land-atmosphere modelling experiment that combined a global climate model, a regional climate model, and a dynamic vegetation model. The regional climate and land cover models were run at high (18 km) resolution over a domain covering the ice-free regions of Europe. Asynchronous coupling between the regional climate model and the vegetation model showed that the land-atmosphere coupling achieves quasi-equilibrium after four iterations. Modelled climate and land cover agree reasonably well with independent reconstructions based on pollen and other paleoenvironmental proxies. To assess the importance of land cover on the LGM climate of Europe, we performed a sensitivity simulation where we used LGM climate but present-day (PD) land cover. Using LGM climate and land cover leads to colder and drier summer conditions around the Alps and warmer and drier climate in southeastern Europe compared to LGM climate determined by PD land cover. This finding demonstrates that LGM land cover plays an important role in regulating the regional climate. Therefore, realistic glacial land cover estimates are needed to accurately simulate regional glacial climate states in areas with interplays between complex topography, large ice sheets and diverse land cover, as observed in Europe.

## 1 Introduction

The Last Glacial Maximum (LGM, 21 ka; Yokoyama et al., 2000; Clark et al., 2009; Van Meerbeeck et al., 2009) is a period of focus for Earth system modelling because it represents a time when boundary conditions were very different from the present and is therefore a good testbed of models' ability to faithfully reproduce a range of climate states (e.g., Mix et al., 2001; Janská et al., 2017; Cleator et al., 2020). In Europe, the LGM is also an interesting period in human history, because small groups of highly mobile Upper Paleolithic hunter-gatherers persisted in the face of inhospitable climate, while Neanderthals disappeared (Finlayson, 2004; Finlayson et al., 2006; Finlayson, 2008; Burke et al., 2014; Maier et al., 2016; Baena Preysler et al., 2019; Klein et al., 2021). However, despite more than three decades of research, the LGM climate of the continents is only poorly understood. Global climate models (GCMs) show little agreement in LGM simulations for Europe (Braconnot

et al., 2012; Kageyama et al., 2017; Ludwig et al., 2019; Kageyama et al., 2020). It has been suggested that a reason for the large uncertainty could be related to the spatial resolution in the climate models (Walsh et al., 2008; Jia et al., 2019b; Ludwig et al., 2019; Raible et al., 2020). Advances in regional climate models have led to the application of such models to the glacial climate of Europe on a high spatial resolution (e.g., Kjellström et al., 2010; Strandberg et al., 2011; Gómez-Navarro et al., 2012, 2013; Ludwig et al., 2017, 2020). Here, we further investigate the importance of land cover for climate during this period.

Paleoclimate reconstructions suggest that the climate of Europe was 10 to 14 °C colder and around 200 mm year$^{-1}$ drier during the LGM compared to present day (PD; Wu et al., 2007; Bartlein et al., 2011). However, uncertainties in the paleoclimate reconstructions are large, the few sites with samples dating to the LGM are not uniformly distributed in space (e.g., Wu et al., 2007), and in some regions, reconstructions are contradictory (e.g., de Vernal et al., 2006). For example, some LGM climate reconstructions suggest that the Iberian Peninsula was dry (Bartlein et al., 2011; Cleator et al., 2020), while others suggest wetter conditions were prevalent (Vegas et al., 2010; Moreno et al., 2012). Some of these discrepancies may result from the fact that many paleoclimate archives record a certain season, while the signal is frequently interpreted as an annual value (Beghin et al., 2016), or because even sites that are close together record strong climatic gradients. Whatever the case, generation of a spatially continuous map of climate and environmental conditions in LGM Europe is currently not possible using a strictly data-driven approach. As an alternative, it should be possible to generate continuous maps using climate models.

GCM simulations are overall consistent with reconstructions in simulating an LGM climate that is largely colder and drier than PD (e.g., Ludwig et al., 2016; Hofer et al., 2012a). At the regional scale, however, GCMs show broad intermodel variety and partly disagree in comparison to proxy reconstructions, particularly concerning the magnitude and spatial patterning of temperature and precipitation (Harrison et al., 2015). For example, GCMs show a broad disagreement in the simulation of precipitation over the Iberian Peninsula, with some models suggesting it was wetter while in others the simulated climate is drier compared to PD (Beghin et al., 2016). One possible explanation for the disagreement is the coarse spatial resolution of the GCMs; at the continental scale, mountains, ice sheets, and water bodies have an important influence on regional circulation and climate that may not be represented appropriately at a typical GCM grid spacing of ca. 100 km (Rauscher et al., 2010; Gómez-Navarro et al., 2011, 2012, 2013; Di Luca et al., 2012; Prein et al., 2013; Demory et al., 2020; Iles et al., 2020).

To improve the representation of local and regional climate, GCMs can be dynamically downscaled using regional climate models (RCMs). Ludwig et al. (2019) found that downscaling using an RCM offers a clear benefit to answer paleoclimate research questions and to improve interpretation of climate modelling and proxy reconstructions. They also found that the regional climate models require appropriate surface boundary conditions to properly represent the lower troposphere. Studies have demonstrated that a realistic representation of surface conditions is essential for the accuracy of the simulated regional climate as they play a crucial role in regulating water and energy fluxes between the land surface and the atmosphere (e.g., Crowley and Baum, 1997; Kjellström et al., 2010; Strandberg et al., 2011, 2014; Gómez-Navarro et al., 2015; Jia et al., 2019a; Ludwig et al., 2017).

As noted above, the sparse distribution of paleoecological samples in Europe, that are securely dated to the LGM, precludes the development of a continuous map of land cover that can be used as a boundary condition for climate modelling and other purposes, e.g., archaeological and botanical research. Since climate affects land cover and land cover in turn affects climate,

it is not sufficient to simply use climate model output to generate a vegetation map. To overcome this dichotomy, one may adopt a coupled modelling approach, where a climate model simulation is initialised with an estimate of land cover and the resulting climate output fields are used to simulate land cover. This process, which is called asynchronous coupling, is repeated between the climate and land cover models until the land-atmosphere system is in quasi-equilibrium. Asynchronous coupling is computationally inexpensive and has been successfully employed in several modelling studies to investigate problems in paleoclimate science (e.g., Texier et al., 1997; Noblet et al., 1996). For example, Kjellström et al. (2010) uses an iterative coupling of an RCM and a land-cover model and found that asynchronous coupling produces a vegetation cover being close to paleo reconstructions. Also, Strandberg et al. (2011) and Ludwig et al. (2017) showed that fine scale land cover is important for representing the climate and needs to be included in regional climate simulations.

Here, we perform an asynchronous coupled modelling study to simulate the climate and land cover of Europe at the LGM. The asynchronous coupled modelling starts with a GCM (CCSM4; Gent et al., 2011) which serves as input to drive a dynamic vegetation model (LPJ-LMfire; Pfeiffer et al., 2013). In a next step, the atmospheric boundary conditions from the GCM and the output of LPJ-LMfire are passed to an RCM (WRF; Skamarock and Klemp, 2008). The resulting RCM output is in turn used to drive LPJ-LMfire which again returns land cover to the RCM. The RCM simulation is then repeated with the new land cover as boundary condition. We evaluate the results of our coupled model experiment using independent reconstructions of land cover and climate, and we perform a sensitivity test to better understand the importance of land cover for LGM climate in Europe by forcing the RCM with an alternative set of land-surface boundary conditions.

## 2 Models and methods

### 2.1 General circulation model: CCSM4

In this study, we dynamically downscaled one global climate simulation for PD conditions (1990 CE conditions) and another one for LGM. These global simulations were performed with the atmospheric and land component of the Community Climate System Model (version 4, CCSM4; Gent et al., 2011). A horizontal grid spacing of $1.25\,° \times 0.9\,°$ (longitude $\times$ latitude) was used in both components. The vertical dimension is discretised in 26 vertical hybrid sigma-pressure levels in the atmospheric component (CAM4; Neale et al., 2010) and 15 soil layers in the land component (CLM4; Oleson et al., 2010), respectively. CCSM4 was coupled to so-called *data models* for the ocean and sea ice. These surface boundary conditions were obtained from a fully coupled simulation with CCSM3 at lower resolution (see details in: Hofer et al., 2012a). CCSM3 provided monthly mean time-varying sea-ice cover and sea-surface temperatures (SSTs). Furthermore, the Community Ice Code (version 4, CICE4; Hunke and Lipscomb, 2010) was set to its thermodynamic-only mode. This means that sea-ice cover was prescribed and surface fluxes through the ice were computed by considering snow depth, albedo, and surface temperature as simulated by CAM4 (Merz et al., 2015). Further details of the global model setting were presented in Hofer et al. (2012a,b) and Merz et al. (2015).

Each CCSM4 simulation was run for 33 years, from which only the last 30 years and 2 months were used in this study. Present-day (PD) boundary conditions were set to 1990 CE values, whereas LGM boundary conditions were modified as

follows: lower concentrations of greenhouse gases ($CO_2$ = 185 ppm, $N_2O$ = 200 ppb and $CH_4$ = 350 ppb), changed Earth's orbital parameters (Berger, 1978), addition of major continental ice sheets (Peltier, 2004) and associated sea-level changes (120 m lower than today; Clark et al., 2009). Note that land cover was set to pre-industrial conditions in the LGM simulation.

Additional land cells of the LGM simulation are filled with vegetation and soil types of the mean values of nearby cells and in the ice-covered regions the model's standard values are used for such conditions. The simulations further provided 6-hourly data, which is necessary to drive regional climate models.

These PD and LGM CCSM4 simulations have been analysed in a variety of studies, including additional simulations for other glacial and interglacial states (e.g., Hofer et al., 2012a,b; Merz et al., 2013, 2014a,b, 2015, 2016; Landais et al., 2016). The

focus of these studies was in particular on the model's ability to simulate LGM climate and atmospheric circulation changes during glacial times. Hofer et al. (2012a) showed that the model performs reasonably well under PD conditions, showing a cold bias in the global mean temperature of 0.3 °C. The reason for this bias is the rather coarse resolution of the ocean, which led to an underestimation of the northward heat transport in the North Atlantic and an overestimation in the horizontal extension of sea-ice cover (Hofer et al., 2012a). The LGM CCSM4 simulation agrees with models used in the second phase

of the Paleoclimate Modelling Intercomparison Project (PMIP2; Braconnot et al., 2007) showing a global mean temperature response between LGM and preindustrial conditions of -5.6 °C. However, the temperature response over Europe shows a better agreement with proxy data (Wu et al., 2007) than the multi-model mean response in Braconnot et al. (2007). The global mean precipitation response of the LGM simulation used in this study is similar to the multi-model mean response of Braconnot et al. (2007), although the regional pattern and seasonal behaviour show some deviations from proxy data over Europe (Wu et al.,

2007; Hofer et al., 2012a). The LGM simulation further reveals a clear southward shift and a more zonal orientation of the storm track over the North Atlantic compared to PD conditions (Hofer et al., 2012a). This shift and substantial changes in the weather patterns (Hofer et al., 2012b) are able to explain precipitation anomalies over the Iberian Peninsula and the western part of the Mediterranean Sea. Sensitivity simulations in Merz et al. (2015) suggested that the shift can be traced back to the height of the Laurentide ice sheet and the effect of it on stationary and transient waves and the eddy-driven jet over the North

Atlantic. Such a shift is also reported in several other modelling studies (see review of Raible et al., 2020). Overall, CCSM4 simulations of LGM climate were state-of-the-art in 2012 and they are still today as their horizontal resolution is similar to models used in phase 4 of the Paleoclimate Model Intercomparison Project (PMIP4; Kageyama et al., 2017, 2020).

## 2.2 Regional climate model: WRF

To investigate the importance of model resolution and land cover on the climate of LGM Europe, we dynamically downscaled

the global CCSM4 simulations using the Weather Research and Forecasting (WRF) model (version 3.8.1, Skamarock et al., 2008). This regional climate model was set up with two domains that are two-way nested. These domains have 40 vertical eta levels and a horizontal grid spacing of 54 and 18 km, respectively. The inner domain is centred on the Alpine region and the outer domain includes an extended westward and northward area to capture the influence of the North Atlantic Ocean and the Fennoscandian ice sheet on the European climate (Fig. 1). The relevant parameterisation schemes chosen to run WRF are

described in Velasquez et al. (2020).

The initial and boundary conditions for the WRF model were provided by CCSM4 simulations, including the Fennoscandian ice sheet and reduced sea levels during the LGM. Other external forcing functions followed the PMIP3 protocol (for more details see: Hofer et al., 2012a; Ludwig et al., 2017). Furthermore, no nudging was applied in the RCM simulations. LGM glaciation over the Alpine region was included in the regional climate model using estimates from Seguinot et al. (2018) and

additional LGM glaciated areas (e.g., Pyrenees, Carpathians) from Ehlers et al. (2011). The LGM land cover is described in Sect. 2.4. These settings are used to produce the main simulation ($LGM_{LGM}$) which at the same time is the final product of the asynchronous coupling design (described in Sect. 2.4).

To perform the regional simulations in this study, we used the so-called adaptive time-step method as described in Skamarock et al. (2008), i.e., the integration time step can vary from time to time. For example, the model is stable with a time step of

160 seconds during most integration steps but it might need a reduction to 60 seconds during convective situations to maintain stability. With a fixed time step, the entire simulation must be run with 60 seconds to overcome these convective situations, while the adaptive time-step method is able to make use of the larger time step 160 seconds during most of the simulation. The advantage of this approach is to substantially save computer resources. Furthermore, each simulation was driven by the 30 years of the corresponding GCM simulation (excluding the 3-year spin-up of the GCM simulation). These 30 years were

split up into two single 15-year periods which both are preceded by a 2-month spin-up to account for the time required for land surface to come into quasi equilibrium. We used the last two months of the 3-year spin-up of the GCM simulation for the first 15 years. A spin-up of two months in the regional model is sufficient as soil moisture reaches a quasi equilibrium, i.e., no significant trend after 15 days in the four layers of the WRF land-surface scheme, i.e., up to the level of 1 m.

We also carried out a control simulation under PD conditions ($PD_{PD}$) to assess the simulated LGM climate and land cover

response against proxy data. $PD_{PD}$ was driven by the GCM simulation with 1990 CE conditions (Hofer et al., 2012a), and used the default PD MODIS-based land-cover dataset from WRF (Skamarock and Klemp, 2008).

Finally, we conducted a sensitivity simulation to quantify the importance of land cover for the LGM climate in Europe ($LGM_{PD}$). This simulation used the GCM simulation with LGM conditions (Hofer et al., 2012a), but with the default PD MODIS-based land-cover dataset from WRF for the land surface (Skamarock and Klemp, 2008).

Comparing $LGM_{PD}$ with $PD_{PD}$ illustrates the atmospheric response to changes only in the atmospheric forcing, i.e., without changes in land cover. The comparison of $LGM_{LGM}$ and the $LGM_{PD}$ allows us to extract the influence of land cover on the atmosphere, i.e., without changes in atmospheric boundary conditions. These simulations are summarised in Table 1.

To assess the statistical significance of the responses, we use a bootstrapping technique (Wilks, 2011). This technique consists of randomly selecting elements from the original sample to generate a new sample. This is also called resampling whereby

the number of elements remains unchanged. This procedure is repeated 1000 times. A new mean value is calculated from each resampling obtaining 1000 mean values that are used to build a probabilistic distribution function (PDF). We assess the significance of the mean value using a significance level of 0.01 for each PDF's tail. The bootstrapping technique is applied to the spatially averaged values using as elements the climatological mean values across Europe. We use one experiment to build the PDF on which we allocate the spatially averaged value of another experiment to assess the significance. Also, the

bootstrapping technique is applied at each grid point using as elements the 30 yearly mean values. At each grid point, we obtain

the PDF from one experiment on which we allocate the climatological mean value of the another experiment to estimate the significance.

## 2.3 Dynamic global vegetation model: LPJ-LMfire

Land cover for the LGM is simulated by the LPJ-LMfire dynamic global vegetation model (Pfeiffer et al., 2013), which is an
evolution of LPJ (Sitch et al., 2003). LPJ-LMfire is a processed-based, large-scale representation of vegetation dynamics and land-atmosphere water and carbon exchanges that simulates land cover patterns in response to climate, soils, and atmospheric $CO_2$ concentrations (Prentice et al., 1992; Haxeltine and Prentice, 1996; Haxeltine et al., 1996; Kaplan, 2001; Kaplan et al., 2016). LPJ-LMfire simulates land cover in the form of the fractional coverage of nine plant functional types (PFTs), including tropical, temperate, and boreal trees, and tropical and extratropical herbaceous vegetation (Sitch et al., 2003).

In each of our simulations, we drove LPJ-LMfire for 1020 years with the climate and forcing (greenhouse gases: $CO_2$, $N_2O$ and $CH_4$) from the GCM, and PD soil physical properties extrapolated out on to the continental shelves (Kaplan et al., 2016). Such a long simulation is not necessary to bring above-ground vegetation into quasi-equilibrium with climate, but it allows soil organic matter to equilibrate. Since the vegetation model is computationally inexpensive, we performed these millennium-long simulations so that they could be analysed for other purposes in the future.

## 2.4 Iterative asynchronous coupling design

To create the best possible estimate of European land cover for the LGM, we used an iterative asynchronous coupling design that combines CCSM4/WRF with LPJ-LMfire model (resulting in the $LGM_{LGM}$ climate simulation). This coupling design consists of four steps: (i) the fully coupled CCSM4 provides atmospheric variables for the LGM to generate the first approximation of LGM land cover with LPJ-LMfire at a horizontal grid spacing of 1.25 ° × 0.9 ° (longitude × latitude), (ii) WRF is
driven by the CCSM4 with LGM conditions and the first approximation of LGM land cover created in step (i) to generate the first downscaled atmospheric variables for the LGM at 54 and 18 km grid spacing, (iii) LPJ-LMfire is run with the downscaled LGM atmospheric variables (from step ii) to regenerate the LGM land cover at the RCM resolutions, (iv) same as in (ii) but WRF uses the land surface boundary conditions simulated at 54 and 18 km. Step (iii) and (iv) are carried out asynchronously over five additional iterations to achieve a quasi-equilibrium between the climate and land cover. Parts (i) and (ii) are consid-
ered as the first iteration and the iterations of (iii) and (iv) are considered as the second-to-seventh iterations. The variables that are passed between the climate and vegetation models are summarised in Table 2. Vegetation cover fraction is defined as the fraction of ground covered by vegetation at each grid point, with values between 0 and 100 %. Also, to classify vegetation cover fraction into the land cover categories required by WRF (according to NOAH-MP MODIS; Niu et al., 2011), we used a simple scheme based only on the cover fraction of the LPJ-LMfire PFTs. Note that we identified a problem with the land-sea mask
and around glaciated areas which was fixed between the third and fourth iteration. To test whether the asynchronous coupling has reached a quasi-equilibrium state, we assess the statistical significance with a bootstrapping technique that is introduced at the end of Sect. 2.2.

## 3 Results of the iterative asynchronous coupling

The offline coupling design (Sect. 2.4) aims at generating a simulation of the LGM climate and land cover that is as realistic as possible. Thereby, it is important that the land cover and the climate is in quasi-equilibrium (Strandberg et al., 2011) in order to discard the source of uncertainty related to an unbalanced climate system. In this study, we determine the quasi-equilibrium in the land cover and the climate, first, through empirical observation and second, through a statistical test applied to a set of variables (see Sect. 2.4). To illustrate the differences between the iterations, we concentrate on climate and land cover changes over the ice-free land areas of Europe at LGM (in domain 2) using the following variables: the spatial climatology of total precipitation, temperature at 2 m, albedo, deep-soil temperature, cloud cover, leaf area index and vegetation cover fraction, and the number of grid points dominated by the following land cover categories: sparsely vegetated, tundra, forest, and shrublands (NOAH-MP MODIS categories, Niu et al., 2011). Land cover categories that are functionally similar are grouped together, e.g., wooded tundra, mixed tundra and barren tundra are all combined to the category tundra. Some land cover categories are not considered in our analysis as they are poorly represented in both periods, e.g., savanna, grassland and wetland, or are not relevant for the LGM, e.g., cropland and urban (Fig. 2a-b).

Results show that the most notable and statistically significant changes in the variables exchanged between land cover and atmosphere occur within the first four iterations (Fig. 3). Only albedo and leaf area index show significant changes also in the fifth iteration. The significance of the differences is assessed using a two-tailed bootstrapping technique with a significance level of 2 % (Sect. 2.2) and is marked in each panel of Fig. 3. Note that the significance for the land cover categories is not shown. The reason is that this significance can be summarised using the significance of the vegetation cover fraction. The variables level off from the fifth to the seventh iteration. In particular, we observe two sharp changes in all variables within the first five iterations. The first important change is found between the first and second iteration and is present in the atmospheric and land surface variables. The reasoning is twofold: (i) There are significant changes in the land cover classes, e.g., forest fraction is reduced from 35 to 2 %. (ii) The horizontal resolution of the land cover is increased from approximately 100 to 18 km (horizontal grid spacing of GCM and RCM, respectively). The higher spatial resolution of the RCM results in a better representation of the regional-to-local scale processes and interactions with other components of the climate system compared to a GCM (Ludwig et al., 2019). The second change happens between the third and fourth iteration in precipitation and cloud cover (Fig. 3a and 3d) and between the fourth and fifth in albedo and leaf area index (Fig. 3c and 3d). Note that the improvements in the land sea mask and around glaciated areas between the third and fourth iteration can partially explain the significantly sharp change in precipitation and cloud cover between the third and fourth iteration. We consider the significant changes from the fourth to the fifth iteration in albedo and leaf area index as a delayed effect of the variation in cloud cover and precipitation and thus an effect of the improvement.

Spatially averaged total precipitation significantly decreases in the second iteration (drop of 15 mm) and significantly increases in the fourth iteration (increase of 9 mm) with small and no significant changes thereafter (blue line in Fig. 3a). A significant decrease in the spatially averaged temperature at 2 m is observed in the second iteration (cooling of around 0.5 °C), which turns into small and insignificant fluctuations in the range of a tenth of a degree afterwards (red line in Fig. 3a). Albedo

significantly decreases until the third iteration (change of around 1.3 %) and significantly increases in the fifth iteration with small and insignificant changes afterwards (blue line Fig. 3c). A significant cooling is also observed in the spatial-averaged deep-soil temperature from the first to the third iteration (red line in Fig. 3c). Deep-soil temperature stabilises from the fourth to the seventh iteration. Similar to total precipitation, we observe that the spatially averaged cloud cover fraction significantly decreases in the second iteration (change of 0.009) and significantly increases in the fourth iteration (change of 0.003) with very small and insignificant variations afterwards (blue line in Fig. 3d). Leaf area index significantly fluctuates till the fifth iteration (maximum change of 0.5) with minimal and insignificant changes thereafter (red line Fig. 3d). Additionally, changes in vegetation cover fraction are observed in the first four iterations (32, 18, 16 and 15 %, respectively). In the following iterations, the changes remain rather small and insignificant (Fig. 3b). The land cover categories change mostly between the first and second iteration. The category sparsely vegetated is strongly increased in the second iteration and at the same time forest is strongly reduced (Fig. 3b). Thus, the quasi-equilibrium state is achieved after the fourth to fifth iteration.

In the following, we analyse the spatial patterns of climate and land cover between the iterations that represent the transient progression towards quasi-equilibrium (fourth minus first iteration) and the quasi-equilibrium state (seventh minus fourth iteration). We consider temperature at 2 m, total precipitation and vegetation cover fraction as variables that summarise the coupled land-atmosphere response. Note that temperature, precipitation and vegetation cover fraction are displayed using absolute differences (Fig. 4a-f).

During the transient state (Fig. 4a, 4c and 4e), the southwestern part of the Iberian Peninsula and some areas in Italy and Greece warms, the rest of Europe experiences a cooling. In addition, precipitation reveals a wetting over the Iberian Peninsula, in parts of France and Balkan Peninsula, and a drying over eastern Europe, north of the Alps and over some regions of France (Fig 4c). The vegetation cover fraction shows a strong decrease during the transient state, particularly in the flat lands of eastern Europe (over 50 % reduction) and the Italian Peninsula, and an increase over the Iberian Peninsula (around 20 %) and northwest of the Alps (around 40 %; Fig. 4e). Vegetation response is related to changes in temperature and precipitation. Namely, many regions that experience a cooling are related to a reduction in vegetation. Drying and wetting are overall related to a reduction and an increase in vegetation cover, respectively. This is true except for few areas in the north of the Alps and along the Mediterranean coast such as the eastern region of the Iberian Peninsula, southern Greece and southern Italy. North of the Alps, the poor relation between precipitation and vegetation cover fraction could be explained by a lesser pronounced cooling. In the eastern part of the Iberian Peninsula and southern Greece, the reduction of vegetation seems to be related to an increase in temperature.

The changes between the seventh and fourth iteration, which illustrates the quasi-equilibrium state, are minimal for the three variables (Fig. 4b, 4d and 4f). The remaining small differences are interpreted as a part of the internal climate variability and uncertainties predominantly caused by parameterisations in the models, e.g., cloud formation and microphysical processes (Casanueva et al., 2016; Rajczak and Schär, 2017; Shrestha et al., 2017; Knist et al., 2018; Yang et al., 2019).

## 4 Comparison and discussion of the modelled and reconstructed climate

To evaluate the $LGM_{LGM}$ climate simulation, we compared temperature and precipitation to pollen-based reconstructions. Wu et al. (2007) provided reconstructions of temperature and precipitation for the coldest and warmest months of the LGM at 14 sites in Europe. Thus, we considered 56 samples (14 sites $\times$ 2 variables $\times$ 2 months) in this comparison. For the model-proxy comparison, we use the nearest model grid point to the pollen site and consider the model and proxy reconstruction to agree when the model-based anomaly is within the 90 % confidence interval of the pollen-based anomaly (more details about the

proxies in: Wu et al., 2007). Note that the simulated temperature and precipitation are anomalies with respect to $PD_{PD}$ and that January and July values are selected to mimic the coldest and warmest months.

In general, cooler and drier anomalies are observed in the $LGM_{LGM}$ with especially pronounced cooling in January and drying in July (Fig. 5). This resembles the proxy evidence given by the pollen-based reconstruction of Wu et al. (2007). In January, we observe a positive precipitation anomaly of up to 7 mm day$^{-1}$ over the Iberian Peninsula, northern Italy and

270 the Dinaric Alps (Fig. 5c). Overall, the $LGM_{LGM}$ climate agrees with the pollen-based paleoclimate reconstructions at three quarters of the 56 samples.

Still, some samples, e.g., over the Iberian Peninsula, show considerable differences between the pollen-based and model-based climate anomalies, in line with similar findings mentioned in earlier studies (e.g., Beghin et al., 2016; Ludwig et al., 2016; Cleator et al., 2020). These differences can be associated with shortcomings within the GCM-RCM modelling chain and/or

275 uncertainties in the proxy reconstructions (Bartlein et al., 2011; Ludwig et al., 2019; Cleator et al., 2020). Kageyama et al. (2006) suggested that terrestrial paleoclimate proxies may be more sensitive to climatic extremes than to the climatological mean state, which could partly explain the discrepancies between pollen-based reconstructions and the model simulations. One important model-proxy disagreement is the precipitation anomaly over the Iberian Peninsula in January. Based on evidence for the presence of certain tree species in the northwestern part of the Iberian Peninsula, Roucoux et al. (2005) suggested that

the LGM was not necessarily the period of the most severe, i.e., cold and dry, climatic conditions everywhere. Roucoux et al. (2005) and Ludwig et al. (2018) also suggested that this region during LGM sensu strictu was warmer and wetter than the end of Marine Isotope Stage 3 (MIS3, ca. 23 ka; Voelker et al., 1997; Kreveld et al., 2000) and the start of the Heinrich event 1 (H1, ca. 19 ka; Sanchez Goñi and Harrison, 2010; Álvarez-Solas et al., 2011; Stanford et al., 2011). This could be a hint that model-proxy comparison fails because the proxies refer to $21 \pm 2$ ka (Wu et al., 2007), i.e., either the end of MIS3 or

beginning of H1. Compared to the pre-industrial period, Beghin et al. (2016) found evidence in a model-proxy comparison that the interior and northwestern Iberian Peninsula experience wetter conditions during the LGM. These wetter conditions can be explained by a southward shift in the North Atlantic storm track during LGM compared to PD as suggested by many studies (e.g., Hofer et al., 2012a; Luetscher et al., 2015; Merz et al., 2015; Ludwig et al., 2016; Wang et al., 2018; Raible et al., 2020; Lofverstrom, 2020). Note further that we had only two pollen-based quantitative climate reconstructions from Iberia for the

LGM; we therefore consider the model-proxy intercomparison in this region equivocal.

## 5   Comparison and discussion of the modelled and reconstructed land cover

To evaluate the $LGM_{LGM}$ and land cover simulation, we compare the simulated tree cover with pollen-based biome reconstructions from the BIOME6000 data product (Prentice and Jolly, 2000; Wu et al., 2007) and with a newer synthesis by Kaplan et al. (2016). For the purposes of this comparison, we define tree cover as the fraction of ground covered by trees at each grid point excluding herbaceous and grass, whose value varies between 0 and 100 %.

The $LGM_{LGM}$ simulation generally shows low values for vegetation cover fraction (Fig. 2d), which reflects lower temperatures, reduced precipitation, and lower global atmospheric $CO_2$ concentrations that were present at the LGM compared to the Holocene (Gerhart and Ward, 2010; Woillez et al., 2011; Chen et al., 2019; Lu et al., 2019). Our simulated $LGM_{LGM}$ land cover is generally in good agreement with the pollen-based biome reconstructions (Fig. 2b). We interpret the pollen reconstructions of steppe vegetation as sparsely vegetated in the WRF land cover categories (Niu et al., 2011). Using the nine nearest 18 km grid points surrounding each pollen site to compare the model results with pollen-based reconstructions of the land cover categories, we define good model-proxy agreement when at least one of the grid points matches the proxy reconstruction. For example, the dominant land cover category northwest of the Alps (47.73° N, 6.5° E) reconstructed from pollen (steppe) agrees with the surrounding simulated land cover (sparse vegetation). For the Carpathian Basin, an area with few proxy reconstructions, the modelled LGM land cover categories are tundra and grassland, which is in agreement with results found by Magyari et al. (2014a,b). Additionally, we simulate an extended area of tundra categories (i.e., wooded and mixed tundra) between the Alps and the Fennoscandian ice sheet which can be considered as the northernmost ice-free area of Europe. Similarly, Kjellström et al. (2010) simulated an extended area of tundra-like vegetation in the northernmost ice-free areas of Europe for Marine Isotope Stage 3.

We further compared tree cover fraction simulated by LPJ-LMfire with a reconstruction of relative landscape openness from 71 pollen sites across Europe containing samples securely dated to the LGM based on a compilation by Davis et al. (2015) and Kaplan et al. (2016). This compilation represents a substantial improvement in spatial coverage and dating precision compared to the 14 sites of BIOME6000 used by Wu et al. (2007). Comparison between modelled tree cover and relative landscape openness is shown in Fig. 6. Generally, LPJ-LMfire moderately underestimates tree cover compared with the pollen-based openness reconstructions. Modelled tree cover has a maximum value of about 60 %, while there are eight sites where the relative tree cover reconstruction is > 60 %, and two samples with 100 % arboreal pollen percentage. As noted by Kaplan et al. (2016), these sites with very high reconstructed tree cover fraction should be treated with caution because they may represent locations with very little vegetation, e.g., at the edge of the Alpine ice sheet or at high-altitude in the Carpathian Mountains. In high mountain areas where we expect local vegetation to be very sparse if present at all, the pollen signal in sedimentary bodies may be dominated by the long-distance transport of tree pollen; this phenomenon is also observed in the analysis of pollen trapped in glacier ice (Brugger et al., 2019). At the bulk of the sites, LPJ-LMfire simulates 10-20 % lower tree cover than the relative tree cover inferred by the pollen. While this discrepancy is well within the uncertainty of both datasets and could be related to the calibration of arboreal pollen percentage with tree cover (Kaplan et al., 2016), it could also suggest

that the modelled climate is too cold and/or too dry, or that the LPJ-LMfire model is too sensitive to lower atmospheric $CO_2$
concentrations.

## 6 Influence of external forcing and land cover on climate

We assess the atmospheric response to changes in the entire climate system, in external forcing, and in land cover, separately, to better understand the importance of the land surface for the LGM climate in Europe. Namely, $LGM_{LGM}$ is compared to $PD_{PD}$ to determine the atmospheric response to *complete* LGM conditions. Then, we investigate the atmospheric response to changes
in orbital forcing by comparing $LGM_{PD}$ with $PD_{PD}$. Finally, the differences between $LGM_{LGM}$ and $LGM_{PD}$ determine the atmospheric response to changes in land cover. Our assessment considers the land areas without snow/ice that are shared by both LGM and PD climate, i.e., we discard glaciated areas and land areas on the continental shelves that were exposed at the LGM. Temperature and precipitation are selected as main indicators of the atmospheric response and latent and sensible heat fluxes as secondary indicators. Note that we use a two-tailed bootstrapping technique with a significance level of 2 % to assess
the significance of the differences (Sect. 2.2), which is illustrated by bold numbers in Table 3.

Comparing $LGM_{LGM}$ to $PD_{PD}$ shows a statistically significant cooling of -11.99 °C in the annual value (Table 3). This cooling is significantly enhanced to -15.34 °C in DJF (December-January-February), remains similar to the annual mean in MAM and SON (March-April-May and September-October-November), and significantly weakens to -7.24 °C in JJA (June-July-August; Table 3). This clearly illustrates a seasonality in the temperature response to *complete* LGM conditions ($LGM_{LGM}$
minus $PD_{PD}$). Broccoli and Manabe (1987) mentioned that one reason for the seasonality in the temperature response can be the fluctuations in the horizontal thermal advection from glaciers and ice sheets to ice-free regions, predominantly in winter. Additionally, we find a statistically significant dryness in the annual value of around -0.67 mm day$^{-1}$ when comparing $LGM_{LGM}$ to $PD_{PD}$. A significant drying is evident in most months, in particular in summer months, where precipitation is reduced by -1.55 mm day$^{-1}$. Only in the winter months, we observe a marginal increase in precipitation (Table 3). Cao et al.
(2019) on one hand attributed the overall decrease of precipitation to the strong anticyclonic circulations over the ice sheets during LGM compared to PD, especially to the low-level divergent cold air (Schaffernicht et al., 2020). On the other hand, Luetscher et al. (2015) and Lofverstrom (2020) found wetter conditions in southern parts of Europe in LGM wintertime and they attributed them to atmospheric rivers and Rossby-wave breaking, respectively. This together with the LGM southward shift of the storm track (found by Hofer et al., 2012a; Luetscher et al., 2015; Ludwig et al., 2016; Wang et al., 2018; Raible
et al., 2020) could then compensate an expected dryness in wintertime (i.e., $LGM_{LGM}$ minus $PD_{PD}$), which would not only affect the statistical significance in wintertime, but also lead to the seasonality in the precipitation response to *complete* LGM conditions. The comparison ($LGM_{LGM}$ minus $PD_{PD}$) also shows a statistically significant decrease of latent heat flux in the annual value (-25.63 W m$^{-2}$), which is true for most months and particularly strong for JJA (-52.47 W m$^{-2}$). Moreover, we observe a statistically significant increase in sensible heat flux of 7.48 W m$^{-2}$ (Table 3). This increase is strongest in JJA when
it reaches an addition of 33.97 W m$^{-2}$ and weakest in SON as we find a small but still significant increase of 2.69 W m$^{-2}$. A statistically significant decrease in sensible heat flux of -4.30 and -2.44 W m$^{-2}$ is simulated in DJF and MAM, respectively.

To further understand the atmospheric response, we investigate the role of the forcing (i.e., $LGM_{PD} - PD_{PD}$) and the land cover (i.e., $LGM_{LGM} - LGM_{PD}$), separately. The temperature response is clearly dominated by changes in the forcing. Changes in land cover can only slightly influence temperature by an additional cooling of 0.66 °C in MAM and a warming of 0.85 °C in JJA, both statistically significant (Table 3). Similarly, Jahn et al. (2005) found that the LGM-like vegetation cover produces colder temperatures (ca. $-0.6$ °C globally), especially in areas with the greatest decrease in tree cover. The precipitation anomalies are also dominated by changes in the forcing, whose values are statistically significant except in DJF, but also changes in the land cover contribute to a reduction in precipitation, especially in MAM (significant reduction of 0.09 mm day$^{-1}$) and JJA (reduction of 0.40 mm day$^{-1}$). The response of the latent heat flux is also dominated by changes in the forcing with statistically significant values. Changes in the land cover moderately influence the latent heat flux by an additional reduction of 8.06 W m$^{-2}$ in the annual mean, while changes in land cover account for almost half of the reduction in the latent heat flux in JJA (-24.33 W m$^{-2}$). Moreover, the response of the sensible heat flux is dominated by changes in the orbital forcing in the annual mean, JJA and SON. Modifications in land cover only dominate DJF and MAM by an additional significant reduction of 4.40 W m$^{-2}$ and 8.19 W m$^{-2}$, respectively. Still, changes in the land cover influence summer sensible heat by an additional increase of 14.95 W m$^{-2}$.

The analysis so far demonstrates that the seasonality of the atmospheric response is overall driven by changes in the forcing but its intensity can be modulated by changes in the land cover, in particular in the latent heat flux in JJA and sensible heat flux in DJF, MAM and JJA. A possible reason for the modulated intensity in the response may be a modification of the stability in the lowest levels of the atmosphere that is produced by the changes in the land cover. A cooling (warming) in the lower layer may lead to an inversion (unstable) zone that therefore weakens (enhances) precipitation processes. Another reason is that the differences in land cover lead to modifications in available moisture coming from the surface, i.e., evapotranspiration or latent heat. A reduction in latent heat is interpreted as reduced availability of surface moisture, which leads to a reduction of precipitation. Ludwig et al. (2017) suggested that including LGM-like vegetation into regional climate models causes changes in heat fluxes that lead to impacts on temperature and precipitation. Based on a similar coupling design, Strandberg et al. (2011) found that the impact of a different land cover on LGM climate simulations is small compared to the uncertainties in the proxy reconstructions. Even though this is also true in our study, our results and discussion suggest that modifications in land cover like deforestation could play an important role when other forcing agents marginally change, as it is observed in some climate change scenarios like RCP 2.6 and 4.5 (Strandberg and Kjellström, 2019; Davin et al., 2020; Jia et al., 2020).

To obtain a more detailed understanding of the atmospheric response to changes in land cover ($LGM_{LGM} - LGM_{PD}$), we further analyse the differences in the spatial patterns in January and July to be consistent with the evaluation done in Sect. 5. We focus on temperature at 2 m, precipitation and latent and sensible heat fluxes. We use a two-tailed bootstrapping technique with a significance level of 2 % to assess the significance of the differences at each grid point (Sect. 2.2), which is illustrated by crosshatched areas in Fig. 7 and 8.

The annual mean temperature shows a statistically significant cooling of around 2 °C in the vicinity of glaciers and in high-altitude regions; while a statistically significant warming is visible in lower-elevation areas including the southwestern part of the Iberian Peninsula, France and the Carpathian Basin. (Fig. 7a). A similar spatial pattern is observed for January and July

temperatures: A significantly stronger warming is evident for the northern part of Italy in January (Fig. 7b), whereas the rest of the continent does not show significant changes. In July, the amplitude of the temperature anomaly becomes significantly stronger, especially where the positive temperature anomaly covers a large area, e.g., over eastern Europe (Fig. 7c). The precipitation response is moderate in the annual mean. A general and statistically significant decrease is observed over the rest of Europe. Changes in January precipitation are overall insignificant, except for some areas in eastern Europe where a significant dryness is observed. LGM land cover leads to a negative and statistically significant precipitation anomaly in July, which is especially strong around the Alps and in eastern Europe. The response of the latent heat flux is also moderate in the annual mean (Fig. 8a). We observe a general and statistically significant reduction, especially in eastern Europe. A similar significant but weakened pattern is observed in January, which even shows few small areas with an increase in latent heat flux (Fig. 8b). In July, a stronger reduction in the latent heat flux is observed with largest reductions around the Alps and over eastern Europe (Fig. 8c). Note that some areas with strong increases in the latent heat flux (reddish) are associated with large PD urban areas. Moreover, the annual mean sensible heat flux shows a statistically significant reduction of about 30 W m$^{-2}$ around mountainous areas, i.e., Pyrenees, Alps and Carpathian Mountains (Fig. 8d), while a statistically significant increase of sensible heat is visible in lower-elevation areas, especially over France and some areas in eastern Europe (Fig. 8d). In January, the pattern of the sensible heat flux is overall moderately reduced (still statistically significant, Fig. 8e). In July, we find an enhanced amplitude of the sensible heat flux with small changes in the spatial pattern with respect to the annual one: There is an additional statistically significant decrease of sensible heat flux by around 60 W m$^{-2}$ around mountainous areas except most of the Carpathian Mountains (Fig. 8f). A statistically significant increase of sensible heat flux dominates the rest of Europe with values up to 40 W m$^{-2}$ in some areas over central and eastern Europe.

Even though changes in land cover have a small-to-moderate effect on the response of temperature, precipitation and the latent and sensible heat fluxes (Table 3), their spatial pattern strongly changes across Europe (Fig. 7). Important spatial changes are statistically significant over eastern Europe in July. Strandberg et al. (2011) and Kjellström et al. (2010), in similar coupling designs, compared glacial simulations using two land cover settings and found that the simulated regional climate patterns in parts of Europe are sensitive to feedbacks from large differences in vegetation. Particularly, Kjellström et al. (2010) found that glacial-like vegetation leads to warmer conditions over Eastern Europe compared to modern vegetation. Strandberg et al. (2014) showed in their RCM experiments for the Holocene that summer temperature and precipitation are sensitive to changes in land cover in eastern Europe due to evapotranspiration (in our results as latent heat) feedbacks (see Fig. 8 in Strandberg et al., 2014). They found that a reduction in tree cover leads to warmer and drier summers in eastern Europe, which is similar to our finding as we observe that a reduction of vegetation cover fraction is associated with a warmer and drier July in the same region. This suggests that the land-atmosphere coupling-strength may be stronger in eastern Europe compared to other parts of Europe, especially during summer.

## 7 Conclusions

In this study, we investigated the importance of land-atmosphere feedbacks for the climate of Europe during the Last Glacial
Maximum. To this end, we performed a series of high-resolution asynchronously coupled atmosphere-vegetation model simulations. We simulated the European climate and vegetation using the WRF regional climate model and LPJ-LMfire vegetation model with a 54 and an 18 km horizontal grid spacing.

Results of the asynchronous coupling show that quasi-equilibrium between climate and land cover is reached after the fourth to fifth iteration. Between the first and fourth iteration, the climate becomes progressively wetter in southern Europe, while it becomes drier in eastern Europe. Once the coupled model system reaches quasi-equilibrium (from fourth to seventh iterations), we identified only marginal spatial differences that can be attributed to internal variability in the climate and vegetation models. The final iteration of the asynchronous coupling represents our best estimate of the atmospheric and land-surface conditions in Europe at the LGM. Consistent with many previous studies (e.g., Wu et al., 2007; Bartlein et al., 2011; Újvári et al., 2017; Cleator et al., 2020), we observe that the LGM climate of Europe was generally much colder and drier compared to PD. The $LGM_{LGM}$ land cover was characterised by tundra and sparse vegetation, although open forest parkland (transition from grass to forest during the LGM) may have been common in many parts of central Europe, which is supported by comparisons with pollen-based vegetation reconstructions.

Using two additional sensitivity simulations: $PD_{PD}$ and $LGM_{PD}$, we quantified the direct effects of external forcing and land cover on the LGM climate. Comparing $LGM_{LGM}$, i.e. the complete LGM conditions, to $PD_{PD}$ shows not only a general cooling and drying, but also a seasonality in the atmospheric response. Comparing $LGM_{PD}$ to $PD_{PD}$ illustrates that the seasonality is mainly driven by changes in forcing. The comparison between $LGM_{LGM}$ to $LGM_{PD}$ shows that, even in Europe where we would generally expect a weak land-atmosphere coupling compared, e.g., to the monsoon tropics, the atmosphere is sensitive to changes in land cover. The land-atmosphere response also has a seasonality which differs across Europe with a stronger coupling-strength in eastern Europe. These features can be partially explained by the variable spatial and temporal influence of vegetation cover (albedo) and heat fluxes (sensible and latent heat fluxes) to the lower troposphere. Our results show that dry conditions in LGM are partially attributed to LGM land cover as a reduction in vegetation overall led to stronger dryness compared to PD land cover. This is particularly true for central and eastern Europe during summer.

An evaluation of the modelled $LGM_{LGM}$ climate should be performed with independent paleoclimate reconstructions from more sites than the 14 published points that are in the spatial domain of this study. Since the publication of Wu et al. (2007) and Bartlein et al. (2011), more than 70 well-dated pollen records from Europe that cover the LGM have become available (Kaplan et al., 2016). However, these data have not been transformed into paleoclimate reconstructions to-date and such an effort would be beyond the scope of the current study. Additionally, as more paleoenvironmental reconstructions become available in the future, these simulations will be worthy of further evaluation and more detailed examination of specific areas. For instance, future work that improves pollen-based land-cover reconstructions, e.g., using multi-proxy approaches that combine pollen data with presence-absence information from DNA (e.g., Alsos et al., 2020), will be very valuable for quantitative evaluation of model results with using paleoenvironmental data. Although 18 km is a relatively high grid spacing for regional climate

models, future studies will benefit from even more detailed climate simulations, particularly to better understand precipitation patterns in complex terrain such as Iberia, across the Mediterranean, and in the Carpathians. This is also true for studies on the local and regional paleobotany and archaeology of this important period in Europe's history.

*Code and data availability.* WRF is a community model that can be downloaded from its web page (http://www2.mmm.ucar.edu/wrf/users/code_admin.php, last access 12 October 2020) (Skamarock and Klemp, 2008). The source code of LPJ-LMfire can be downloaded from Github (https://github.com/ARVE-Research/LPJ-LMfire/tree/v1.3, last access: 04 November 2020) (Kaplan et al., 2018). The climate simulations (global: CCSM4 and regional: WRF) and land cover simulations (LPJ-LMfire) occupy several terabytes and thus are not freely available. Nevertheless, they can be accessed upon request to the contributing authors. Simple calculations carried out at a grid point level
are performed with Climate Data Operator (CDO, Schulzweida, 2019) and NCAR Command Language (NCL, UCAR/NCAR/CISL/TDD, 2019). The figures are performed with NCL (UCAR/NCAR/CISL/TDD, 2019). Source code of the program to classify vegetation cover fraction into the WRF land cover categories is archived on Github (https://github.com/ARVE-Research/lpj2wrf).

*Author contributions.* PV, JOK, and CCR contributed to the design of the experiments. PV carried out the climate simulations and wrote the first draft. JOK carried out the land cover simulations. P.L. provided the guidelines for introducing new land-cover and LGM boundary
conditions into WRF. M.M. provided support in the application of these guidelines. All authors contributed to the writing and scientific discussion.

*Competing interests.* The authors declare no competing interests.

*Acknowledgements.* This work was supported by the Swiss National Science Foundation (SNF) within the project 'Modelling the ice flow in the western Alps during the last glacial cycle' (grant: 200021-162444). JOK is grateful for computing support from the School of Geography,
University of Oxford. The simulations are performed on the super computing architecture of the Swiss National Supercomputing Centre (CSCS). PL thanks the Helmholtz initiative REKLIM for funding. MM is supported by the Early Postdoc. Mobility program (SNF, grant: P2BEP_181837). Data is locally stored on the oschgerstore provided by the Oeschger Center for Climate Change Research (OCCR). This study contributes to the PALEOLINK project as part of the PAGES 2k Network.

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

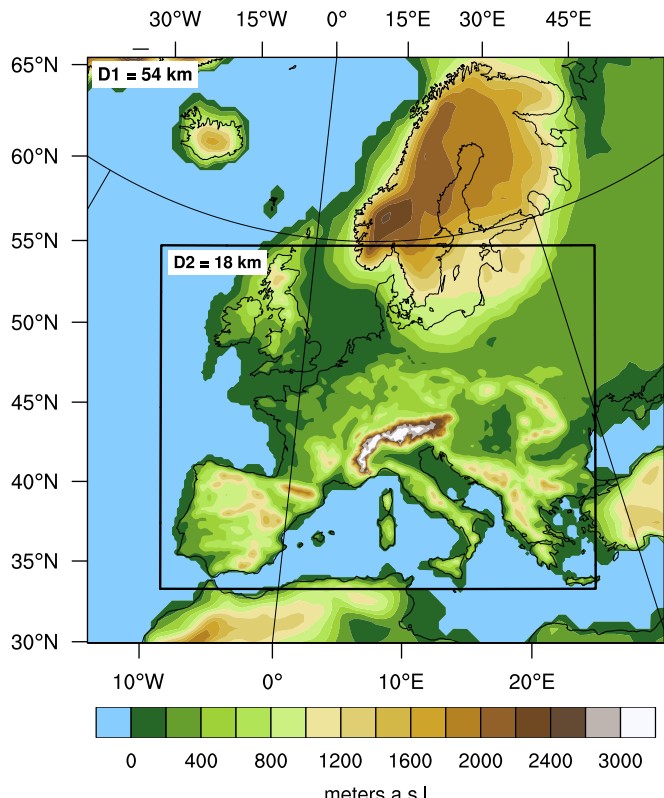

**Figure 1.** Topography and the two domains for the WRF LGM simulations.

**Table 1.** Set of simulations used in the asynchronous coupling and sensitivity experiments. First column indicates the name of the simulation, second and third columns the forcing used in the global and regional climate models, and fourth column the purpose of the comparison.

| Name | GCM simulations | RCM simulations | | Aim |
|---|---|---|---|---|
| | (Hofer et al. 2012a) | topography and other forcing | land cover | insights into the responses to changes in the: |
| $PD_{PD}$ | 1990s | 1990s | 1990s | forcing |
| $LGM_{PD}$ | LGM | LGM | 1990s | |
| $LGM_{LGM}$ | LGM | LGM | LGM | land cover |

**Table 2.** Variables passed between CCSM4/WRF and LPJ-LMfire.

| CCSM4/WRF to LPJ-LMfire | |
|---|---|
| 30-year monthly values | |
| mean temperature at 2 m | convective available potential energy |
| daily max. temperature at 2 m | horizontal wind velocity at 10 m |
| daily min. temperature at 2 m | precipitation (liquid and solid) |
| total cloud cover fraction | |

| LPJ-LMfire to WRF | |
|---|---|
| 30-year monthly values | climatological value |
| vegetation cover fraction | land cover fraction (category) |
| leaf area index | dominant land cover type (category) |
| | deep-soil temperature |

# Land Cover

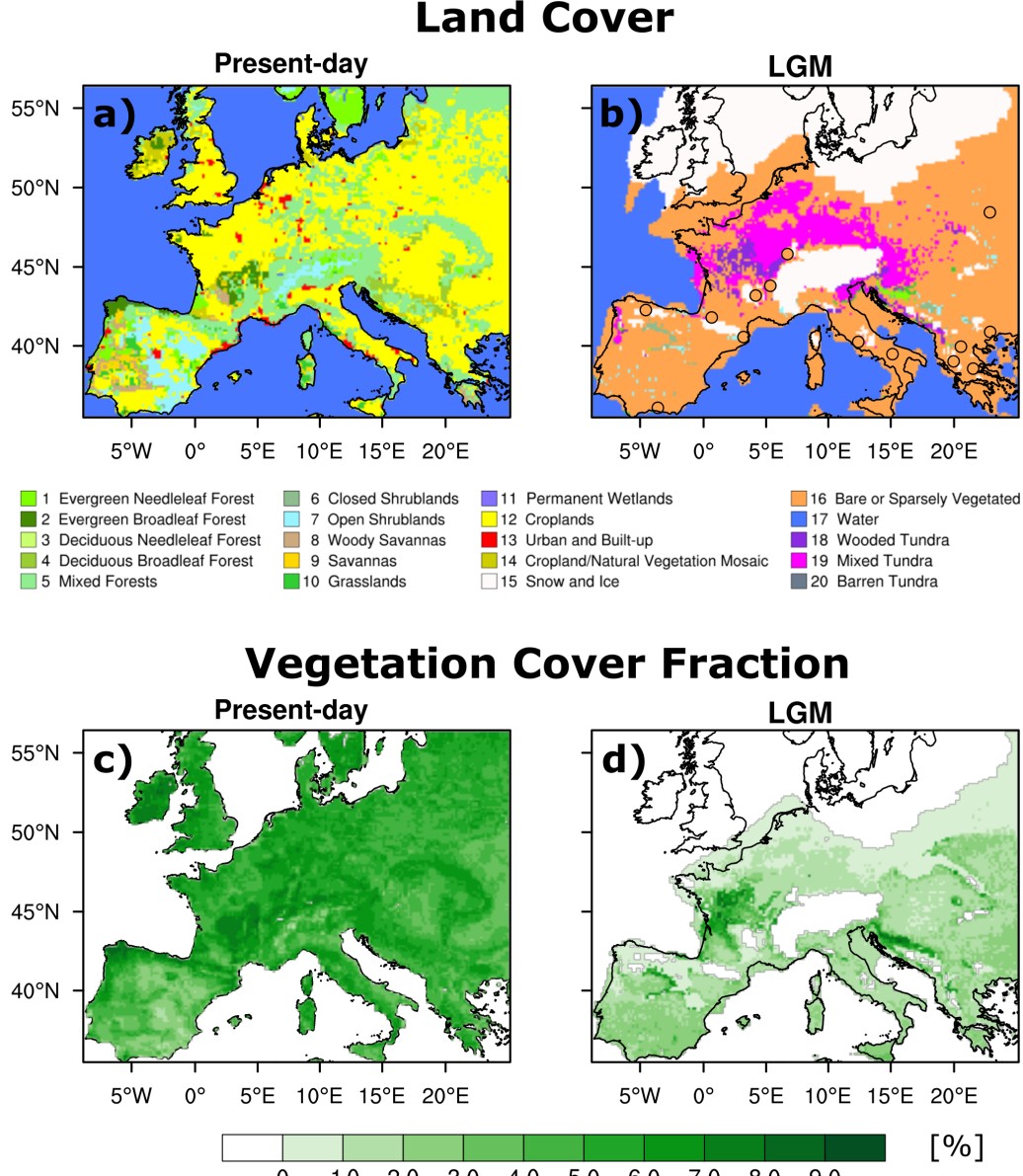

**Figure 2.** Land cover used by WRF. Panel (a) represents the dominant land cover category during PD. Panel (b) as (a) but during the LGM. Panels (c) and (d) as (a) and (b) but for vegetation cover fraction. Circles in (b) represent proxy evidences from Wu et al. (2007).

# Spatial Values

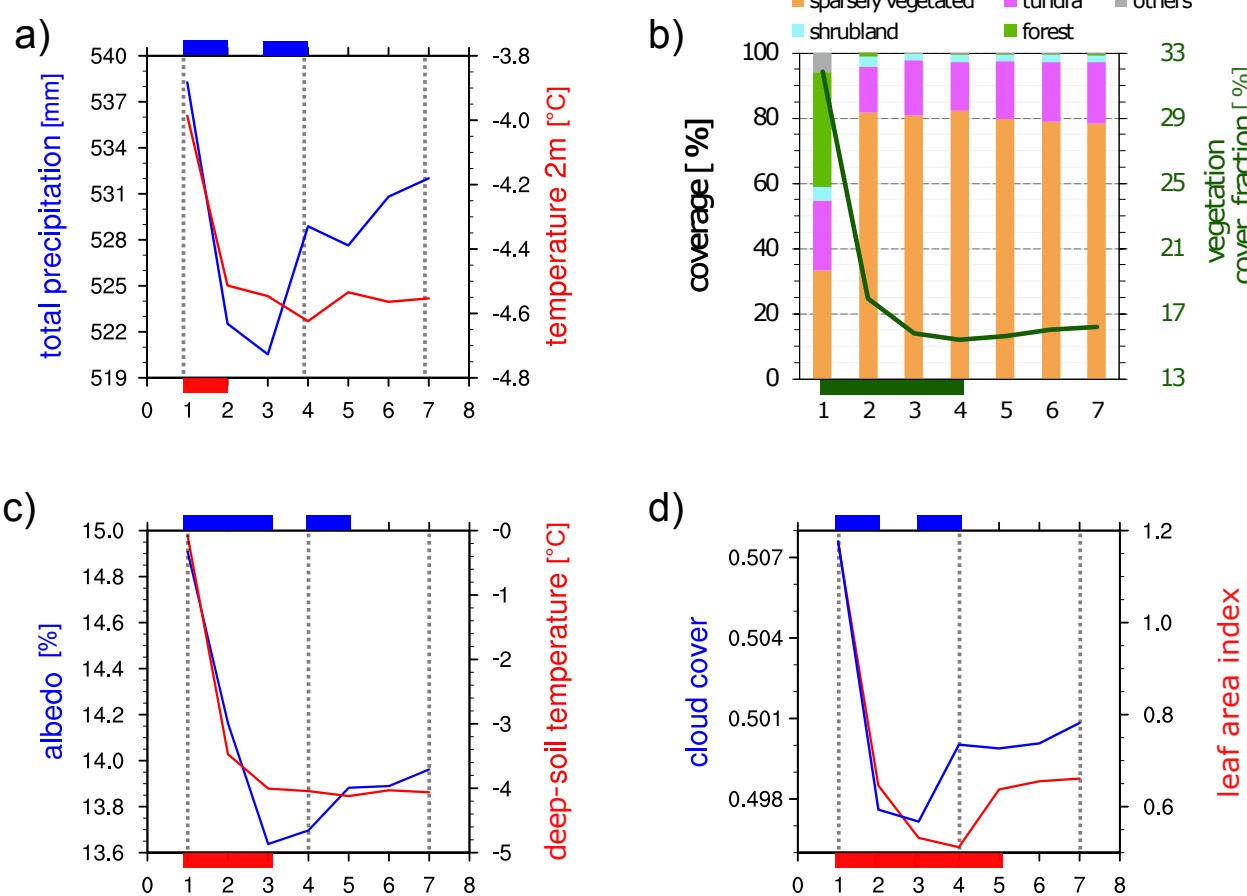

**Figure 3.** 30-year spatial climatology of annual mean values throughout the iterations. Panel (a) represents total precipitation (blue line) and temperature at 2 m (red line), (b) the percentage spatial fraction of bare (orange), tundra (pink), shrubland (sky blue), forest (light green), others (grey), and the spatial mean value of vegetation cover fraction (dark green line), (c) as (a) but for albedo and deep-soil temperature, and (d) as (a) but for cloud cover and leaf area index. The grey dotted lines in (a, c and d) represent the first, fourth and seventh iterations. Blue, red and green boxes represent statistically significant differences between iterations at a 2 % significance level (using a two-tailed bootstrapping technique).

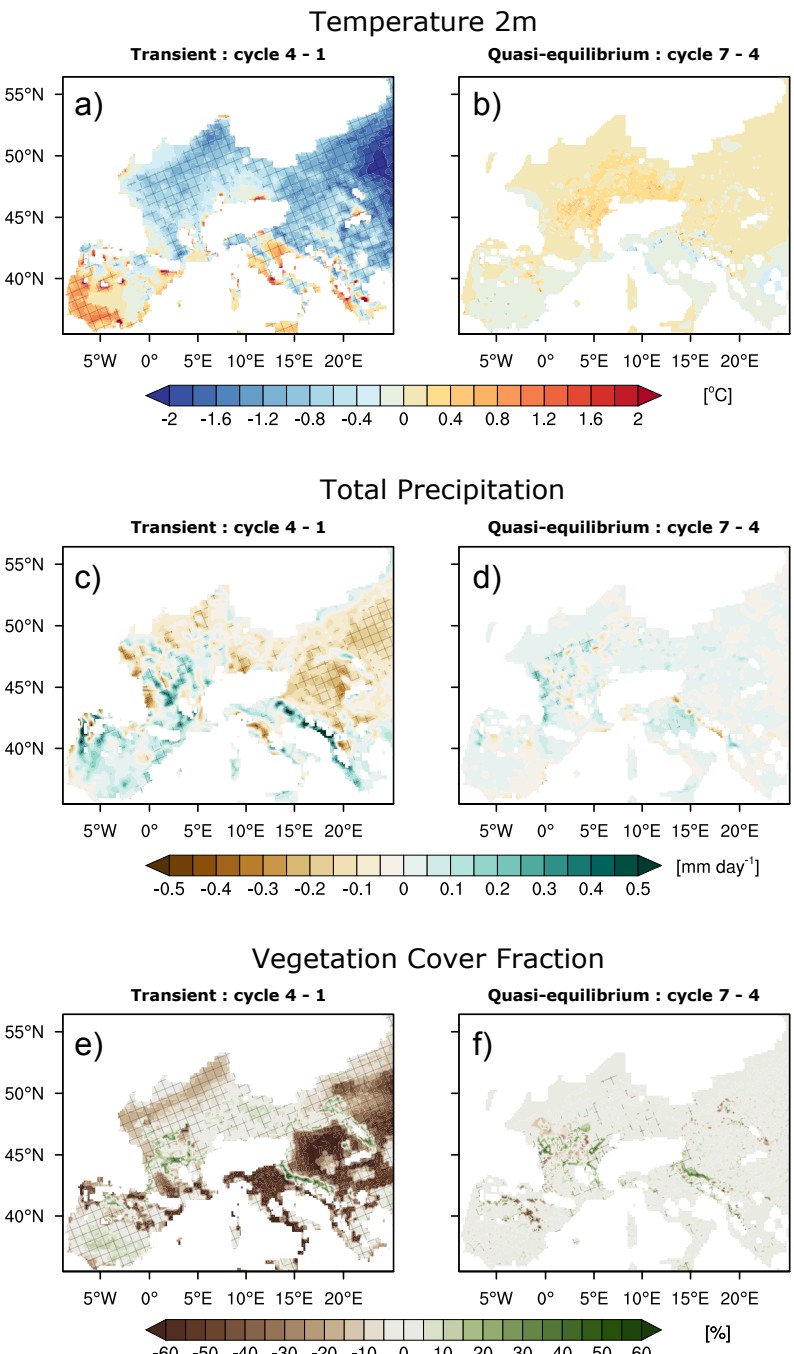

**Figure 4.** Differences in 30-year mean values. Panel (a) represents the difference in temperature at 2 m between the first and fourth iteration (transient), (b) as (a) but between the fourth and seventh iteration (quasi-equilibrium). Panels (c)-(d) and (e)-(f) as (a)-(b) but for total precipitation and vegetation cover fraction, respectively. Masked out areas are in white. Crosshatched areas indicate statistically significant differences using a two-tailed bootstrapping technique with 2 % significance level.

# Temperature 2m

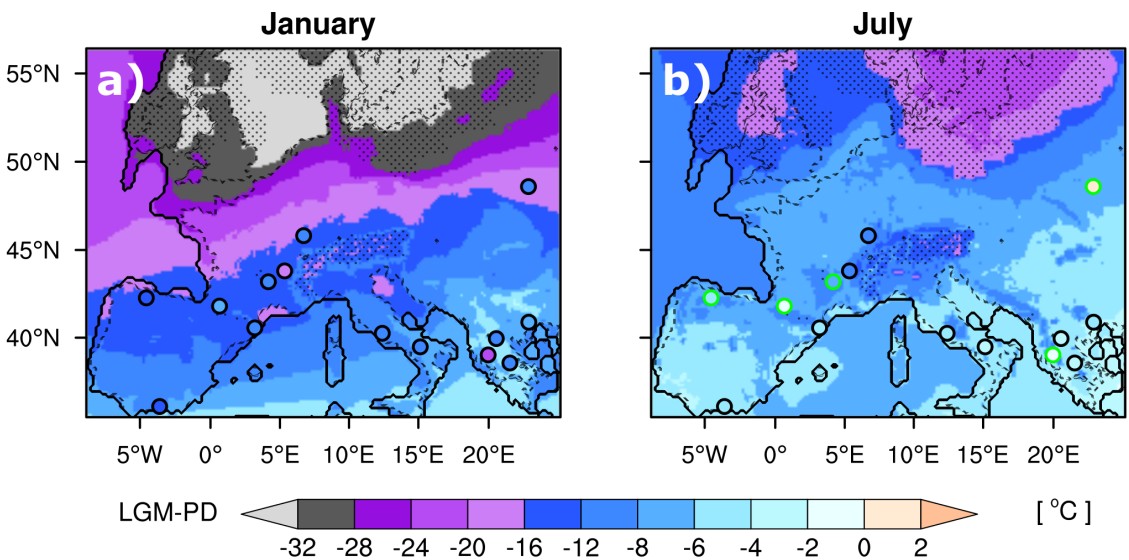

# Precipitation

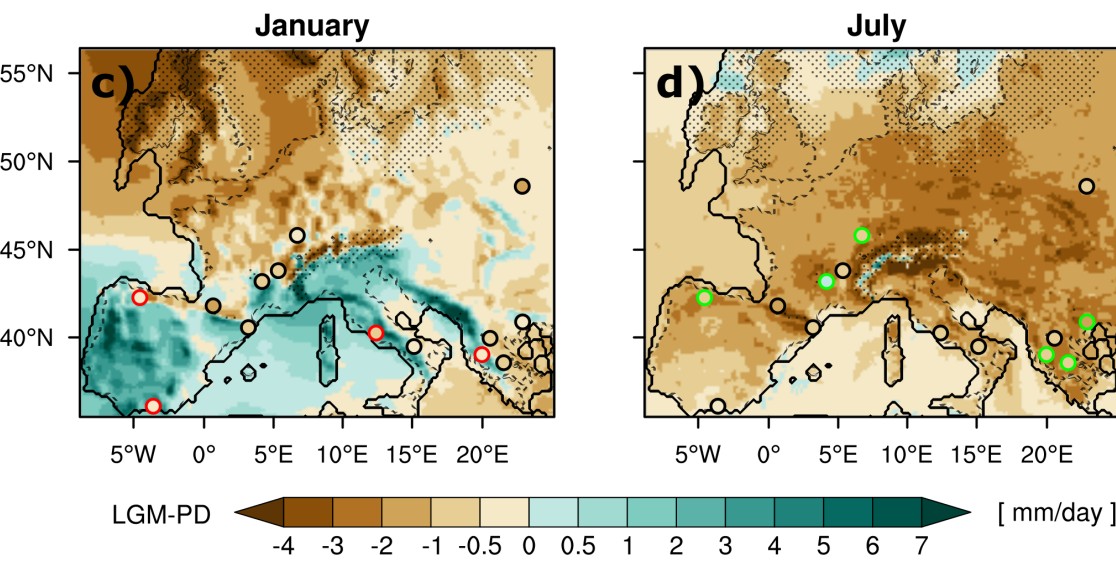

**Figure 5.** Changes in temperature and precipitation patterns. Panel (a) represents the differences in 30-year mean temperature between LGM and PD ($LGM_{LGM} - PD_{PD}$) for January. Panel (b) as (a) but for July. Panels (c) and (d) as (a) and (b) but for precipitation differences. Circles represent proxy evidences: a red (green) border indicates that the simulated value is significantly above (below) the proxy value at the closest grid cell of the model (outside the 90 % confidence interval, Wu et al., 2007). Solid line represents the LGM coastline, dashed line PD coastline and dots the area covered by glaciers.

# Tree Cover Fraction

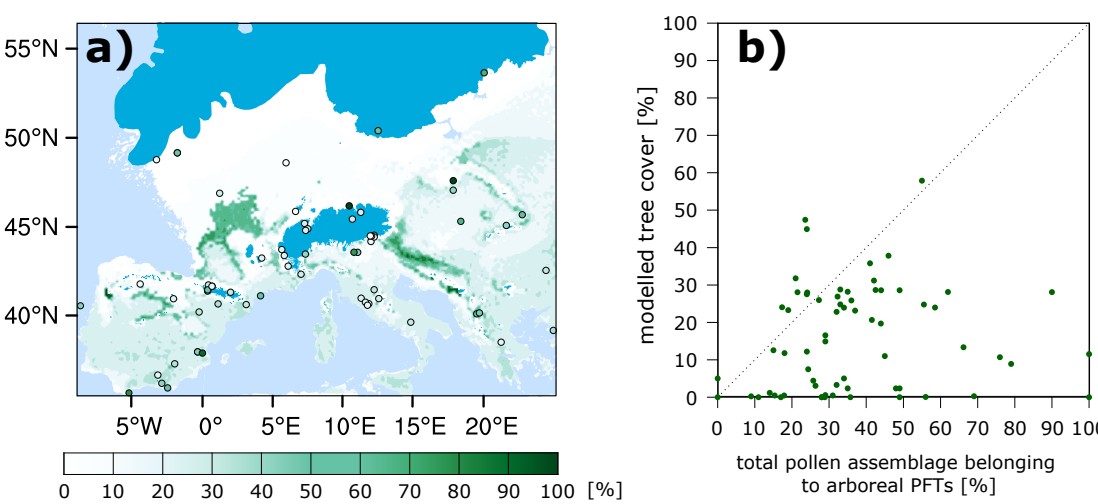

**Figure 6.** Comparison between modelled and reconstructed tree cover. Panel (a) shows the LPJ-LMfire simulated tree cover fraction from LGM$_{\text{LGM}}$. Circles represent the 71 pollen samples securely dated to LGM from Kaplan et al. (2016). Panel (b) shows a scatter plot of reconstructed vs. modelled LGM tree cover.

**Table 3.** Assessment of the atmospheric response using 30 years of simulated precipitation and temperature data. First column indicates the simulations, second column the annual response, and the other columns the response in each season. Numbers in bold represent statistically significant differences using a two-tailed bootstrapping and a significance level of 2 %. Note that the assessment considers land areas without snow/ice that are shared by both LGM and PD climate and discards the continental shelves exposed at the LGM.

| | Annual | DJF | MAM | JJA | SON |
|---|---|---|---|---|---|
| **Temperature response [°C]** | | | | | |
| $LGM_{LGM}$ - $PD_{PD}$ | **-11.99** | **-15.34** | **-13.85** | **-7.24** | **-11.53** |
| $LGM_{PD}$ - $PD_{PD}$ | **-12.06** | **-15.44** | **-13.19** | **-8.09** | **-11.52** |
| $LGM_{LGM}$ - $LGM_{PD}$ | 0.07 | 0.10 | **-0.66** | **0.85** | -0.01 |
| **Precipitation response [mm day$^{-1}$]** | | | | | |
| $LGM_{LGM}$ - $PD_{PD}$ | **-0.67** | 0.09 | **-0.86** | **-1.55** | **-0.37** |
| $LGM_{PD}$ - $PD_{PD}$ | **-0.53** | 0.16 | **-0.77** | **-1.15** | **-0.37** |
| $LGM_{LGM}$ - $LGM_{PD}$ | -0.14 | -0.07 | **-0.09** | -0.40 | 0 |
| **Latent heat response [W m$^{-2}$]** | | | | | |
| $LGM_{LGM}$ - $PD_{PD}$ | **-25.63** | **-6.09** | **-32.44** | **-52.47** | **-11.51** |
| $LGM_{PD}$ - $PD_{PD}$ | **-17.57** | **-5.34** | **-27.23** | **-28.14** | **-9.57** |
| $LGM_{LGM}$ - $LGM_{PD}$ | **-8.06** | **-0.75** | **-5.21** | **-24.33** | **-1.94** |
| **Sensible heat response [W m$^{-2}$]** | | | | | |
| $LGM_{LGM}$ - $PD_{PD}$ | **7.48** | **-4.30** | **-2.44** | **33.97** | **2.69** |
| $LGM_{PD}$ - $PD_{PD}$ | **7.59** | 0.10 | **5.75** | **19.02** | **5.48** |
| $LGM_{LGM}$ - $LGM_{PD}$ | **-0.11** | **-4.40** | **-8.19** | **14.95** | **-2.79** |

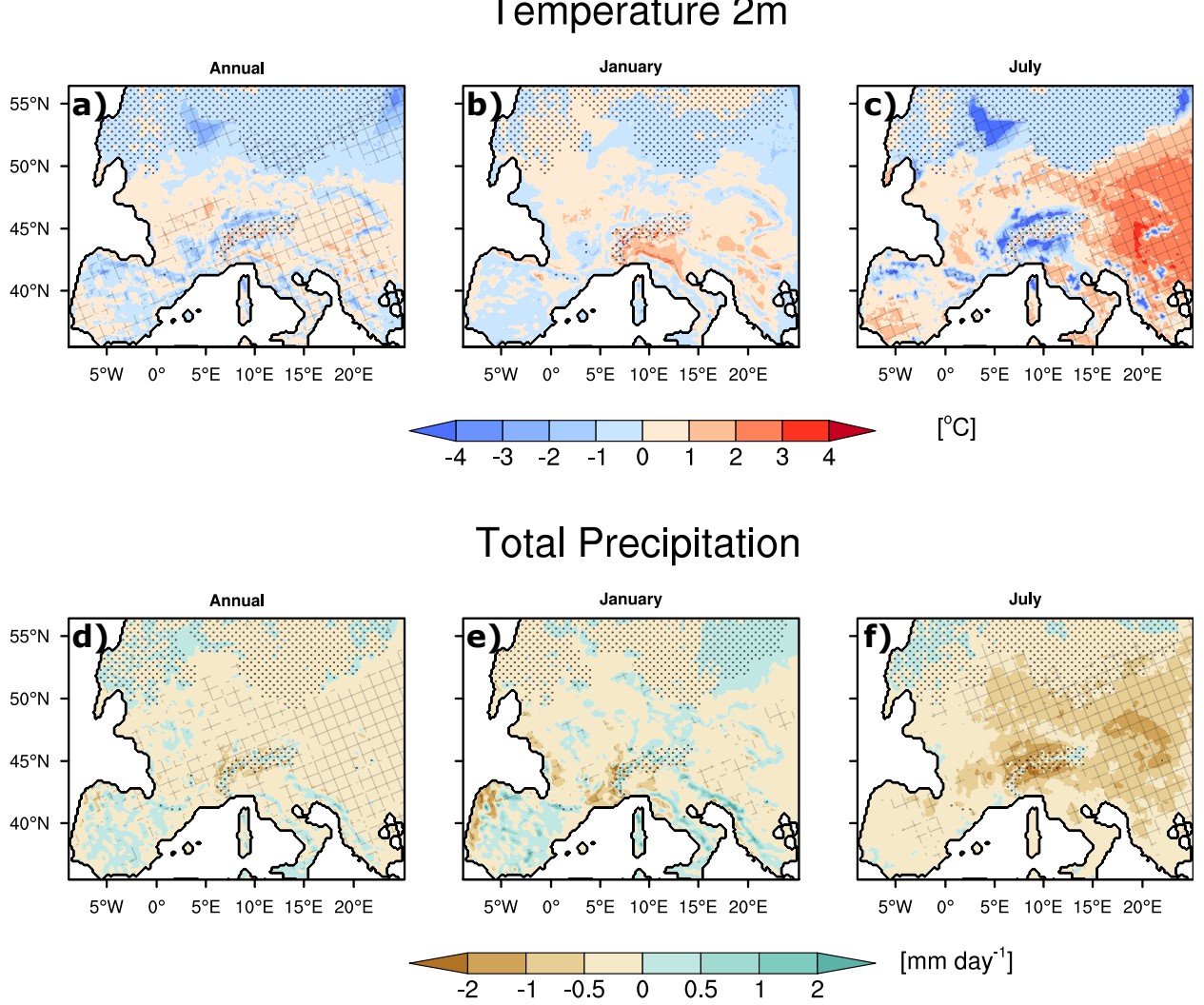

**Figure 7.** Atmospheric response to changes in the land cover. Panel (a) shows differences in the annual mean temperature between LGM$_{LGM}$ – LGM$_{PD}$. Panels (b) and (c) as (a) but for January and July, respectively. Panels (d), (e) and (f) as (a), (b) and (c) but for precipitation. The solid line represents the coastline during the LGM, stippled areas are covered by glaciers and crosshatched areas indicate statistically significant differences using a two-tailed bootstrapping technique with 2 % significance level.

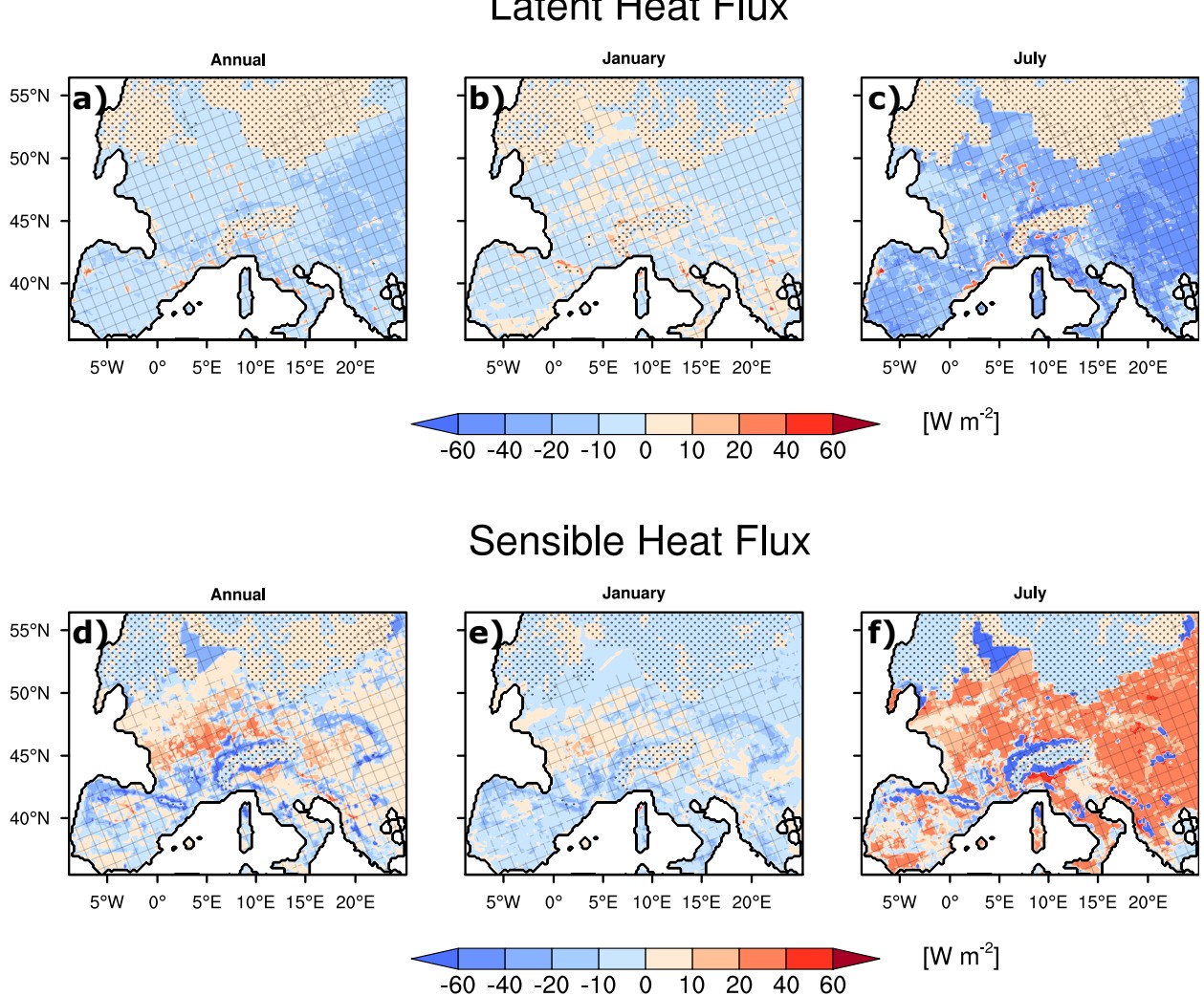

**Figure 8.** Atmospheric response to changes in the land cover. Panel (a) represents differences in the annual mean latent heat flux between $LGM_{LGM} - LGM_{PD}$. Panels (b) and (c) as (a) but for January and July, respectively. Panels (d), (e) and (f) as (a), (b) and (c) but for sensible heat flux. The solid line represents the coastline during the LGM, stippled areas are covered by glaciers and crosshatched areas indicate statistically significant differences using a two-tailed bootstrappping technique with 2 % significance level.