# Peer review of "The role of land cover on the climate of glacial Europe"

_Climate of the Past, 2020_

## Referee Comment (RC1) · Anonymous Referee #1 · 7 Dec 2020

Review of "The role of land cover on the climate of glacial Europe" by Velasquez et al. (cp-2020-147) from reviewer #1

Velasquez et al simulates LGM climate with a chain of asynchronous coupled models (GCM -> RCM -> DVM -> RCM etc). This is exactly what is needed in palaeo climate. High resolution enables better comparison with proxy data, and 18 km grid spacing is impressive in a palaeo context. A good description of vegetation is needed to get LGM conditions as realistic as possible. The authors make sure to have a vegetation compliant with the climate by making as much as 7 iterations with climate and vegetation models. This could be interesting for the readers of Climate of the past, but I think the paper need a lot of improvement before that. That the model set up is relevant is what makes me recommend major revision.

My major concerns are with: i) the lack of discussion of the role of vegetation on LGM climate, ii) the lack of context and comparison with previous studies, iii) how the authors describe results, but don't explain them or try to understand them iv) the structure of the text where results from different sensitivity studies are mixed and where results and discussions are mixed.

Detailed comments follow below. As always, I might have misunderstood some things, and my comments could be invalid because of that. If such misunderstandings do occur, think about if your paper is written in a clear enough way.

* Major comments

The title of the paper is "The role of land cover on the climate of glacial Europe", but I don't think I get any new insights by reading it. Despite the ambitious model set up. It seems you did some sensitivity runs, but don't know what to do with them, and that you didn't study the literature on vegetation climate interactions. It's not enough to just say that LGM was cold and that vegetation affects climate, we know that already. Why does it? If you want to advance climate modelling you need to give physical reasons for your results. Otherwise your results could just be a random effect caused by different initial conditions. Try to explain your results. Look at variables that might be relevant. Albedo and heat fluxes are not analysed at all.

The results of the RCM simulations are highly depending on the driving GCM. But you don't discuss your GCM runs at all. What is LGM vegetation like in the GCM? this explains many of the differences between cycle 1 and 2. What is LGM climate like in the GCM compared to other PMIP3 runs, colder, warmer, wetter, drier? What is the general circulation like? Could that explain the precipitation patterns?

Temperature differences between the different cycles are hardly discussed. Temperature decreases with almost 0.5 °C between cycles 1 and 2. This could perhaps be explained by a large albedo increase when the forest disappears (albedo is not shown). But temperature increases with 0.5 °C between cycles 3 and 4 despite no significant

vegetation changes. This is unexplained. Could it be that the differences are just a result of natural variability and not a result of vegetation at all?

A related issue to the above is the question of significance. The significance of the results are not tested here. Even though vegetation has a clear effect on climate the effect is small compared to the effect of different forcing, and sometimes also compared to natural variability. Therefore it is important to check if the results are significant. I feel that the paper misses to discuss some relevant previous studies, particularly Strandberg et al. (2011), henceforth S11. S11 studies LGM with more or less the same method as you do. The present model set up is an improvement from S11 so you shouldn't be afraid to discuss it. Why not get inspired by how S11 discuss the role of the GCM or the uncertainty in proxy data. Vegetation climate interactions is not the main number in S11, but there is a section about that too. Kjellström et al. (2010) used the same approach for another cold climate, MIS 3.

The structure of the paper makes it a bit difficult to follow. For example, section 4 "Comparison of the simulated land surface conditions to proxy reconstructions" deals in large parts also with vegetation differences between PD and LGM, difference in climate between PD and LGM, difference in climate between models and proxies and a discussion about simulated LGM climate in other studies. In a similar way section 5 "Atmospheric sensitivity to land cover" deals largely with differences between LGM and PD climate. I think it would be good if you could discuss one thing at the time.

* Minor comments

L10-14: I think these conclusions are too general given the title of the paper. Please quantify a bit and perhaps also explain why you get an effect

L11-12: "colder and drier", "warmer and drier". Is this LGM(LGM) - LGM(PD) or LGM(PD) - LGM(LGM)?

L13: "southward displacement". This sentence reads to me like "Even with a southward

displacement of the storm track regional climate is influenced by land cover". Is this what you mean? Why would a southward displacement counteract or diminish the influence of vegetation? Please rephrase.

L13: "increased importance of the Atlantic". What do you mean? In what way is the Atlantic more important? And why? I can't find anything about the importance of the Atlantic in he rest of the text.

L25: "Recent advances" It could be discussed how recent it is since S11 is 9 years old. See discussion above.

L45: There are better references to this than AR5, e.g. Rauscher et al., 2010; Di Luca et al., 2011; Prein et al., 2013; Iles et al., 2019 and Demory et al. 2020.

L52: Is Tao et a. (2013) the appropriate reference here? It's about the effect of vegetation on air quality in the US. If you want vegetation climate interactions in RCMs in palaeo climate (Consistent with Strandberg et al. (2011) and Ludwig et al. (2017)) I would recommend e.g. Kjellström et al. (2010) or Strandberg et al. (2014). If you want a more general reference on vegetation climate interactions you could start with Jia et al. (2019).

L61: There are two studies that use a similar approach as you, and in addition in cold climates: Kjellström et al. (2010) and Strandberg et al. (2011). Especially S11 would be worth to note as it simulates LGM.

L63-68: When I read this I understand that you have the following model chain: GCM->RCM->DVM->RCM-> etc. From section 2.4 I understand that the first DVM simulation is forced by the GCM. I suppose that the description in 2.4 is the correct one. Please, check and correct.

L72-75: Are these 31 years part of a longer simulation, in that case how were they selected? Is the LGM simulation a part of a transient simulation or is it steady state?

L74: Please add an explanation and a reference to these data models.

L87: 31 years were simulated by the GCM, 30 by the RCM. Why is one year not used? Which 30 years are used?

L88: "adaptive time-step to increase ... computer facilities" I don't understand this at all. What does it mean?

L89: If you divide 30 years into two 15 year periods and start every simulation with a two-month spin-up this will give you 2x(14y 10m) = 29 years and 8 months. How do you get 30 years of data from that?

L90: 15 days seems to me to be a bit short. How do you decide that quasi-equilibrium is reached? In Velasquez et al. (2019) I can only find the following: "Tests show that the WRF land scheme reaches a quasi-equilibrium after approximately 15 d." That doesn't explain much. I guess that spin-up time also depends on the season. When do you start your simulation?

L93: What do you mean with "perpetual" here? Do you mean steady-state with constant forcing?

L93: "Reduced sea level and increased ice sheets". This is somewhat ambiguous. I guess you mean that sea level was lover and ice sheets were larger than today. It could also mean that LGM conditions have been revised in the PMIP3 protocol compared to previous protocols. It's not entirely clear.

L115 Are the "reconstructed $CO_2$ concentrations" used in the RCM the same as used in the GCM (PMIP3 forcing)? It seems like an unnecessary complicated way to say that forcing is the same as in the GCM. If it's not the same, why not?

L120-122: What vegetation field is used in the first GCM run? See also comment on L63-68.

L133: I would say it's more correct to call this section "Results of the iterative...". You don't have other simulations to compare with so you can't estimate the effect of the coupling. You could, however, describe the results of your simulations, and that's what

you do.

L135: How do you determine that quasi-equilibrium is reached? Just by eyeballing or do you have a criterion for equilibrium. It seems like you just decide that equilibrium is reached, but how can you be sure without a proper metric?

L136: "this result and its effect" What result and effect? Please clarify.

L137: "variables that mostly govern" Do you mean variables that govern the interaction most of the time, but not always, or do you mean most of the variables that govern the interaction? Please clarify.

L138: "suitable to illustrate the asynchronous coupling" What does this mean? Please rephrase and clarify.

L140: Please define the green fraction. After several readings I'm still not sure what it is.

L147: "in all variables" In both variables?

L148-149: I don't agree that the difference in climate is explained by differences in resolution. I would say that it is the difference in vegetation. A decrease in forest fraction from 35 % to just a few will have an effect on the simulated climate. The fraction forest can be 35 % with 100 km grid spacing and with 10 km grid spacing. The difference in climate between cycle 1 and 2 is an effect of the difference in vegetation, don't you think?

L148-149. This sentence says: "the increase in resolution can be explained by the better representation of the circulation processes". Is this what you mean? What does it mean? consider rephrasing.

L149: "horizontal resolution" Of what?

L149: "1° to 18 km" This is a not so pleasant mix of units. Since you say "approximately" and since your grid spacing is not exactly 1 I think its fine to say "approximately 100 to

18 km".

L155: I would say that no important differences in the land surface variables are seen after cycle 2. If you see differences between cycle 2 and 3, please describe them.

L155: What do you mean with "especially" here? Especially large differences between 1 and 2, or especially small thereafter?

L159-170: Why do you neglect to discuss temperature? Temperature is an important climate variable that responds to changes in vegetation and a variable that vegetation is limited by. You can't discuss equilibria and vegetation climate interactions without discussing temperature.

L164: "In response to the progressive changes in precipitation" Do you mean that vegetation is only sensitive to precipitation changes?

L164-170: Line 164 says that vegetation responds to changes in precipitation. Line 168 says that precipitation responds to changes in vegetation. What's your idea of how climate and vegetation interactions work? You mention temperature as a driver of vegetation on line 181. I don't think you explain it well enough.

L168-169: The correlation is not that good. Look at the Iberian Peninsula, France, the Balkans, Greece. There are lots of regions with increased precipitation and reduced green fraction. Remember that your explanation of precipitation changes is not vegetation but changes in the large scale circulation. This is not affected by vegetation. There is little support that vegetation changes drive large scale changes in mean precipitation (e.g. Belusic et al., 2019; Strandberg & Kjellström, 2019; Davin et al. 2019)

L170: Internal variability of what? I guess you mean in the climate itself. Otherwise you should add it to the list of possible explanations.

L173: This is not a correct naming of this section as it also deals with atmospheric conditions, comparisons between LGM and PD, description of LGM climate and some discussion. Consider reordering this section and to divided into more sections.

L174: What's your definition of tree cover? Is it the same as green fraction?

L180-182: It is true, of course, that LGM vegetation is explained by climatic conditions and $CO_2$ levels. But it doesn't explain why LGM vegetation was different than PD vegetation, because PD vegetation is highly anthropogenic. I don't think its correct to talk about changes here.

L183: Add a reference to Fig 3b after "reconstructions".

L188-189. This sentence is not very precise. It seems like all areas with few reconstructions show tundra and grassland, but actually you are only talking about the Carpathian Basin.

L190: Temperature and precipitation are not land surface conditions. See also comment on line 173.

L199: "few locations" Which locations?

L200: "in line with similar findings" It goes without saying that your results are in line with other results that are similar. Are there also other results? Results that are not in line with yours? All the mentioned studies are made with GCMs. Wouldn't it be appropriate to compare also to S11 which uses a similar setting as yours?

L201: Don't forget that a lot of the shortcomings come from the driving GCM.

L203-217: This is a discussion, not results. Consider moving to "Discussion". It is also a highly confusing paragraph as it in the same time discusses model-proxy disagreement (line 204, line 216), climate anomalies (line 209) and LGM-PD (line 210, line 212). This needs to be straightened up.

L205: Do you have a reference for the model-proxy disagreement in the Iberian Peninsula?

L209: "climate anomalies" What anomalies LGM-PD?

L210-217: Changes in storm tracks could explain the increased precipitation in LGM in southern Europe, but it can't explain the model-proxy disagreement that this paragraph started with. Another important part of the puzzle is the circulation in the GCM. What do the circulation patterns in CCSM look like? Storm tracks should be easy enough to calculate, or at least a map of mslp. You don't offer any descriptions of the climate in the driving GCM. Another reason for the different precipitation patterns in LGM is reduced evaporation from the cold and largely ice covered Atlantic (Strandberg et al., 2011).

L232: This section is not entirely about atmospheric sensitivity to land cover. Consider restructuring.

L235: What do you mean by "again" here. Consider deleting.

L236: The atmospheric response mentioned here is not response to changes in the surface, but rather the models response to different forcing (GHGs, orbital forcing, orography...).

L236: Comparing LGM and PD is not a way to estimate the atmospheric sensitivity to land cover.

L238-248: It starts with a "precipitation decrease" on line 238. This is illustrated by a "temperature response" on line 241. It then goes back to a "decrease of precipitation" on line 243. Please discuss on variable at the time. This is really hard to follow as it is now.

L240-248: It is not clear if this paragraph is only about southern Europe. Please, be more precise with what regions you are discussing.

L240: "atmospheric response" To what? Vegetation, GHGs, orbital forcing, orography...?

L243: "decrease of precipitation" Between what? PD-LGM? LGM(LGM) - LGM(PD)?

L245: "winter wetter conditions". On line 248 you mention a "general dryness in winter". Which is it? Do you discuss different regions?

L248: Do you see a shift in storm tracks in you models? If not it could hardly explain the precipitation patterns. It is not enough to just reference other studies.

L249: "atmopsheric response to the LGM(LGM) with respect to the PD(PD)". What does this mean? I don't understand.

L254: "reduced by 43 % in DJF and enhanced by about 35 % in JJA". I have difficulties to see this in Table 3. First of all the numbers in Table 3 are given in mm/day so its difficult to know the percentages. Second, for LGM(LGM) - LGM(PD), which I guess this is about, precipitation is reduce for both DJF and JJA. I don't understand how JJA could see enhanced precipitation.

L256-259: What is the significance of your results? Is it enough to make conclusions on? Precipitation changes need to be quite large to be significant.

L259-264: This is a discussion, where are the results?

L259-264: Again, I would recommend you to take a look at S11 and see what is said there. In general, you shouldn't have to speculate about how vegetation interacts with climate. There are plenty of papers to read about that. Furthermore, you have your own simulations. Why don't do a proper study of how for example albedo and heat fluxes change in your simulations? If you want to have an example of how that could be done in a palaeo context you could e.g. look at Strandberg et al. (2014). For a more general analysis I can recommend Davin et al. (2019).

L262: "variability in land cover" Do you actually mean difference in land cover?

L266: There are several better references to this than AR5, e.g. Strandberg & Kjellström, 2019, Davin et al. 2019, Jia et al., 2020.

L280: Why do you expect the coupling to be particularly strong here?

L282-285: How would you explain these results? If you can't explain it with physical effects it might as well be random.

L293: Is "parkland" the right way to describe LGM vegetation? Parkland seems highly anthropogenic.

L296-298: I'm not sure if "illustrates" is the right word here. Shows?

L297: "may be related to fluctuations in circulation patterns". In the model one might add.

L302-305: How do you know this? You don't show it.

L304: "water fluxes" I guess you mean heat fluxes.

L303-305: I don't understand this sentence. "LGM land cover led to /.../ when influenced by reduced vegetation fraction". So, the land cover is influenced by the vegetation fraction? Consider rephrasing.

L304: Be careful with the use of parenthesis around "JJA". It don't play well with the other parentheses in this sentence.

Fig 2: You seem to use "land use" and "land cover" interchangeably. Choose one and stick to it. I think land cover is the proper one since there were not much land use during the LGM. Land use is an anthropogenic thing. Define "green fraction". "Green vegetation cover" is not an explanation, just another way to say it.

Fig 4: The colour scale in a) and b) is not good. It's practically impossible to distinguish between colours in the range -24 - -4, and when I see a colour in the map I don't know where to place it in the colour scale. Furthermore, it's very difficult to see the dots in the maps. Find another way to plot them, perhaps with white circles. It's also difficult to see the green and red rings. Think about if there is another way to plot significance.

* Technical comments

L139: "climatological" -> "climatology"

L144: There is no reference to Fig 2 prior to this reference to Fig 3. Consider reordering the figures.

L179 "Fig. 4a and b" -> "Fig. 3a and b"

* References

Belušić, D., Fuentes-Franco, R., Strandberg, G.and Jukimenko, A., 2019: Afforestation reduces cyclone intensity and precipitation extremes over Europe. Environ. Res. Lett. 14,https://doi.org/10.1088/1748-9326/ab23b2 Davin, E. L., Rechid, D., Breil, M., Cardoso, R. M., Coppola, E., Hoffmann, P., Jach, L. L., Katragkou, E., de Noblet-Ducoudré, N., Radtke, K., Raffa, M., Soares, P. M. M., Sofiadis, G., Strada, S., Strandberg, G., Tölle, M. H., Warrach-Sagi, K., and Wulfmeyer, V.: Biogeophysical impacts of forestation in Europe: first results from the LUCAS (Land Use and Climate Across Scales) regional climate model intercomparison, Earth Syst. Dynam., 11, 183–200, https://doi.org/10.5194/esd-11-183-2020, 2020.

Demory, M.-E., Berthou, S., Sørland, S. L., Roberts, M. J., Beyerle, U., Seddon, J., Haarsma, R., Schär, C., Christensen, O. B., Fealy, R., Fernandez, J., Nikulin, G., Peano, D., Putrasahan, D., Roberts, C. D., Steger, C., Teichmann, C., and Vautard, R.: Can high-resolution GCMs reach the level of information provided by 12–50 km CORDEX RCMs in terms of daily precipitation distribution?, Geosci. Model Dev. Discuss., https://doi.org/10.5194/gmd-2019-370, in review, 2020.

Di Luca, A., de Elía, R. and Laprise, R.: Potential for added value in precipitation simulated by high-resolution nested Regional Climate Models and observations, Clim. Dyn. 38, 1229–1247, https://doi.org/10.1007/s00382-011-1068-3, 2011.

Iles, C. E., Vautard, R., Strachan, J., Joussaume, S., Eggen, B. R., and Hewitt, C. D.: The benefits of increasing resolution in global and regional climate simulations for European climate extremes, Geoscientific Model Development Discussion,

https://doi.org/10.5194/gmd-2019-253, 2019.

Jia, G., E. Shevliakova, P. Artaxo, N. De Noblet-Ducoudré, R. Houghton, J. House, K. Kitajima, C. Lennard, A. Popp, A. Sirin, R. Sukumar, L. Verchot, 2019: Land–climate interactions. In: Climate Change and Land: an IPCC special report on climate change, desertification, land degradation, sustainable land management, food security, and greenhouse gas fluxes in terrestrial ecosystems [P.R. Shukla, J. Skea, E. Calvo Buendia, V. Masson-Delmotte, H.-O. Pörtner, D.C. Roberts, P. Zhai, R. Slade, S. Connors, R. van Diemen, M. Ferrat, E. Haughey, S. Luz, S. Neogi, M. Pathak, J. Petzold, J. Portugal Pereira, P. Vyas, E. Huntley, K. Kissick, M, Belkacemi, J. Malley, (eds.)]. In press.

Kjellström, E., Brandefelt, J., Näslund, J.-O., Smith, B., Strandberg, G., Voelker, A. H. L. & Wohlfarth, B. 2010: Simulated climate conditions in Europe during the Marine Isotope Stage 3 stadial. Boreas, 10.1111/j.1502-3885.2010.00143.x. ISSN 0300-9483.

Prein, A. F., Holland, G. J., Rasmussen, R. M., Done, J., Ikeda, K., Clark, M. P. and Liu, C. H.: Importance of Regional Climate Model Grid Spacing for the Simulation of Heavy Precipitation in the Colorado Headwaters. J. Climate, 26: 4848–4857, doi: 10.1175/JCLI-D-12-00727.1, 2013.

Rauscher, S.A., Coppola, E., Piani and Giorgi F.: Resolution effects on regional climate model simulations of seasonal precipitation over Europe. Clim. Dyn. 35, 685–711, https://doi.org/10.1007/s00382-009-0607-7, 2010.

Strandberg, G., Brandefelt, J., Kjellström, E. and Smith, B. 2011: High-resolution regional simulation of last glacial maximum climate over Europe. Tellus 63A, 107-125.DOI: 10.1111/j.1600-0870.2010.00485.x

Strandberg, G., Kjellström, E., Poska, A., Wagner, S., Gaillard, M.-J., Trondman, A.-K., Mauri, A., Davis, B. A. S., Kaplan, J. O., Birks, H. J. B., Bjune, A. E., Fyfe, R., Giesecke, T., Kalnina, L., Kangur, M., van der Knaap, W. O., Kokfelt, U., Kuneš, P., Latalowa, M., Marquer, L., Mazier, F, Nielsen, A. B., Smith, B., Seppä, H., and Sugita,

S.: Regional climate model simulations for Europe at 6 and 0.2 k BP: sensitivity to changes in anthropogenic deforestation, Clim. Past, 10, 661-680, doi:10.5194/cp-10-661-2014, 2014.

---

## Referee Comment (RC2) · Anonymous Referee #2 · 14 Jan 2021

The manuscript presents new simulations aiming at highlighting the role of vegetation cover in climate change in Europe during the LGM. I found this paper particularly interesting and well written. Nevertheless, I have some comments concerning the model-data comparison section.

Regarding the results presented in Fig. 3, the statement "The LGMLGM climate agrees with the pollen-based paleoclimate reconstructions at most sites" (l.198) is not very convincing. I will not say that only "few locations show considerable differences" (l.199) since 5 out of 14 sites show temperatures reconstructed in July significantly different from the simulations and since 6 out of 14 sites show precipitation in July significantly different from the simulated ones. The regional character of the January precipitation in Southern Europe during the LGM compared to PD (higher LGM precipitation) is not

supported by the data, as noted by the authors. They discuss this point but the cited works are not well cited or at least the text as written is misleading for the reader. Roucoux et al. (2005) effectively suggest that the LGM is not the driest and coldest interval of the last ice age, but wetter than the periods before and after it. However, in Roucoux et al. (2005) these colder and dryer periods are the Heinrich events and not the recent period. The Estanya lake record in the NE Iberia (Morellon et al. 2009) is also cited as showing wetter LGM conditions. This is ok but unlike the simulations, these lake data show that the LGM is wetter than the H1 in the NE of the Iberian Peninsula but much drier than the Holocene and in particular the final Holocene. The same applies to the modelling work of Ludwig et al, 2018, showing that the LGM is wetter than H1 but drier than the pre-industrial period. Citing all these works for justifying that other data or modelling experiments show wetter conditions during the LGM but avoiding to say "wetter than what" is misleading for the reader. In any case, they cannot be used as a justification to explain that the simulations show wetter winter conditions at the LGM than at the PD.

A comparison with a larger number of sites would be beneficial for the evaluation of the simulations. It would be good to add sites whose reconstructions are available in the literature and not only those from a compilation made more than 14 years ago. A strong added value to the paper would be to estimate the temperature and precipitation (with a MAT or another method) over the 71 sites used in Figure 6. Doing a model-data comparison on the basis of 71 sites instead of the 14 currently used would bring more robustness to the validation of the simulations by the data. Nevertheless, I would understand that it is a too much work for this paper.

The authors chose to compare simulated tree cover % to the available arboreal PFT % from pollen records to evaluate the model simulations. However, it would be great to take into account in the discussion that arboreal PFT % "is a relative rather than absolute metric of landscape openness" as stated by Davis et al. 2015.

p. 6, l. 179: "Fig. 3" instead of "Fig. 4".

---

## Author Comment (AC1) · 23 Feb 2021

**Final Response to Referee #2**

We appreciate the time the reviewer has invested to read the manuscript in such a careful and thorough manner. The comments have been carefully considered and responded. Please find below our response to each comment.

*1.      Regarding the results presented in Fig. 3, the statement "The LGMLGM climate agrees with the pollen-based paleoclimate reconstructions at most sites" (l.198) is not very convincing. I will not say that only "few locations show considerable differences" (l.199) since 5 out of 14 sites show temperatures reconstructed in July significantly different from the simulations and since 6 out of 14 sites show precipitation in July significantly different from the simulated ones.*

**RESPONSE**:

We thank you for this comment. We agree that the statement in the manuscript is not yet convincing. We have instead considered not only the site as sample, but also the two variables and the two months (14 x 2 x 2 samples). This means that 14 sites offer 56 samples, from which 15 samples do not agree with modelled climate (5 temperature and 6 precipitation samples in July, and 4 precipitation samples in January). To clarify this, we will explain it better and reformulate these sentences in the revised manuscript taking into account that some samples show significant differences, especially in July.

*2.      The regional character of the January precipitation in Southern Europe during the LGM compared to PD (higher LGM precipitation) is not supported by the data, as noted by the authors. They discuss this point but the cited works are not well cited or at least the text as written is misleading for the reader. Roucoux et al. (2005) effectively suggest that the LGM is not the driest and coldest interval of the last ice age, but wetter than the periods before and after it. However, in Roucoux et al. (2005) these colder and dryer periods are the Heinrich events and not the recent period. The Estanya lake record in the NE Iberia (Morellon et al. 2009) is also cited as showing wetter LGM conditions. This is ok but unlike the simulations, these lake data show that the LGM is wetter than the H1 in the NE of the Iberian Peninsula but much drier than the Holocene and in particular the final Holocene. The same applies to the modelling work of Ludwig et al, 2018, showing that the LGM is wetter than H1 but drier than the pre-industrial period. Citing all these works for justifying that other data or modelling experiments show*

*wetter conditions during the LGM but avoiding to say "wetter than what" is misleading for the reader. In any case, they cannot be used as a justification to explain that the simulations show wetter winter conditions at the LGM than at the PD. A comparison with a larger number of sites would be beneficial for the evaluation of the simulations.*

**RESPONSE**:

We thank the reviewer for pointing out that the discussion of the literature is misleading for the reader. We would like to mention that we used these publications to highlight the uncertainties related with past climates. To avoid any further misleading, we will certainly double-check the literature and reformulate this paragraph in the revised manuscript as follows:

"For example, there is large model-proxy disagreement in January precipitation over the Iberian Peninsula. Based on evidence for the presence of certain tree species in the northwestern part of the Iberian Peninsula, Roucoux et al. (2005) suggested that the LGM was not necessarily the period of the most severe, i.e., cold and dry, climatic conditions everywhere. Roucoux et al. (2005) and Ludwig et al., (2018) also suggested that this same region during LGM sensu strictu was warmer and wetter than the end of Marine Isotope Stage 3 (MIS3, ca. 23 ka; Voelker, A. H. L. et al., 1997; Kreveld, S. et al., 2000) and the start of the Heinrich even 1 (H1, ca. 19 ka; Sanchez Goñi and Harrison, 2010; Álvarez-Solas et al., 2011; Stanford et al., 2011). This could be a hint that model-proxy comparison fails because the proxies refer to $21 \pm 2$ ka (Wu et al., 2007), i.e., either the end of MIS3 or beginning of H1. Compared to the pre-industrial period, Beghin et al. (2016) found evidence that the interior and northwestern Iberian Peninsula presented wetter conditions during the LGM, which can be explained by a southward shift in the North Atlantic storm track during LGM compared to present day as suggested by many studies (e.g.; Hofer et al., 2012a; Luetscher et al., 2015; Merz et al., 2015; Ludwig et al., 2016; Wang et al., 2018; Raible et al., 2020). "

***3.     It would be good to add sites whose reconstructions are available in the literature and not only those from a compilation made more than 14 years ago.***

**RESPONSE**:

We appreciate this suggestion. We also agree that adding more sites would be beneficial for the evaluation. At the moment, we are not aware of a more recent publication on terrestrial temperature and precipitation reconstructions for the LGM. Nevertheless, we will search for other studies to be included in the revised manuscript.

*4.      A strong added value to the paper would be to estimate the temperature and precipitation (with a MAT or another method) over the 71 sites used in Figure 6. Doing a modeldata comparison on the basis of 71 sites instead of the 14 currently used would bring more robustness to the validation of the simulations by the data. Nevertheless, I would understand that it is a too much work for this paper.*

**RESPONSE**:

We would like to mention that we do carry out a model-data comparison using tree cover. However, we agree that further model-data comparisons using additional reconstructed information from these 71 sites would certainly add more value. This would surely be an effort that is beyond the scope of this study. We will therefore consider this comment in the outlook of the revised manuscript.

*5.      The authors chose to compare simulated tree cover % to the available arboreal PFT % from pollen records to evaluate the model simulations. However, it would be great to take into account in the discussion that arboreal PFT % "is a relative rather than absolute metric of landscape openness" as stated by Davis et al. 2015. p. 6,*

**RESPONSE**:

We agree that It is currently not possible with any method to make reliable quantitative reconstructions of tree cover using LGM pollen assemblages. We clarify this in the revised version as part of the discussion.

*6.      l. 179: "Fig. 3" instead of "Fig. 4".*

**RESPONSE**:

We thank the reviewer for pointing out the mismatch in the figure references. We have changed this as suggested.

Once again, we would like to thank the referee for reviewing our manuscript so carefully and we are looking forward to meeting his/her expectations.

Best regards,

Patricio Velasquez (on behalf of the author team)

**References**

Álvarez-Solas, J., Montoya, M., Ritz, C., Ramstein, G., Charbit, S., Dumas, C., Nisancioglu, K., Dokken, T., Ganopolski, A., 2011. Heinrich event 1: an example of dynamical icesheet reaction to oceanic changes. Clim. Past 7, 1297–1306. https://doi.org/10.5194/cp-7-1297-2011.

Kreveld, S. van, Sarnthein, M., Erlenkeuser, H., Grootes, P., Jung, S., Nadeau, M. J., Pflaumann, U. and Voelker, A.: Potential links between surging ice sheets, circulation changes, and the Dansgaard-Oeschger Cycles in the Irminger Sea, 60–18 Kyr, Paleoceanography, 15(4), 425–442, https://doi.org/10.1029/1999PA000464, 2000.

Sanchez Goñi, M.F., Harrison, S.P., 2010. Millennial-scale climate variability and vegetation changes during the last glacial: concepts and terminology. Quat. Sci. Rev. 29, 2823–2827. https://doi.org/10.1016/j.quascirev.2009.11.014.

Stanford, J. D., Rohling, E. J., Bacon, S., Roberts, A. P., Grousset, F. E. and Bolshaw, M.: A new concept for the paleoceanographic evolution of Heinrich event 1 in the North Atlantic, Quaternary Science Reviews, 30(9), 1047–1066, https://doi.org/10.1016/j.quascirev.2011.02.003, 2011.

Sugita, S. (2007). Theory of quantitative reconstruction of vegetation I: pollen from large sites REVEALS regional vegetation composition. *The Holocene, 17*(2), 229-241. doi:10.1177/0959683607075837.

Voelker, A. H. L., Sarnthein, M., Grootes, P. M., Erlenkeuser, H., Laj, C., Mazaud, A., Nadeau, M.-J. and Schleicher, M.: Correlation of Marine 14C Ages from the Nordic Seas with the GISP2 Isotope Record: Implications for 14C Calibration Beyond 25 ka BP, Radiocarbon, 40(1), 517–534, https://doi.org/10.1017/S0033822200018397, 1997.

---

## Author Comment (AC2) · 23 Feb 2021

**Final Response to Referee #1**

We thank the reviewer for the careful and thorough reading of our manuscript. The comments have been carefully considered and responded. Please find below our response to each comment.

**General comment:**

*1.      Velasquez et al simulates LGM climate with a chain of asynchronous coupled models (GCM -> RCM -> DVM -> RCM etc). This is exactly what is needed in palaeo climate. High resolution enables better comparison with proxy data, and 18 km grid spacing is impressive in a palaeo context. A good description of vegetation is needed to get LGM conditions as realistic as possible. The authors make sure to have a vegetation compliant with the climate by making as much as 7 iterations with climate and vegetation models. This could be interesting for the readers of Climate of the past, but I think the paper need a lot of improvement before that. That the model set up is relevant is what makes me recommend major revision.*

*My major concerns are with:*
*i)      the lack of discussion of the role of vegetation on LGM climate,*
*ii)     the lack of context and comparison with previous studies,*
*iii)    how the authors describe results, but don't explain them or try to understand them*
*iv)     the structure of the text where results from different sensitivity studies are mixed and where results and discussions are mixed.*

*Detailed comments follow below. As always, I might have misunderstood some things, and my comments could be invalid because of that. If such misunderstandings do occur, think about if your paper is written in a clear enough way.*

**RESPONSE**:

We thank you for your detailed comments which will certainly improve the manuscript. We will take care of the 4 major concerns listed above in the following responses and in the revised manuscript.

**Major comments:**

*1.      The title of the paper is "The role of land cover on the climate of glacial Europe", but I don't think I get any new insights by reading it.  Despite the ambitious model set up. It seems you did some sensitivity runs, but don't know what to do with them, and that you didn't study the literature on vegetation climate interactions. It's not enough to just say that LGM was cold and that vegetation affects climate, we know that already. Why does it? If you want to advance climate modelling you need to give physical reasons for your results.  Otherwise your results could just be a random effect caused by different initial conditions. Try to explain your results. Look at variables that might be relevant. Albedo and heat fluxes are not analysed at all.*

**RESPONSE**:

We thank you for this comment. We realise that there are some shortcomings in the description and interpretation of the results. In our revised manuscript, we will provide more reasoning and extended analysis of the climatological variables.

*2.      The results of the RCM simulations are highly depending on the driving GCM. But you don't discuss your GCM runs at all.  What is LGM vegetation like in the GCM? This explains many of the differences between cycle 1 and 2.  What is LGM climate like in the GCM compared to other PMIP3 runs, colder, warmer, wetter, drier?  What is the general circulation like? Could that explain the precipitation patterns?*

**RESPONSE**:

We agree that the discussion of the GCM is too short. Thus, in the revised manuscript, we will describe in more detail how land cover is included in the GCM simulation; additionally, we will further discuss the GCM and WRF land cover when analysing our results. We also add a brief comparison with other PMIP simulations although there are several papers which specifically focused on atmospheric dynamics, so a detailed repetition of these results is beyond the scope of this paper. We will also include the following passage in the revised manuscript:

"In this study, two global climate simulations, one for present-day conditions (1990 CE conditions) and one for the LGM, were dynamically downscaled. These global simulations were performed with the atmosphere and land component of the Community Climate System Model (version 4; CCSM4; Gent et al., 2011). A horizontal resolution of 1.25° × 0.9° (longitude × latitude)

was used in both components and 26 vertical hybrid sigma-pressure levels in the atmosphere (CAM4, Neale et al., 2010) and 15 soil layers in the land component (CLM4, Oleson et al., 2010), respectively. CCSM4 was coupled to so-called data models for the ocean and sea ice. These surface boundary conditions were obtained from a fully coupled simulation with CCSM3, but at lower resolution (see details in Hofer et al. 2012a). CCSM3 provided monthly mean time varying sea ice cover and sea surface temperatures (SSTs). Furthermore, the Community Ice Code, version 4 (CICE4; Hunke and Lipscomb 2008) was set to its thermodynamic-only mode, so that sea ice cover was prescribed, but surface fluxes through the ice were computed by considering snow depth, albedo, and surface temperature (as simulated by the atmospheric component CAM4, see Merz et al. 2015). Further details of this simulation were presented in Hofer et al. (2012 a; b) and Merz et al. (2015).

For each simulation 33 years were run, but only the last 30 years and 2 months were used in this study. Present-day boundary conditions were set to 1990 CE values, whereas the LGM boundary conditions were modified as follows: lower concentrations of greenhouse gases ($CO_2$ = 185 ppm; $N_2O$ = 200 ppb; $CH_4$ = 350 ppb), changed Earth's orbital parameters (Berger 1978), the addition of major continental ice sheets (Peltier, 2004) and associated sea level changes (120 m lower than today; Clark et al., 2009). Note that in these LGM simulations, land cover was set to pre-industrial conditions. The simulations further provided 6-hourly data, which is necessary to drive regional climate models.

These present-day and LGM CCSM simulations have been analysed in a variety of studies, including additional simulations for other glacial and interglacial states (Hofer et al. 2012 a;b; Merz et al. 2013;2014a;b; 2015; 2016; Landais et al. 2016). The focus of these studies was in particular on the model's ability to simulate LGM climate and atmospheric circulation changes during glacial times. Hofer et al. (2012a) showed that the model performs reasonably well under present-day conditions, showing a cold bias in the global mean temperature of 0.3 °C. The reason for this bias is the rather coarse resolution of the ocean, which led to an underestimation of the northward heat transport in the North Atlantic and an overestimation in the horizonal extension of sea ice cover (Hofer et al. 2012a). The here presented LGM climate simulation agree with PMIP models (Braconnot et al., 2007) showing a global mean temperature response between LGM and preindustrial conditions of 5.6 °C. However, the temperature response over Europe shows a better agreement with proxy data (Wu et al. 2007) than the multi-model mean response in Braconnot et al. (2007). The global mean precipitation response of the LGM simulation used in this study is again similar to the multi-model mean response of Braconnot et al. (2007), although the regional pattern and seasonal behaviour show some deviations to proxy data over Europe (Wu et al. 2007, Hofer et al. 2012a). The LGM model simulation further reveals a clear southward shift and a more zonal orientation of the storm track over the North Atlantic compared to

present-day conditions (Hofer et al., 2012a). This shift, and substantial changes in the weather patterns (Hofer et al. 2012b), explain the precipitation anomalies found over the Iberian Peninsula and the western part of the Mediterranean. The reason for these shifts could be traced back to the height of the Laurentide ice sheet and the effect of it on the eddy driven jet over the North Atlantic and the stationary and transient waves, as sensitivity simulations suggested (Merz et al. 2015). Such a shift is also reported in several other modelling studies (see the review of Raible et al. 2020).

Overall, the CCSM4 simulations of LGM climate were state-of-the-art in 2012 and they are still today as their horizontal resolution is similar to models used in phase 4 of the Paleoclimate Model Intercomparion Project (PMIP4, Kagayama et al. 2017, 2020)."

***3. Temperature differences between the different cycles are hardly discussed. Temperature decreases with almost 0.5◦C between cycles 1 and 2. This could perhaps be explained by a large albedo increase when the forest disappears (albedo is not shown). But temperature increases with 0.5◦C between cycles 3 and 4 despite no significant vegetation changes. This is unexplained. Could it be that the differences are just a result of natural variability and not a result of vegetation at all?***

**RESPONSE**:

We thank the reviewer for this comment. We agree that this part needs clarification. Therefore, we will add a more detailed analysis of temperature in the revised manuscript.

***4. A related issue to the above is the question of significance. The significance of the results is not tested here. Even though vegetation has a clear effect on climate the effect is small compared to the effect of different forcing, and sometimes also compared to natural variability. Therefore, it is important to check if the results are significant.***

**RESPONSE**:

We agree that the significance of results needs to be assessed. We will therefore include the analysis of the significance in the revised manuscript.

*5.      I feel that the paper misses to discuss some relevant previous studies, particularly Strandberg et al. (2011), henceforth S11. S11 studies LGM with more or less the same method as you do. The present model set up is an improvement from S11 so you shouldn't be afraid to discuss it. Why not get inspired by how S11 discuss the role of the GCM or the uncertainty in proxy data. Vegetation climate interactions is not the main number in S11, but there is a section about that too. Kjellström et al. (2010) used the same approach for another cold climate, MIS3.*

**RESPONSE**:

We thank the reviewer for suggesting this relevant literature. Although we mentioned S11, we realize that its discussion was too short. Therefore, we include a more comprehensive discussion of the exiting literature and similarities of our approach to existing ones in the revised manuscript.

*6.      The structure of the paper makes it a bit difficult to follow.   For example, section 4 "Comparison of the simulated land surface conditions to proxy reconstructions" deals in large parts also with vegetation differences between PD and LGM, difference in climate between PD and LGM, difference in climate between models and proxies and a discussion about simulated LGM climate in other studies.  In a similar way section 5 "Atmospheric sensitivity to land cover" deals largely with differences between LGM and PD climate. I think it would be good if you could discuss one thing at the time.*

**RESPONSE**:

We have reorganized the section 4 and 5 into the following sections:

4. Comparison and discussion of modelled and reconstructed climate
5. Comparison and discussion of modelled and reconstructed land cover
6. Influence of land cover and external forcing on climate

This also responds to comments m38 and m51 (related to line L173 and L232, respectively).

**Minor comments**

*m1)    L10-14: I think these conclusions are too general given the title of the paper.  Please quantify a bit and perhaps also explain why you get an effect.*

RESPONSE:

Please see response to comment m4 (related to line L13).

*m2)    L11-12: "colder and drier", "warmer and drier".  Is this LGM(LGM) - LGM(PD) or LGM(PD) - LGM(LGM)?*

RESPONSE:

It is LGM$_{LGM}$ -  LGM$_{PD}$. See response to comment m4 (related to line L13).

*m3)    L13: "southward displacement". This sentence reads to me like "Even with a southward displacement of the storm track regional climate is influenced by land cover".  Is this what you mean?  Why would a southward displacement counteract or diminish the influence of vegetation? Please rephrase.*

RESPONSE:

The part is too complicated for the abstract and is removed. See response to comment m4 (related to line L13).

*m4)    L13: "increased importance of the Atlantic".  What do you mean?  In what way is the Atlantic more important?  And why?  I can't find anything about the importance of the Atlantic in the rest of the text.*

RESPONSE:

Answer to the first 4 minor comments: We rephrased the 4 lines of the abstract:

"To assess the importance of land cover on the LGM climate of Europe, we performed a sensitivity simulation where we used LGM climate but present-day (PD) land cover. Using LGM climate and land cover leads to colder and drier summer conditions around the Alps and warmer and drier climate in southeastern Europe compared to LGM climate determined by PD land cover. This demonstrates that LGM cover plays an important role in regulating regional climate. Therefore, realistic glacial land cover estimates are needed to accurately simulate regional glacial climate states in areas with interplays between complex topography, large ice sheets and diverse land cover, as observed in Europe."

*m5) L25: "Recent advances" It could be discussed how recent it is since S11 is 9 years old. See discussion above.*

**RESPONSE**:

We rephrase the sentence to "Advances in regional climate models have led to the application of such models to the glacial climate of Europe on a high spatial resolution (e.g.; Kjellström et al. 2010, Strandberg et al. 2011, Ludwig et al., 2017; 2020).".

*m6) L45: There are better references to this than AR5, e.g. Rauscher et al., 2010; Di Lucaet al., 2011; Prein et al., 2013; Iles et al., 2019 and Demory et al. 2020.*

**RESPONSE**:

We replace the AR5 reference with the ones mentioned and add also:

Gomez-Navarro, J. J., O. Boethe, S. Wagner, E. Zorita, J. P. Werner, J. Luterbacher, C.C. Raible, and J.P. Montavez, 2015: A regional climate palaeo simulation for Europe in the period 1501-1990. Part II: comparison with gridded reconstructions. *Climate of the Past*, **11**, 1077-1095
Gómez-Navarro, J. J., Montávez, J. P., Jerez, S., Jiménez-Guerrero,P., Lorente-Plazas, R., González-Rouco, J. F., and Zorita, E.: A regional climate simulation over the Iberian Peninsula for the last millennium, Clim. Past, 7, 451–472, doi:10.5194/cp-7-451-2011,2011.

Gómez-Navarro, J. J., Montávez, J. P., Jiménez-Guerrero, P., Jerez,S., Lorente-Plazas, R., González-Rouco, J. F., and Zorita, E.: Internal and external variability in regional simulations of the Iberian Peninsula climate over the last millennium, Clim. Past,8, 25–36, doi:10.5194/cp-8-25-2012, 2012.

Gómez-Navarro, J. J., Montávez, J. P., Wagner, S., and Zorita, E.:A regional climate palaeo simulation for Europe in the period1500–1990 – Part 1: Model validation, Clim. Past, 9, 1667–1682,doi:10.5194/cp-9-1667-2013, 2013.

*m7)   L52:  Is Tao et a.  (2013) the appropriate reference here?  It's about the effect of vegetation on air quality in the US. If you want vegetation climate interactions in RCMs in palaeo climate (Consistent with Strandberg et al.  (2011) and Ludwig et al.  (2017)) I would recommend e.g. Kjellström et al. (2010) or Strandberg et al. (2014). If you want a more general reference on vegetation climate interactions you could start with Jia et al. (2019).*

**RESPONSE**:

We agree that the research by Tao et al. is not the optimal reference in the context of paleo studies. We will replace it with the mentioned publications.

*m8)   L61: There are two studies that use a similar approach as you, and in addition in cold climates: Kjellström et al.  (2010) and Strandberg et al.  (2011).  Especially S11 would be worth to note as it simulates LGM.*

**RESPONSE**:

We agree. Therefore, we add a paragraph in the revised manuscript highlighting that the approach is similar to the ones suggested in Kjellström et al.  (2010) and Strandberg et al.  (2011). See also response to the fifth major comments.

*m9)   L63-68: When I read this I understand that you have the following model chain: GCM->RCM->DVM->RCM-> etc. From section 2.4 I understand that the first DVM simulation is forced by the GCM. I suppose that the description in 2.4 is the correct one. Please, check and correct.*

**RESPONSE**:

Yes, this is somewhat misleading. We rephrase it to:

"… The coupled modelling starts with a GCM (CCSM4; Gent et al., 2011) which serves as input to drive a dynamic vegetation model (LPJ-LMfire; Pfeiffer et al., 2013). In a next step, the

atmospheric boundary conditions from the GCM and the output of LPJ-LMfire are passed to an RCM (WRF; Skamarock and Klemp, 2008). The resulting RCM output is in turn used to drive LPJ-LMfire which again returns land cover to the RCM. The RCM simulation is then repeated with the new land cover as boundary condition. …"

*m10) L72-75: Are these 31 years part of a longer simulation, in that case how were they selected? Is the LGM simulation a part of a transient simulation or is it steady state?*

**RESPONSE**:

We realize that the description of the GCM was too short. We include a better description of the GCM in the revised manuscript. Please refer to the response to the second major comment and response to m12 (related to line L87).

*m11) L74: Please add an explanation and a reference to these data models.*

**RESPONSE**:

The data model is a specific expression in the CESM world. The data models (for ocean and sea ice) simply prescribe time-varying sea-surface temperatures over the oceans and sea-ice conditions, which are obtained from a fully coupled simulation. This explanation is now included in the revised manuscript. See response to second major comment.

*m12) L87: 31 years were simulated by the GCM, 30 by the RCM. Why is one year not used? Which 30 years are used?*

**RESPONSE**:

We agree that this was unclear. 33 years in total were run with the GCM in an equilibrium state, whereas the first 3 years are considered as spin-up time. We used the last two months of these 3 years as spin-up of our RCM and the following 30 years are downscaled by the RCM and provide the basis for our analysis. Note that the 30-year simulation with the RCM are divided in two 15-year chunks. While the first 15-year simulation makes use of the 2 months spin-up that coincides with the one of the GCM, the second chunk overlaps with the last two months of the first 15 years RCM simulation. We clarify this in the GCM subsection of the revised manuscript and here:

"The relevant parameterisation schemes chosen to run WRF are described in Velasquez et al. (2020). To perform the regional simulations in this study, we used the so-called adaptive time-step method as described in Skamarock and Klemp (2008), i.e., the integration time step can vary from time to time. For example, the model is stable with a timestep of 160 seconds during most integration steps but it might need a reduction to 60 seconds during convective situations to maintain stability. With a fixed timestep, the entire simulation must be run with 60 seconds to overcome these convective situations, while the adaptive time-step method is able to make use of the larger time step 160 seconds during most of the simulation. The advantage is to substantially save computer resources. Furthermore, each simulation is driven by the 30 years of the corresponding GCM simulation (excluding the 3-year spin-up of the GCM simulation). These 30 years are split up into two single 15-year periods which both preceded by a 2-month spin-up to account for the time required for the land surface to come into equilibrium. We used the last 2 months of the 3-year spin-up of the GCM simulation for the first 15 years. A spin-up of the regional model of 2 months seems to be sufficient as soil moisture show no significant trend after 15 days up to a level of 1 m.  The initial and boundary conditions for WRF were provided by the global CCSM4 simulations, including the Fennoscandian ice sheet and reduced sea levels during the LGM. The other external forcing functions follow the PMIP3 protocol (for more details see: Hofer et al., 2012a; Ludwig et al., 2017). Furthermore, no nudging is applied in the RCM simulations. Finally, the LGM glaciation over the Alpine region is included in the regional climate model using estimates from Seguinot et al. (2018) and additional LGM glaciated areas (e.g., Pyrenees, Carpathians) from Ehlers et al. (2011). Calculation of the LGM land cover is described in Sect. 2.4. These settings are used to produce the main simulation ($LGM_{LGM}$) being the final product of the asynchronous coupling design (described in Sec 2.4).

Additionally, a control simulation under present day conditions is carried out to compare the simulated LGM climate response and land cover response with proxy data. This simulation ($PD_{PD}$) is driven by the GCM simulation with 1990 CE conditions (Hofer et al., 2012a), and uses the default present-day MODIS-based land cover dataset from WRF as the land surface boundary condition (Skamarock and Klemp, 2008).

Finally, a sensitivity simulation is conducted to extract the importance of land cover for the LGM climate in Europe. This simulation (called $LGM_{PD}$) uses the GCM simulation with LGM conditions (Hofer et al., 2012a), but with the default present-day MODIS-based land cover dataset from WRF as for the land surface.

Comparing $LGM_{PD}$ with $PD_{PD}$ illustrates the atmospheric response to changes only in the atmospheric forcing, i.e., without changes in land cover. The comparison of $LGM_{LGM}$ and the

LGM$_{PD}$ allows us to extract the influence of land cover on the atmosphere, i.e., without changes in atmospheric boundary conditions. All WRF simulations are summarised in Table 1."

***m13) L88: "adaptive time-step to increase ... computer facilities" I don't understand this at all. What does it mean?***

**RESPONSE**:

Climate models usually use a fixed time step to integrate the equations, i.e., every 160 seconds in the coarsest domain. However, WRF model allows to integrate the equations with variable time steps depending on the situation, i.e., stability of the current model integration step indicated by the CFL criterion, the so-called adaptive time-step method as described in Skamarock and Klemp (2008). The advantage of this method is to substantially save computer resources as the calculation time can be reduced when using bigger time steps. For example, the model simulation is stable with a timestep of 160 seconds during most integration steps but might need a reduction to 60 seconds during convective situations. With a fixed timestep the whole simulation must be run with 60 seconds to overcome these few convective situations, while the adaptive time-step method is able to make use of the larger time step 160 seconds during most of the simulation. This behaviour of the adaptive time-step can save considerable amounts of computational time.

The details are certainly beyond the scope of this manuscript but if a reader would like to follow our approach it is an important information when setting up WRF. We will clarify it in the revised manuscript.

***m14) L89: If you divide 30 years into two 15 year periods and start every simulation with a two-month spin-up this will give you 2x(14y 10m) = 29 years and 8 months. How do you get 30 years of data from that?***

**RESPONSE**:

We simply start 2 months before the actual 15-year period begins. For instance, we started on November 1st of the 14[th] year when simulating the second 15 years. See also response to comment m12 (related to line L87).

We understand that this is somewhat unclear in the manuscript and have rephrased this sentence to: "The two 15-year simulations are both preceded by a 2-month spin-up to account for the time required for the land surface to come into equilibrium."

***m15)   L90: 15 days seems to me to be a bit short. How do you decide that quasi-equilibrium is reached? In Velasquez et al. (2019) I can only find the following: "Tests show that the WRF land scheme reaches a quasi-equilibrium after approximately 15 d." That doesn't explain much.  I guess that spin-up time also depends on the season.  When do you start your simulation?***

RESPONSE:

We performed several tests on this issue. Therefore, we assessed the soil moisture in different levels. A notable trend is observed in the four layers within the first 10 days, which initially suggests that the spin-up period could be set at around 15 days. To ensure that the regional model is in quasi-equilibrium, we defined a longer spin-up that covers 61 days. This information will be included in the revised manuscript.

***m16)   L93:  What do you mean with "perpetual" here?  Do you mean steady-state with constant forcing?***

RESPONSE:

Yes, we mean with constant LGM conditions. Note that "perpetual" is often used in this context but we decided to remove this word in the revised manuscript.

***m17)   L93: "Reduced sea level and increased ice sheets".  This is somewhat ambiguous.  I guess you mean that sea level was lower and ice sheets were larger than today. It could also mean that LGM conditions have been revised in the PMIP3 protocol compared to previous protocols. It's not entirely clear.***

RESPONSE:

We will reformulate and move this sentence to another position of the paragraph.

*m18)   L115 Are the "reconstructed CO2 concentrations" used in the RCM the same as used in the GCM (PMIP3 forcing)?  It seems like an unnecessary complicated way to say that forcing is the same as in the GCM. If it's not the same, why not?*

**RESPONSE**:

They are the same, so we decided to reformulate the sentence to "In each of our simulations, we drove LPJ-LMfire for 1020 years with the climate and forcing (greenhouse gases: $CO_2$, $N_2O$ and $CH_4$) from the GCM and RCM, and present-day soil physical properties extrapolated out on to the continental shelves (Kaplan et al., 2016)."

*m19)   L120-122:  What vegetation field is used in the first GCM run?  See also comment onL63-68.*

**RESPONSE**:

As mentioned in Hofer et al. (2012) the vegetation and the soil types in the LGM simulation with the GCM are prescribed to the preindustrial distribution except for the additional land areas and the regions that are covered by ice sheets. In the additional land cells, vegetation and soil types are set to the mean values of nearby cells and in the ice covered regions the model's standard value for such conditions are used.  We extend our model description accordingly in the revised manuscript. See the response to the second major comment.

*m20)   L133:  I would say it's more correct to call this section "Results of the iterative...".  You don't have other simulations to compare with so you can't estimate the effect of the coupling. You could, however, describe the results of your simulations, and that's what you do.*

**RESPONSE**:

We change to "Results of the iterative asynchronous coupling" in the revised manuscript.

*m21)   L135: How do you determine that quasi-equilibrium is reached?  Just by eyeballing or do you have a criterion for equilibrium. It seems like you just decide that equilibrium is reached, but how can you be sure without a proper metric?*

**RESPONSE**:

It is always difficult to give a quantitative measure when only having 7 iterations, so the first selection was done by eyeballing, i.e., a rather stable mean and no coherent structure when plotting differences between iterations for the variables temperature, precipitation and green fraction. Still, we also used a second criterium that is based on a statistical test. We will clarify this here and describe it in more detail in section 2.4 of the revised manuscript.

*m22)   L136: "this result and its effect" What result and effect? Please clarify.*

**RESPONSE**:

We delete "result and its effect" to avoid any misleading.

*m23)   L137: "variables that mostly govern" Do you mean variables that govern the interaction most of the time, but not always, or do you mean most of the variables that govern the interaction? Please clarify.*

**RESPONSE**:

We mean most of the variables. We rephrase this in the revised manuscript as follows:

"To illustrate the changes between the different iterations, we concentrate on the climate and land cover changes over the ice-free land areas of Europe at LGM using the following variables: the spatial climatological of total precipitation, temperature at 2 m and vegetation cover fraction, and the number of grid points dominated by the following land cover categories: sparsely vegetated, tundra, forest, and shrublands …".

*m24)   L138: "suitable to illustrate the asynchronous coupling" What does this mean? Please rephrase and clarify.*

**RESPONSE**:

Please see response to comment m23 (related to line L137)

***m25)  L140: Please define the green fraction. After several readings I'm still not sure what it is.***

**RESPONSE**:

We will change this terms to vegetation cover fraction including its definition at the end of section 2.4 of the revised manuscript. Vegetation cover fraction is the fraction of ground covered by vegetation at each grid point. It varies between 0 and 1.

***m26)  L147: "in all variables" In both variables?***

**RESPONSE**:

We agree that this is unclear as we refer to all variables for the first sharp change and two variables for the second sharp change. This will be reformulated in the revised manuscript according to additional analysis mentioned in previous responses.

***m27)  L148-149:  I don't agree that the difference in climate is explained by differences in resolution. I would say that it is the difference in vegetation. A decrease in forest fraction from 35 % to just a few will have an effect on the simulated climate.   The fraction forest can be 35 % with 100 km grid spacing and with 10 km grid spacing. The difference in climate between cycle 1 and 2 is an effect of the difference in vegetation, don't you think?***

**RESPONSE**:

We partly agree with the reviewer. We think that it is not possible to disentangle the effect of increased resolution and the changes in the vegetation. In the revised manuscript, we will further discuss the GCM and WRF land cover when analysing our results (see also response to second major comment). Also, we will reformulate this part as follows.

"The first important change is found between the first and second iteration and is present in the atmospheric and land surface variables. The reasoning is twofold: (i) There are significant changes in the land cover classes, e.g., the forest fraction is reduced from 35 % to just 2 %. (ii) The horizontal resolution of the land cover is increased from approximately 100 km (spatial resolution of the GCM) to 18 km (spatial resolution of RCM). This results in a better representation of the regional-to-local processes and interactions of the climate system (Ludwig et al., 2019)."

*m28)   L148-149.  This sentence says: "the increase in resolution can be explained by the better representation of the circulation processes". Is this what you mean? What does it mean? consider rephrasing.*

**RESPONSE**:

Please see response to comment m27 (related to line L148).

*m29)   L149: "horizontal resolution" Of what?*

**RESPONSE**:

Please see response to comment m27 (related to line L148).

*m30)   L149: "1◦to 18 km" This is a not so pleasant mix of units. Since you say "approximately" and since your grid spacing is not exactly 1 I think its fine to say "approximately 100 to 18 km".*

**RESPONSE**:

We follow the suggestion of the reviewer. Please see response to comment m27 (related to line L148).

*m31)   L155: I would say that no important differences in the land surface variables are seen after cycle 2. If you see differences between cycle 2 and 3, please describe them.*

**RESPONSE**:

We disagree with the reviewer as we still see a further reduction of the green fraction from 20 to roughly 10%. We will describe this in the revised manuscript.

*m32)   L155: What do you mean with "especially" here? Especially large differences between1 and 2, or especially small thereafter?*

**RESPONSE**:

We reformulate this to "….. This is particularly true for green vegetation fraction and the category sparsely vegetated as they stay almost constant (Fig. 2d)."

*m33) L159-170: Why do you neglect to discuss temperature? Temperature is an important climate variable that responds to changes in vegetation and a variable that vegetation is limited by. You can't discuss equilibria and vegetation climate interactions without discussing temperature.*

**RESPONSE**:

We agree and include a discussion on temperature in the next manuscript.

*m34) L164: "In response to the progressive changes in precipitation" Do you mean that vegetation is only sensitive to precipitation changes?*

**RESPONSE**:

No, but it is certainly an important parameter. We will present a more detailed analysis and rephrase the entire paragraph in the revised manuscript.

*m35) L164-170: Line 164 says that vegetation responds to changes in precipitation. Line168 says that precipitation responds to changes in vegetation. What's your idea of how climate and vegetation interactions work? You mention temperature as a driver of vegetation on line 181. I don't think you explain it well enough.*

**RESPONSE**:

We agree and will include, as stated before, a more detailed discussion on precipitation and temperature changes.

*m36) L168-169: The correlation is not that good. Look at the Iberian Peninsula, France, the Balkans, Greece. There are lots of regions with increased precipitation and reduced green*

*fraction. Remember that your explanation of precipitation changes is not vegetation but changes in the large scale circulation. This is not affected by vegetation. There is little support that vegetation changes drive large scale changes in mean precipitation (e.g. Belusic et al., 2019; Strandberg & Kjellström, 2019; Davin et al. 2019).*

**RESPONSE**:

We agree that the interpretation was too superficial. By using also temperature changes and being more precise in circulation changes, we hope that the new version of the paragraph would be clearer in the revised manuscript.

*m37)   L170:  Internal variability of what?  I guess you mean in the climate itself.  Otherwise you should add it to the list of possible explanations.*

**RESPONSE**:

Yes, we mean climate internal variability. We will present a more detailed analysis and rephrase the entire paragraph in the revised manuscript.

*m38)   L173:  This is not a correct naming of this section as it also deals with atmospheric conditions, comparisons between LGM and PD, description of LGM climate and some discussion. Consider reordering this section and to divided into more sections.*

**RESPONSE**:

We change the title of the section as we restructured the section 4 and 5. Please refer to response to sixth major comment.

*m39)   L174: What's your definition of tree cover? Is it the same as green fraction?*

**RESPONSE**:

Tree cover is not exactly the same as green fraction, as the first is calculated by using only the ground covered by trees and excludes herbaceous and grass cover. Tree cover is used to facilitate the comparison with pollen data. We add a definition in the revised version.

*m40) L180-182: It is true, of course, that LGM vegetation is explained by climatic conditions and CO2 levels. But it doesn't explain why LGM vegetation was different than PD vegetation, because PD vegetation is highly anthropogenic. I don't think its correct to talk about changes here.*

**RESPONSE**:

We will reformulate these lines in the revised version to avoid any misleading.

*m41) L183: Add a reference to Fig 3b after "reconstructions".*

**RESPONSE**:

We include the following reference: Kaplan et al. (2016).

*m42) L188-189. This sentence is not very precise. It seems like all areas with few reconstructions show tundra and grassland, but actually you are only talking about the Carpathian Basin.*

**RESPONSE**:

We agree and clarify it in the revised manuscript.

 "For the Carpathian Basin, an area with few proxy reconstructions, the modelled LGM land cover categories show tundra and grassland, which is in agreement with results found by Magyari et al. (2014a,b)."

*m43) L190: Temperature and precipitation are not land surface conditions. See also comment on line 173.*

**RESPONSE**:

Yes, we agree. This section will be modified in the revised manuscript. Please also see response to comment m38 (related to line L173).

*m44)    L199: "few locations" Which locations?*

**RESPONSE**:

We specify the locations: "Still a few locations, e.g., over the Iberian Peninsula, show …"

*m45)    L200:  "in line with similar findings" It goes without saying that your results are in line with other results that are similar.  Are there also other results? Results that are not in line with yours? All the mentioned studies are made with GCMs. Wouldn't it be appropriate to compare also to S11 which uses a similar setting as yours?*

**RESPONSE**:

We thank you for highlighting this reference. We include a discussion on the results of S11 in the revised manuscript.

*m46)    L201: Don't forget that a lot of the shortcomings come from the driving GCM.*

**RESPONSE**:

Yes, we agree. We replace "RCM" with the "GCM-RCM modelling chain".

*m47)    L203-217:  This is a discussion, not results.   Consider moving to "Discussion".   It is also a highly confusing paragraph as it in the same time discusses model-proxy dis-agreement (line 204, line 216), climate anomalies (line 209) and LGM-PD (line 210, line 212). This needs to be straightened up.*

**RESPONSE**:

We will reformulate this section taking this comment into account in the revised manuscript.

*m48)    L205: Do you have a reference for the model-proxy disagreement in the Iberian Peninsula?*

**RESPONSE**:

We rephrase this sentence to: "Studies for example present a model-proxy disagreement over the Iberian Peninsula: ". We will present an additional reformulation for the following sentences in the revised manuscript.

*m49)   L209: "climate anomalies" What anomalies LGM-PD?*

**RESPONSE**:

Yes, we change this to "... same wetter conditions in the interior and northwestern Iberian Peninsula during the LGM compared to PD. To explain these wetter conditions, studies have suggested that the North Atlantic storm track…"

*m50)   L210-217: Changes in storm tracks could explain the increased precipitation in LGM in southern Europe, but it can't explain the model-proxy disagreement that this paragraph started with.  Another important part of the puzzle is the circulation in the GCM. What do the circulation patterns in CCSM look like?  Storm tracks should be easy enough to calculate, or at least a map of mslp.  You don't offer any descriptions of the climate in the driving GCM. Another reason for the different precipitation patterns in LGM is reduced evaporation from the cold and largely ice covered Atlantic (Strandberg et al.,2011).*

**RESPONSE**:

The GCM results were presented in several publications (e.g., Hofer et al. 2012a, 2012b, Merz et al. 2015). We think it is beyond the scope of this paper which focus on regional modelling rather than repeating figures from the GCM analysis. Though, we agree that some more information is needed. We decide to place this when we introduce a description of the main findings of the GCM simulation in section 2.1. See answer to the second major comment.

*m51)   L232: This section is not entirely about atmospheric sensitivity to land cover. Consider restructuring.*

**RESPONSE**:

We will present a new version of this section in the revised manuscript. Please refer to response to sixth major comment.

*m52)   L235: What do you mean by "again" here. Consider deleting.*

**RESPONSE**:

It will be deleted in the next manuscript.

*m53)   L236:  The atmospheric response mentioned here is not response to changes in the surface, but rather the models response to different forcing (GHGs, orbital forcing, orography...).*

**RESPONSE**:

We agree. We hope this will be clearer in the new version of this section. See response to comment m51 (related to line L232).

*m54)   L236: Comparing LGM and PD is not a way to estimate the atmospheric sensitivity to land cover.*

**RESPONSE**:

We agree. We try to enhance the structuring of this section in the new version of the manuscript. See response to comment m51 (related to line L232).

*m55)   L238-248:  It starts with a "precipitation decrease" on line 238.  This is illustrated by a "temperature response" on line 241. It then goes back to a "decrease of precipitation" on line 243. Please discuss on variable at the time. This is really hard to follow as it is now.*

**RESPONSE**:

We agree. This will be better discussed in the new version of this section. See response to comment m51 (related to line L232).

*m56)   L240-248:  It is not clear if this paragraph is only about southern Europe.  Please, be more precise with what regions you are discussing.*

**RESPONSE**:

We agree. This will be better discussed in the new version of this section. See response to comment m51 (related to line L232).

*m57)   L240:  "atmospheric response" To what? Vegetation, GHGs, orbital forcing, orography …?*

**RESPONSE**:

This will be clarified in the new version of this section. See response to comment m51 (related to line L232).

*m58)   L243: "decrease of precipitation" Between what? PD-LGM? LGM(LGM) - LGM(PD)?*

**RESPONSE**:

This will be better discussed in the new version of this section. See response to comment m51 (related to line L232).

*m59)   L245: "winter wetter conditions". On line 248 you mention a "general dryness in winter". Which is it? Do you discuss different regions?*

**RESPONSE**:

We will clarify this in the new version of this section. See response to comment m51 (related to line L232).

*m60)   L248:  Do you see a shift in storm tracks in you models?  If not it could hardly explain the precipitation patterns. It is not enough to just reference other studies.*

**RESPONSE**:

We see this shift in the driving GCM simulation (see e.g. Hofer et al. 2012a). By extending section 2.1, we hope it becomes clear. We specify that the driving GCM shows this change.

*m61)   L249:  "atmospheric response to the LGM(LGM) with respect to the PD(PD)". What does this mean? I don't understand.*

**RESPONSE**:

We hope this will be clear in the new version of this section. See response to comment m51 (related to line L232).

*m62)   L254: "reduced by 43 % in DJF and enhanced by about 35 % in JJA". I have difficulties to see this in Table 3.  First of all the numbers in Table 3 are given in mm/day so it is difficult to know the percentages.  Second, for LGM(LGM) - LGM(PD), which I guess this is about, precipitation is reduce for both DJF and JJA. I don't understand how JJA could see enhanced precipitation.*

**RESPONSE**:

This will be better explained in the new version of this section. See response to comment m51 (related to line L232).

*m63)   L256-259:  What is the significance of your results? Is it enough to make conclusions on? Precipitation changes need to be quite large to be significant.*

**RESPONSE**:

In the revised manuscript, we will include a statistical test to assess the significance (with a 5 % confidence level) and concentrate our discussion on significant changes.

*m64)   L259-264: This is a discussion, where are the results?*

**RESPONSE**:

We hope this would be clarified in the new version of this section. See response to comment m51 (related to line L232).

*m65)   L259-264:  Again, I would recommend you to take a look at S11 and see what is said there. In general, you shouldn't have to speculate about how vegetation interacts with climate. There are plenty of papers to read about that.  Furthermore, you have your own simulations. Why don't do a proper study of how for example albedo and heat fluxes change in your simulations?  If you want to have an example of how that could be done in a palaeo context you could e.g. look at Strandberg et al. (2014). For a more general analysis I can recommend Davin et al. (2019).*

**RESPONSE**:

We follow the suggestion of the reviewer. We will extend our analysis on albedo and heat fluxes change in our simulations in the revised manuscript.

*m66)   L262: "variability in land cover" Do you actually mean difference in land cover?*

**RESPONSE**:

Yes, this is corrected in the revised manuscript.

*m67)   L266:  There are several better references to this than AR5, e.g.  Strandberg & Kjellström, 2019, Davin et al. 2019, Jia et al., 2020.*

**RESPONSE**:

We change this accordingly.

*m68)   L280: Why do you expect the coupling to be particularly strong here?*

**RESPONSE**:

Strandberg et al. (2014) have shown in his RCM experiments for the Holocene that Eastern Europe summer precipitation is sensitive to land cover because there is an evapotranspiration feedback (see Fig. 8 in that paper). Temperature shows a similar effect, with reduction in tree cover leading to warmer and drier summers. The feedback is localized and especially strong in Eastern Europe. Therefore, we expected the coupling strength to be strong here, relative to other parts of Europe. We will clarify this in the revised manuscript.

*m69)   L282-285:  How would you explain these results?  If you can't explain it with physical effects it might as well be random .*

**RESPONSE**:

We regret saying that we hardly understand what the reviewer refers to. If it refers to lines 279-280, this will be better presented in a new version of the manuscript.

*m70)   L293: Is "parkland" the right way to describe LGM vegetation? Parkland seems highly anthropogenic.*

**RESPONSE**:

We agree that this can lead to misunderstandings. Boreal parkland is a common term used especially to describe a unique vegetation formation that was prevalent at LGM (e.g.; Prentice et al., 2011; Loehle, 2007). It is still used to describe the transition zone from grassland to forest in Canada. We will clarify this in the revised version.

*m71)   L296-298: I'm not sure if "illustrates" is the right word here. Shows?*

**RESPONSE**:

We change this to "Comparing LGM$_{LGM}$ to PD$_{PD}$ shows not only a general cooling and drying, but also a seasonality in the atmospheric response."

*m72)   L297:  "may be related to fluctuations in circulation patterns".  In the model one might add.*

**RESPONSE**:

It is done: "… may be related to fluctuations in circulation patterns in the model simulations."

*m73)   L302-305: How do you know this? You don't show it.*

**RESPONSE**:

This will be clarified in the revised manuscript.

*m74)   L304: "water fluxes" I guess you mean heat fluxes.*

**RESPONSE**:

Yes, it is a typo.

*m75)   L303-305:  I don't understand this sentence.  "LGM land cover led to /.../ when influenced by reduced vegetation fraction". So, the land cover is influenced by the vegetation fraction? Consider rephrasing.*

**RESPONSE**:

We agree the sentence is misleading. We rephrase it in the revised manuscript:

*m76)  L304:  Be careful with the use of parenthesis around "JJA". It don't play well with the other parentheses in this sentence.*

**RESPONSE**:

We fully agree and remove "(JJA)" to avoid any misunderstanding.

*m77)  Fig 3: You seem to use "land use" and "land cover" interchangeably. Choose one and stick to it.  I think land cover is the proper one since there were not much land use during the LGM. Land use is an anthropogenic thing.  Define "green fraction".  "Green vegetation cover" is not an explanation, just another way to say it.*

**RESPONSE**:

We will change this to "land cover" as suggested in the revised manuscript. We will also modify the term green fraction to vegetation cover fraction and include its definition in the text but not in the figure caption.

*m78)  Fig 4: The colour scale in a) and b) is not good. It's practically impossible to distinguish between colours in the range -24 - -4, and when I see a colour in the map I don't know where to place it in the colour scale.  Furthermore, it's very difficult to see the dots in the maps.  Find another way to plot them, perhaps with white circles.  It's also difficult to see the green and red rings. Think about if there is another way to plot significance.*

**RESPONSE**:

We will adjust the colour scale and plot the dots with white circles in the revised manuscript.

**Technical comments:**

*t1)      L139: "climatological" -> "climatology"*

**RESPONSE**:

It is done.

*t2)     L144: There is no reference to Fig 2 prior to this reference to Fig 3. Consider reordering the figures.*

**RESPONSE**:

We reorder the figures in the revised manuscript.

*t3)     L179 "Fig. 4a and b" -> "Fig. 3a and b"*

**RESPONSE**:

It is done.

We would like to thank the referee for the time invested in reviewing the manuscript so carefully. We are looking forward to meeting her/his expectations.

Best regards,

Patricio Velasquez (on behalf of the author team)

**References**

Belušíc, D., Fuentes-Franco, R., Strandberg, G.and Jukimenko, A., 2019: Afforestation reduces cyclone intensity and precipitation extremes over Europe. Environ. Res.Lett. 14,https://doi.org/10.1088/1748-9326/ab23b2

Davin, E. L., Rechid, D., Breil, M.,Cardoso, R. M., Coppola, E., Hoffmann, P., Jach, L. L., Katragkou, E., de Noblet-Ducoudré, N., Radtke, K., Raffa, M., Soares, P. M. M., Sofiadis, G., Strada, S., Strand-berg, G., Tölle, M. H., Warrach-Sagi, K., and Wulfmeyer, V.: Biogeophysical impacts of forestation in Europe: first results from the LUCAS (Land Use and Climate Across Scales) regional climate model intercomparison, Earth Syst. Dynam., 11, 183–200, https://doi.org/10.5194/esd-11-183-2020, 2020.

Demory, M.-E., Berthou, S., Sørland, S. L., Roberts, M. J., Beyerle, U., Seddon, J.,Haarsma, R., Schär, C., Christensen, O. B., Fealy, R., Fernandez, J., Nikulin, G.,Peano, D., Putrasahan, D., Roberts, C. D., Steger, C., Teichmann, C., and Vautard,R.: Can high-resolution GCMs reach the level of information provided by 12–50 kmCORDEX RCMs in terms of daily precipitation distribution?, Geosci. Model Dev. Dis-cuss., https://doi.org/10.5194/gmd-2019-370, in review, 2020.

Di Luca, A., de Elía, R. and Laprise, R.: Potential for added value in precipitation simulated by high-resolution nested Regional Climate Models and observations, Clim. Dyn. 38, 1229–1247, https://doi.org/10.1007/s00382-011-1068-3, 2011.

Iles, C. E., Vautard, R., Strachan, J., Joussaume, S., Eggen, B. R., and Hewitt, C. D.: The benefits of increasing resolution in global and regional climate simulations for European climate extremes, Geoscientific Model Development Discussion, https://doi.org/10.5194/gmd-2019-253, 2019

.Jia, G., E. Shevliakova, P. Artaxo, N. De Noblet-Ducoudré, R. Houghton, J. House, K.Kitajima, C. Lennard, A. Popp, A. Sirin, R. Sukumar, L. Verchot, 2019: Land–climate interactions. In: Climate Change and Land: an IPCC special report on climate change, desertification, land degradation, sustainable land management, food security, and greenhouse gas fluxes in terrestrial ecosystems [P.R. Shukla, J. Skea, E. Calvo Buendia, V. Masson-Delmotte, H.-O. Pörtner, D.C. Roberts, P. Zhai, R. Slade, S. Connors, R.van Diemen, M. Ferrat, E. Haughey, S. Luz, S. Neogi, M. Pathak, J. Petzold, J. PortugalPereira, P. Vyas, E. Huntley, K. Kissick, M, Belkacemi, J. Malley, (eds.)]. In press.

Kjellström, E., Brandefelt, J., Näslund, J.-O., Smith, B., Strandberg, G., Voelker, A. H. L.& Wohlfarth, B. 2010: Simulated climate conditions in Europe during the Marine IsotopeStage 3 stadial. Boreas, 10.1111/j.1502-3885.2010.00143.x. ISSN 0300-9483.

Loehle, C. (2007). Predicting Pleistocene climate from vegetation in North America. Clim Past, 3(1), 109-118. doi:10.5194/cp-3-109-2007

Prentice, I. C., Harrison, S. P., & Bartlein, P. J. (2011). Global vegetation and terrestrial carbon cycle changes after the last ice age. New Phytol, 189(4), 988-998. doi:10.1111/j.1469-8137.2010.03620.x

Prein, A. F., Holland, G. J., Rasmussen, R. M., Done, J., Ikeda, K., Clark, M. P. and Liu, C. H.: Importance of Regional Climate Model Grid Spacing for the Simulation of Heavy Precipitation in the Colorado Headwaters. J. Climate, 26: 4848–4857, doi:10.1175/JCLI-D-12-00727.1, 2013.

Rauscher, S.A., Coppola, E., Piani and Giorgi F.: Resolution effects on regional climate model simulations of seasonal precipitation over Europe. Clim. Dyn. 35, 685–711, https://doi.org/10.1007/s00382-009-0607-7, 2010.

Strandberg, G., Brandefelt, J., Kjellström, E. and Smith, B. 2011: High-resolution regional simulation of last glacial maximum climate over Europe. Tellus 63A, 107-125.DOI: 10.1111/j.1600-0870.2010.00485.x

Strandberg, G., Kjellström, E., Poska, A., Wagner, S., Gaillard, M.-J., Trondman, A.-K., Mauri, A., Davis, B. A. S., Kaplan, J. O., Birks, H. J. B., Bjune, A. E., Fyfe, R.,Giesecke, T., Kalnina, L., Kangur, M., van der Knaap, W. O., Kokfelt, U., Kuneš, P.,Latalowa, M., Marquer, L., Mazier, F., Nielsen, A. B., Smith, B., Seppä, H., and Sugita S.: Regional climate model simulations for Europe at 6 and 0.2 k BP: sensitivity to changes in anthropogenic deforestation, Clim. Past, 10, 661-680, doi:10.5194/cp-10-661-2014, 2014.

---

## Author Response (AR1)

**Final Response**

We thank the reviewers for the careful and thorough reading of our manuscript. The comments have been carefully considered and responded. Please find below our response to each comment.

**To Referee #1**

*General comment:*

*1.      Velasquez et al simulates LGM climate with a chain of asynchronous coupled models (GCM -> RCM -> DVM -> RCM etc). This is exactly what is needed in palaeo climate. High resolution enables better comparison with proxy data, and 18 km grid spacing is impressive in a palaeo context. A good description of vegetation is needed to get LGM conditions as realistic as possible. The authors make sure to have a vegetation compliant with the climate by making as much as 7 iterations with climate and vegetation models. This could be interesting for the readers of Climate of the past, but I think the paper need a lot of improvement before that. That the model set up is relevant is what makes me recommend major revision.*
*       My major concerns are with:*
*i)      the lack of discussion of the role of vegetation on LGM climate,*
*ii)     the lack of context and comparison with previous studies,*
*iii)    how the authors describe results, but don't explain them or try to understand them*
*iv)     the structure of the text where results from different sensitivity studies are mixed and where results and discussions are mixed.*
*       Detailed comments follow below. As always, I might have misunderstood some things, and my comments could be invalid because of that. If such misunderstandings do occur, think about if your paper is written in a clear enough way.*

**RESPONSE**:

We thank you for your detailed comments which has certainly improved the manuscript. We take care of the 4 major concerns listed above in the following responses and in the revised manuscript.

*Major comments:*

*1.      The title of the paper is "The role of land cover on the climate of glacial Europe", but I don't think I get any new insights by reading it.  Despite the ambitious model set up. It seems you did some sensitivity runs, but don't know what to do with them, and that you didn't study the literature on vegetation climate interactions. It's not enough to just say that LGM was cold and that vegetation affects climate, we know that already. Why does it? If you want to advance climate modelling you need to give physical reasons for your results.  Otherwise your results could just be a random effect caused by different initial conditions. Try to explain your results. Look at variables that might be relevant. Albedo and heat fluxes are not analysed at all.*

**RESPONSE**:

We thank you for this comment. We realise that there were some shortcomings in the description and interpretation of the results. In our revised manuscript, we have provided more reasoning and extended the analysis of the climatological variables, e.g., latent and sensible heat fluxes, in Sect. 6.

*2.      The results of the RCM simulations are highly depending on the driving GCM. But you don't discuss your GCM runs at all.  What is LGM vegetation like in the GCM? This explains many of the differences between cycle 1 and 2.  What is LGM climate like in the GCM compared to other PMIP3 runs, colder, warmer, wetter, drier?  What is the general circulation like? Could that explain the precipitation patterns?*

**RESPONSE**:

We agree that the discussion of the GCM was too short. Thus, in the revised manuscript, we have described in more detail how land cover is included in the GCM simulation; additionally, we further discussed the GCM and WRF land cover when analysing our results. We also added a brief comparison with other PMIP simulations although there are several papers which specifically focused on atmospheric dynamics, so a detailed repetition of these results is beyond the scope of this paper. We included the following passage in the revised manuscript:

"In this study, we dynamically downscaled one global climate simulation for PD conditions (1990 CE conditions) and another one for LGM. These global simulations were performed with the atmospheric and land component of the Community Climate System Model (version 4, CCSM4;

Gent et al., 2011). A horizontal grid spacing of 1.25° × 0.9° (longitude × latitude) was used in both components. The vertical dimension is discretised in 26 vertical hybrid sigma-pressure levels in the atmospheric component (CAM4, Neale et al., 2010) and 15 soil layers in the land component (CLM4, Oleson et al., 2010), respectively. CCSM4 was coupled to so-called *data models* for the ocean and sea ice. These surface boundary conditions were obtained from a fully coupled simulation with CCSM3, at lower resolution (see details in: Hofer et al. 2012a). CCSM3 provided monthly mean time-varying sea-ice cover and sea-surface temperatures (SSTs). Furthermore, the Community Ice Code (version 4 ,CICE4; Hunke and Lipscomb 2008) was set to its thermodynamic-only mode. This means that  sea-ice cover was prescribed, but surface fluxes through the ice were computed by considering snow depth, albedo, and surface temperature as simulated by CAM4 (Merz et al. 2015). Further details of the global model setting were presented in Hofer et al. (2012 a; b) and Merz et al. (2015).

Each CCSM4 simulation was run for 33 years, from which only the last 30 years and 2 months were used in this study. Present-day (PD) boundary conditions were set to 1990~CE values, whereas LGM boundary conditions were modified as follows: lower concentrations of greenhouse gases ($CO_2$ = 185 ppm; $N_2O$ = 200 ppb; $CH_4$ = 350 ppb), changed Earth's orbital parameters (Berger 1978), addition of major continental ice sheets (Peltier, 2004) and associated sea-level changes (120 m lower than today; Clark et al., 2009). Note that land cover was set to pre-industrial conditions in the LGM simulation. Additional land cells of the LGM simulation are filled with vegetation and soil types of the mean values of nearby cells and in the ice-covered regions the model's standard values are used for such conditions. The simulations further provided 6-hourly data, which is necessary to drive regional climate models.

These PD and LGM CCSM4 simulations have been analysed in a variety of studies, including additional simulations for other glacial and interglacial states (e.g., Hofer et al. 2012 a;b; Merz et al. 2013;2014a;b; 2015; 2016; Landais et al. 2016). The focus of these studies was in particular on the model's ability to simulate LGM climate and atmospheric circulation changes during glacial times. Hofer et al. (2012a) showed that the model performs reasonably well under PD conditions, showing a cold bias in the global mean temperature of 0.3 °C. The reason for this bias is the rather coarse resolution of the ocean, which led to an underestimation of the northward heat transport in the North Atlantic and an overestimation in the horizontal extension of sea ice-cover (Hofer et al. 2012a). The LGM CCSM4 simulation agrees with models used in the second phase of the Paleoclimate Modelling Intercomparison Project (PMIP2; Braconnot et al., 2007) showing a global mean temperature response between LGM and preindustrial conditions of -5.6 °C. However, the temperature response over Europe shows a better agreement with proxy data (Wu et al. 2007) than the multi-model mean response in Braconnot et al. (2007). The global mean precipitation response of the LGM simulation used in this study is similar to the multi-model mean response

of Braconnot et al. (2007), although the regional pattern and seasonal behaviour show some deviations from proxy data over Europe (Wu et al. 2007, Hofer et al. 2012a). The LGM simulation further reveals a clear southward shift and a more zonal orientation of the storm track over the North Atlantic compared to PD conditions (Hofer et al., 2012a). This shift, and substantial changes in the weather patterns (Hofer et al. 2012b), are able to explain precipitation anomalies over the Iberian Peninsula and the western part of the Mediterranean Sea. Sensitivity simulations in Merz et al. (2015) suggested that the shift can be traced back to the height of the Laurentide ice sheet and the effect of it on stationary and transient waves and the eddy-driven jet over the North Atlantic. Such a shift is also reported in several other modelling studies (see the review of Raible et al. 2020). Overall, CCSM4 simulations of LGM climate were state-of-the-art in 2012 and they are still today as their horizontal resolution is similar to models used in phase 4 of the Paleoclimate Model Intercomparison Project (PMIP4; Kagayama et al. 2017, 2020)."

*3.      Temperature differences between the different cycles are hardly discussed. Temperature decreases with almost 0.5◦C between cycles 1 and 2. This could perhaps be explained by a large albedo increase when the forest disappears (albedo is not shown). But temperature increases with 0.5◦C between cycles 3 and 4 despite no significant vegetation changes. This is unexplained. Could it be that the differences are just a result of natural variability and not a result of vegetation at all?*

**RESPONSE**:

We thank the reviewer for this comment. We agree that this part needed clarification. Therefore, we added a more detailed analysis of temperature in the revised manuscript in Sect. 3. We also mentioned in Sect. 2 of the revised manuscript that we identified a problem with the land-sea mask and around glaciated areas which was fixed between the third and fourth iteration.

*4.      A related issue to the above is the question of significance.  The significance of the results is not tested here.  Even though vegetation has a clear effect on climate the effect is small compared to the effect of different forcing, and sometimes also compared to natural variability. Therefore, it is important to check if the results are significant.*

**RESPONSE**:

We agree that the significance of results needed to be assessed. We have therefore included the analysis of the significance in the revised manuscript. To test whether the asynchronous coupling

has reached a quasi-equilibrium state, we applied a statistical test to the spatial pattern and at each grid point, separately. The statistical significance between one and another iteration is deduced with a bootstrapping technique. This method is described in the last paragraph of Sect. 2.2 and mentioned in Sect. 2.4.

*5. I feel that the paper misses to discuss some relevant previous studies, particularly Strandberg et al. (2011), henceforth S11. S11 studies LGM with more or less the same method as you do. The present model set up is an improvement from S11 so you shouldn't be afraid to discuss it. Why not get inspired by how S11 discuss the role of the GCM or the uncertainty in proxy data. Vegetation climate interactions is not the main number in S11, but there is a section about that too. Kjellström et al. (2010) used the same approach for another cold climate, MIS3.*

**RESPONSE**:

We thank the reviewer for suggesting this relevant literature. Although we mentioned S11, we realize that its discussion was too short. Therefore, we included a more comprehensive discussion of the existing literature and similarities of our approach to existing ones in the revised manuscript. Descriptions and discussions have been introduced in the introduction, section 3 and in section 6.

*6. The structure of the paper makes it a bit difficult to follow. For example, section 4 "Comparison of the simulated land surface conditions to proxy reconstructions" deals in large parts also with vegetation differences between PD and LGM, difference in climate between PD and LGM, difference in climate between models and proxies and a discussion about simulated LGM climate in other studies. In a similar way section 5 "Atmospheric sensitivity to land cover" deals largely with differences between LGM and PD climate. I think it would be good if you could discuss one thing at the time.*

**RESPONSE**:

We have reorganized section 4 and 5 into the following sections:

4. Comparison and discussion of the modelled and reconstructed climate
5. Comparison and discussion of the modelled and reconstructed land cover
6. Influence of land cover and external forcing on climate

This also responds to comments m38 and m51 (related to line L173 and L232, respectively). Note that we keep the discussion of the results in these sections rather than having a section "discussion" as this would enlarge the manuscript substantially and lead to some repetitions.

*Minor comments:*

*m1)    L10-14: I think these conclusions are too general given the title of the paper.  Please quantify a bit and perhaps also explain why you get an effect.*

**RESPONSE**:

Please see response to comment m4 (related to line L13).

*m2)    L11-12: "colder and drier", "warmer and drier".  Is this LGM(LGM) - LGM(PD) or LGM(PD) - LGM(LGM)?*

**RESPONSE**:

It is $LGM_{LGM}$ - $LGM_{PD}$. See response to comment m4 (related to line L13).

*m3)    L13: "southward displacement". This sentence reads to me like "Even with a southward displacement of the storm track regional climate is influenced by land cover".  Is this what you mean?  Why would a southward displacement counteract or diminish the influence of vegetation? Please rephrase.*

**RESPONSE**:

This part is too specific for the abstract and has been removed. See response to comment m4 (related to line L13).

*m4)    L13: "increased importance of the Atlantic".  What do you mean?  In what way is the Atlantic more important?  And why?  I can't find anything about the importance of the Atlantic in the rest of the text.*

**RESPONSE**:

Response to the first 4 minor comments. We rephrased the 4 lines of the abstract:

"To assess the importance of land cover on the LGM climate of Europe, we performed a sensitivity simulation where we used LGM climate but present-day (PD) land cover. Using LGM climate and land cover leads to colder and drier summer conditions around the Alps and warmer and drier climate in southeastern Europe compared to LGM climate determined by PD land cover. This finding demonstrates that LGM land cover plays an important role in regulating the regional climate. Therefore, realistic glacial land cover estimates are needed to accurately simulate regional glacial climate states in areas with interplays between complex topography, large ice sheets and diverse land cover, as observed in Europe.

*m5)    L25: "Recent advances" It could be discussed how recent it is since S11 is 9 years old. See discussion above.*

**RESPONSE**:

We rephrased the sentence to "Advances in regional climate models have led to the application of such models to the glacial climate of Europe on a high spatial resolution (e.g., Kjellström et al. 2010, Strandberg et al. 2011, Ludwig et al., 2017; 2020).".

*m6)    L45: There are better references to this than AR5, e.g. Rauscher et al., 2010; Di Luca et al., 2011; Prein et al., 2013; Iles et al., 2019 and Demory et al. 2020.*

**RESPONSE**:

We replaced the AR5 reference with the ones mentioned and add also:

Gomez-Navarro, J. J., O. Boethe, S. Wagner, E. Zorita, J. P. Werner, J. Luterbacher, C.C. Raible, and J.P. Montavez, 2015: A regional climate palaeo simulation for Europe in the period 1501-1990. Part II: comparison with gridded reconstructions. *Climate of the Past*, **11**, 1077-1095

Gómez-Navarro, J. J., Montávez, J. P., Jerez, S., Jiménez-Guerrero,P., Lorente-Plazas, R., González-Rouco, J. F., and Zorita, E.: A regional climate simulation over the Iberian Peninsula for the last millennium, Clim. Past, 7, 451–472, doi:10.5194/cp-7-451-2011,2011.

Gómez-Navarro, J. J., Montávez, J. P., Jiménez-Guerrero, P., Jerez,S., Lorente-Plazas, R., González-Rouco, J. F., and Zorita, E.: Internal and external variability in regional simulations of the Iberian Peninsula climate over the last millennium, Clim. Past,8, 25–36, doi:10.5194/cp-8-25-2012, 2012.

Gómez-Navarro, J. J., Montávez, J. P., Wagner, S., and Zorita, E.:A regional climate palaeo simulation for Europe in the period1500–1990 – Part 1: Model validation, Clim. Past, 9, 1667–1682,doi:10.5194/cp-9-1667-2013, 2013.

*m7)    L52: Is Tao et a. (2013) the appropriate reference here? It's about the effect of vegetation on air quality in the US. If you want vegetation climate interactions in RCMs in palaeo climate (Consistent with Strandberg et al. (2011) and Ludwig et al. (2017)) I would recommend e.g. Kjellström et al. (2010) or Strandberg et al. (2014). If you want a more general reference on vegetation climate interactions you could start with Jia et al. (2019).*

**RESPONSE**:

We agree that the research by Tao et al. was not the optimal reference in the context of paleo studies. We replaced it with the mentioned publications.

*m8)    L61: There are two studies that use a similar approach as you, and in addition in cold climates: Kjellström et al. (2010) and Strandberg et al. (2011). Especially S11 would be worth to note as it simulates LGM.*

**RESPONSE**:

We agree. Therefore, we added the information that the approach is similar to the ones suggested in Kjellström et al. (2010) and Strandberg et al. (2011) in the revised manuscript. See also response to the fifth major comments.

*m9)    L63-68: When I read this I understand that you have the following model chain: GCM->RCM->DVM->RCM-> etc. From section 2.4 I understand that the first DVM simulation is forced by the GCM. I suppose that the description in 2.4 is the correct one. Please, check and correct.*

**RESPONSE**:

Yes, this is somewhat misleading. We rephrased it to:

"… The asynchronous coupled modelling starts with a GCM (CCSM4; Gent et al., 2011) which serves as input to drive a dynamic vegetation model (LPJ-LMfire; Pfeiffer et al., 2013). In a next step, the atmospheric boundary conditions from the GCM and the output of LPJ-LMfire are passed to an RCM (WRF; Skamarock and Klemp, 2008). The resulting RCM output is in turn used to drive LPJ-LMfire which again returns land cover to the RCM. The RCM simulation is then repeated with the new land cover as boundary condition. …"

*m10)  L72-75:  Are these 31 years part of a longer simulation, in that case how were they selected? Is the LGM simulation a part of a transient simulation or is it steady state?*

**RESPONSE**:

We realise that the description of the GCM was too short. We included a better description of the GCM in the revised manuscript. Please refer to the response to the second major comment and response to m12 (related to line L87).

*m11)  L74: Please add an explanation and a reference to these data models.*

**RESPONSE**:

The data model is a specific expression in the CESM world. The data models (for ocean and sea ice) simply prescribe time-varying sea-surface temperatures over the oceans and sea-ice conditions, which are obtained from a fully coupled simulation. This explanation is now included in the revised manuscript. See response to second major comment.

*m12)  L87: 31 years were simulated by the GCM, 30 by the RCM. Why is one year not used? Which 30 years are used?*

**RESPONSE**:

We agree that this was unclear. 33 years in total were run with the GCM in an equilibrium state, whereas the first 3 years are considered as spin-up time. We used the last two months of these 3 years as spin-up of our RCM and the following 30 years are downscaled by the RCM and provide the basis for our analysis. Note that the 30-year simulation with the RCM are divided in two 15-year chunks. While the first 15-year simulation makes use of the 2 months spin-up that coincides with the one of the GCM, the second chunk overlaps with the last two months of the first 15 years RCM simulation. We have clarified this in Sect. 2.2 of the revised manuscript, it now reads as follows:

"The initial and boundary conditions for the WRF model were provided by CCSM4 simulations, including the Fennoscandian ice sheet and reduced sea levels during the LGM. Other external forcing functions followed the PMIP3 protocol (for more details see: Hofer et al., 2012a; Ludwig et al., 2017). Furthermore, no nudging was applied in the RCM simulations. LGM glaciation over the Alpine region was included in the regional climate model using estimates from Seguinot et al. (2018) and additional LGM glaciated areas (e.g., Pyrenees, Carpathians) from Ehlers et al. (2011). The LGM land cover is described in Sect. in Sect. 2.4. These settings are used to produce the main simulation (LGM$_{LGM}$) which at the same time is the final product of the asynchronous coupling design (described in Sect. in Sec 2.4).

To perform the regional simulations in this study, we used the so-called adaptive time-step method as described in Skamarock et al. (2008), i.e., the integration time step can vary from time to time. For example, the model is stable with a time step of 160 seconds during most integration steps but it might need a reduction to 60 seconds during convective situations to maintain stability. With a fixed time step, the entire simulation must be run with 60 seconds to overcome these convective situations, while the adaptive time-step method is able to make use of the larger time step 160 seconds during most of the simulation. The advantage of this approach is to substantially save computer resources. Furthermore, each simulation was driven by the 30 years of the corresponding GCM simulation (excluding the 3-year spin-up of the GCM simulation). These 30 years were split up into two single 15-year periods which both are preceded by a 2-month spin-up to account for the time required for land surface to come into quasi equilibrium. We used the last two months of the 3-year spin-up of the GCM simulation for the first 15 years. A spin-up of two months in the regional model is sufficient as soil moisture reaches a quasi equilibrium, i.e., no significant trend after 15 days in the four layers of the WRF land-surface scheme, i.e., up to the level of 1 m..

We also carried out a control simulation under PD conditions (PD$_{PD}$) to assess the simulated LGM climate and land cover response against proxy data. PD$_{PD}$ was driven by the GCM simulation with 1990 CE conditions (Hofer et al., 2012a), and used the default PD MODIS-based land-cover dataset from WRF (Skamarock and Klemp, 2008).

Finally, we conducted a sensitivity simulation to quantify the importance of land cover for the LGM climate in Europe (LGM$_{PD}$). This simulation used the GCM simulation with LGM conditions (Hofer et al., 2012a), but with the default PD MODIS-based land-cover dataset from WRF for the land surface.

Comparing LGM$_{PD}$ with PD$_{PD}$ illustrates the atmospheric response to changes only in the atmospheric forcing, i.e., without changes in land cover. The comparison of LGM$_{LGM}$ and the LGM$_{PD}$ allows us to extract the influence of land cover on the atmosphere, i.e., without changes in atmospheric boundary conditions. These simulations are summarised in Table 1."

***m13) L88: "adaptive time-step to increase … computer facilities" I don't understand this at all. What does it mean?***

**RESPONSE**:

Climate models usually use a fixed time step to integrate the equations, i.e., every 160 seconds in the coarsest domain. However, the WRF model allows to integrate the equations with variable time steps depending on the situation, i.e., stability of the current model integration step indicated by the CFL criterion, the so-called adaptive time-step method as described in Skamarock et al. (2008). The advantage of this method is to substantially save computer resources as the calculation time can be reduced when using bigger time steps. For example, the model simulation is stable with a timestep of 160 seconds during most integration steps but might need a reduction to 60 seconds during convective situations. With a fixed timestep the whole simulation must be run with 60 seconds to overcome these few convective situations, while the adaptive time-step method is able to make use of the larger time step 160 seconds during most of the simulation. This behaviour of the adaptive time-step can save considerable amounts of computational time.

The details are certainly beyond the scope of this manuscript but if a reader would like to follow our approach it is an important information when setting up WRF. We have clarified it in the revised manuscript, see also response to comment m12.

***m14)   L89:  If you divide 30 years into two 15 year periods and start every simulation with a two-month spin-up this will give you 2x(14y 10m) = 29 years and 8 months.  How do you get 30 years of data from that?***

**RESPONSE**:

We simply start 2 months before the actual 15-year period begins. For instance, we started on November 1st of the 14th year when simulating the second 15 years. See also response to comment m12 (related to line L87).

We understand that this is somewhat unclear in the manuscript and rephrased this sentence to: "These 30 years were split up into two single 15-year periods which are both preceded by a 2-month spin-up to account for the time required for land surface to come into quasi equilibrium."

***m15)   L90: 15 days seems to me to be a bit short. How do you decide that quasi-equilibrium is reached? In Velasquez et al. (2019) I can only find the following: "Tests show that the WRF land scheme reaches a quasi-equilibrium after approximately 15 d." That doesn't explain much.  I guess that spin-up time also depends on the season.  When do you start your simulation?***

**RESPONSE**:

We performed several tests on this issue. Therefore, we assessed the soil moisture in different levels. A notable trend is observed in the four layers within the first 10 days, which initially suggests that the spin-up period could be set at around 15 days. To ensure that the regional model is in quasi-equilibrium, we defined a longer spin-up that covers 61 days. This information is included in the Sect 2.2. of the revised manuscript.

***m16)   L93:  What do you mean with "perpetual" here?  Do you mean steady-state with constant forcing?***

**RESPONSE**:

Yes, we mean with constant LGM conditions. Note that "perpetual" is often used in this context but we decided to remove this word in the revised manuscript.

*m17)   L93: "Reduced sea level and increased ice sheets".  This is somewhat ambiguous.  I guess you mean that sea level was lower and ice sheets were larger than today. It could also mean that LGM conditions have been revised in the PMIP3 protocol compared to previous protocols. It's not entirely clear.*

**RESPONSE**:

We reformulated and moved this information to the second paragraph of Sect. 2.2 of the revised manuscript.

*m18)   L115 Are the "reconstructed CO2 concentrations" used in the RCM the same as used in the GCM (PMIP3 forcing)?  It seems like an unnecessary complicated way to say that forcing is the same as in the GCM. If it's not the same, why not?*

**RESPONSE**:

They are the same, so we decided to reformulate the sentence to "In each of our simulations, we drove LPJ-LMfire for 1020 years with the climate and forcing (greenhouse gases: CO2, N2O and CH4) from the GCM, and PD soil physical properties extrapolated out on to the continental shelves (Kaplan et al., 2016)."

*m19)   L120-122:  What vegetation field is used in the first GCM run?  See also comment onL63-68.*

**RESPONSE**:

As mentioned in Hofer et al. (2012) the vegetation and the soil types in the LGM simulation with the GCM are prescribed to the preindustrial distribution except for the additional land areas and the regions that are covered by ice sheets. In the additional land cells, vegetation and soil types are set to the mean values of nearby cells and in the ice covered regions the model's standard value for such conditions are used. We extended our model description accordingly in the revised manuscript. See the response to the second major comment.

*m20)   L133:  I would say it's more correct to call this section "Results of the iterative...".  You don't have other simulations to compare with so you can't estimate the effect of the coupling. You could, however, describe the results of your simulations, and that's what you do.*

**RESPONSE**:

We changed to "Results of the iterative asynchronous coupling" in the revised manuscript.

*m21)   L135: How do you determine that quasi-equilibrium is reached?  Just by eyeballing or do you have a criterion for equilibrium. It seems like you just decide that equilibrium is reached, but how can you be sure without a proper metric?*

**RESPONSE**:

It is always difficult to give a quantitative measure when only having 7 iterations, so the first selection was done by eyeballing, i.e., a rather stable mean and no coherent structure when plotting differences between iterations for the variables temperature at 2 m, precipitation and vegetation cover fraction. Still, we also used a second criterion that is based on a statistical test. We have clarified this and described it in more detail in Sect. 2.2 and mentioned it again in Sect. 2.4 of the revised manuscript.

*m22)   L136: "this result and its effect" What result and effect? Please clarify.*

**RESPONSE**:

We deleted "result and its effect" to avoid any misleading.

*m23)   L137: "variables that mostly govern" Do you mean variables that govern the interaction most of the time, but not always, or do you mean most of the variables that govern the interaction? Please clarify.*

**RESPONSE**:

We mean most of the variables. We rephrased this in the revised manuscript as follows:

"To illustrate the differences between the iterations, we concentrate on climate and land cover changes over the ice-free land areas of Europe at LGM (in domain 2) using the following variables: the spatial climatology of total precipitation, temperature at 2 m, albedo, deep-soil temperature, cloud cover, leaf area index and vegetation cover fraction, and the number of grid points dominated by the following land cover categories: sparsely vegetated, tundra, forest, and shrublands…".

*m24)   L138: "suitable to illustrate the asynchronous coupling" What does this mean? Please rephrase and clarify.*

**RESPONSE**:

We removed this sentence, please see response to comment m23 (related to line L137)

*m25)   L140: Please define the green fraction. After several readings I'm still not sure what it is.*

**RESPONSE**:

We changed this terms to vegetation cover fraction including its definition in section 2.4 of the revised manuscript. Vegetation cover fraction is the fraction of ground covered by vegetation at each grid point. It varies between 0 and 100 %.

We added in Sect. 2.4:

"Vegetation cover fraction is defined as the fraction of ground covered by vegetation at each grid point, with values between 0 and 100 %."

*m26)   L147: "in all variables" In both variables?*

**RESPONSE**:

We agree that this is unclear as we referred to all variables for the first sharp change and two variables for the second sharp change. This has been reformulated in the revised manuscript according to additional analysis mentioned in previous responses.

*m27) L148-149: I don't agree that the difference in climate is explained by differences in resolution. I would say that it is the difference in vegetation. A decrease in forest fraction from 35 % to just a few will have an effect on the simulated climate. The fraction forest can be 35 % with 100 km grid spacing and with 10 km grid spacing. The difference in climate between cycle 1 and 2 is an effect of the difference in vegetation, don't you think?*

**RESPONSE**:

We partly agree with the reviewer. We think that it is not possible to disentangle the effect of increased resolution and the changes in the vegetation. In the revised manuscript, we included a description of the land cover conditions of the GCM (see also response to second major comment). Also, we reformulated this part as follows.

"The first important change is found between the first and second iteration and is present in the atmospheric and land surface variables. The reasoning is twofold: (i) There are significant changes in the land cover classes, e.g., forest fraction is reduced from 35 to 2 %. (ii) The horizontal resolution of the land cover is increased from approximately 100 km to 18 km (horizontal grid spacing of GCM and RCM, respectively). The higher spatial resolution of the RCM results in a better representation of the regional-to-local scale processes and interactions with other components of the climate system compared to a GCM (Ludwig et al., 2019)."

*m28) L148-149. This sentence says: "the increase in resolution can be explained by the better representation of the circulation processes". Is this what you mean? What does it mean? consider rephrasing.*

**RESPONSE**:

We have rephrased this sentence. Please see response to comment m27 (related to line L148).

*m29) L149: "horizontal resolution" Of what?*

**RESPONSE**:

We have clarified this in the new version of the manuscript to: "The higher spatial resolution of the RCM …". Please see response to comment m27 (related to line L148).

***m30)   L149: "1∘to 18 km" This is a not so pleasant mix of units. Since you say "approximately" and since your grid spacing is not exactly 1 I think its fine to say "approximately 100 to 18 km".***

**RESPONSE**:

We have followed the suggestion of the reviewer. Please see response to comment m27 (related to line L148).

***m31)   L155: I would say that no important differences in the land surface variables are seen after cycle 2. If you see differences between cycle 2 and 3, please describe them.***

**RESPONSE**:

It is true, that the land cover categories mainly change between the first and second iteration. Nevertheless, the vegetation cover fraction, which is also considered as land surface variable, sees a further reduction from 18 to 15% between the second and fourth iteration. We described this in the revised manuscript:

"Additionally, changes in vegetation cover fraction are observed in the first four iterations (32, 18, 16 and 15 %, respectively  In the following iterations, the changes remain rather small and insignificant (Fig. 3b). The land cover categories change mostly between the first and second iteration. The category sparsely vegetated is strongly increased in the second iteration and at the same time forest is strongly reduced (Fig. 3b)…"

***m32)   L155: What do you mean with "especially" here? Especially large differences between1 and 2, or especially small thereafter?***

**RESPONSE**:

We reformulated this sentence. See response to m31 (related to L155).

***m33)   L159-170:  Why do you neglect to discuss temperature?  Temperature is an important climate variable that responds to changes in vegetation and a variable that vegetation is limited by.  You can't discuss equilibria and vegetation climate interactions without discussing temperature.***

**RESPONSE**:

We agree and a discussion on temperature is included in the new version of the manuscript.

*m34)  L164: "In response to the progressive changes in precipitation" Do you mean that vegetation is only sensitive to precipitation changes?*

**RESPONSE**:

No, but it is certainly an important parameter. We presented a better analysis and rephrased the entire paragraph in the revised manuscript.

*m35)  L164-170:  Line 164 says that vegetation responds to changes in precipitation.  Line168 says that precipitation responds to changes in vegetation.   What's your idea of how climate and vegetation interactions work? You mention temperature as a driver of vegetation on line 181. I don't think you explain it well enough.*

**RESPONSE**:

We agree and included, as stated before, a better discussion on precipitation and temperature changes.

*m36)  L168-169: The correlation is not that good. Look at the Iberian Peninsula, France, the Balkans, Greece.  There are lots of regions with increased precipitation and reduced green fraction. Remember that your explanation of precipitation changes is not vegetation but changes in the large scale circulation. This is not affected by vegetation. There is little support that vegetation changes drive large scale changes in mean precipitation (e.g. Belusic et al., 2019; Strandberg & Kjellström, 2019; Davin et al. 2019).*

**RESPONSE**:

We agree that the interpretation was too superficial. By using also temperature changes and being more precise in circulation changes, we hope that the reformulated paragraph would be clearer in the revised manuscript.

*m37)   L170:  Internal variability of what?  I guess you mean in the climate itself.  Otherwise you should add it to the list of possible explanations.*

**RESPONSE**:

Yes, we mean climate internal variability. We presented a more detailed analysis and rephrased the entire paragraph in the revised manuscript.

*m38)   L173:  This is not a correct naming of this section as it also deals with atmospheric conditions, comparisons between LGM and PD, description of LGM climate and some discussion. Consider reordering this section and to divided into more sections.*

**RESPONSE**:

We changed the title of the section as we restructured section 4 and 5. Please refer to response to the sixth major comment.

*m39)   L174: What's your definition of tree cover? Is it the same as green fraction?*

**RESPONSE**:

Tree cover is not exactly the same as green fraction, as the first is calculated by using only the ground covered by trees and excludes herbaceous and grass cover. Tree cover is used to facilitate the comparison with pollen data. We added a definition in the first paragraph of Sect. 5 in the revised manuscript.

*m40)   L180-182: It is true, of course, that LGM vegetation is explained by climatic conditions and CO2 levels.   But it doesn't explain why LGM vegetation was different than PD vegetation, because PD vegetation is highly anthropogenic. I don't think its correct to talk about changes here.*

**RESPONSE**:

We have reformulated these lines in the revised version to avoid any misleading.

***m41) L183: Add a reference to Fig 3b after "reconstructions".***

**RESPONSE**:

We included the reference to Fig 2b (order of figures has changed in the revised manuscript) as suggested by the reviewer.

***m42) L188-189. This sentence is not very precise. It seems like all areas with few reconstructions show tundra and grassland, but actually you are only talking about the Carpathian Basin.***

**RESPONSE**:

We agree and clarified it in the revised manuscript.

"For the Carpathian Basin, an area with few proxy reconstructions, the modelled LGM land cover categories are tundra and grassland, which is in agreement with results found by Magyari et al. (2014a,b)."

***m43) L190: Temperature and precipitation are not land surface conditions. See also comment on line 173.***

**RESPONSE**:

Yes, we agree. This section was modified in the revised manuscript. Please also see response to comment m38 (related to line L173).

***m44) L199: "few locations" Which locations?***

**RESPONSE**:

We have specified the locations: "Still, some samples, e.g., over the Iberian Peninsula, show …"

*m45)   L200: "in line with similar findings" It goes without saying that your results are in line with other results that are similar.  Are there also other results? Results that are not in line with yours? All the mentioned studies are made with GCMs. Wouldn't it be appropriate to compare also to S11 which uses a similar setting as yours?*

**RESPONSE**:

We thank you for highlighting this reference. We included a discussion on the results of S11 in the revised manuscript here and in section 6.

*m46)   L201: Don't forget that a lot of the shortcomings come from the driving GCM.*

**RESPONSE**:

Yes, we agree. We replaced "RCM" with the "GCM-RCM modelling chain".

*m47)   L203-217:  This is a discussion, not results.   Consider moving to "Discussion".   It is also a highly confusing paragraph as it in the same time discusses model-proxy dis-agreement (line 204, line 216), climate anomalies (line 209) and LGM-PD (line 210, line 212). This needs to be straightened up.*

**RESPONSE**:

We reformulated this section taking this comment into account in the revised manuscript.

*m48)   L205: Do you have a reference for the model-proxy disagreement in the Iberian Peninsula?*

**RESPONSE**:

We reformulated this paragraph in the revised manuscript (last paragraph of section 4).

***m49)  L209: "climate anomalies" What anomalies LGM-PD?***

**RESPONSE**:

Yes, we changed this to "... the interior and northwestern Iberian Peninsula experience wetter conditions during the LGM. These wetter conditions can be explained by a southward shift in the North Atlantic storm track …" in the last paragraph of Sect. 4 in the revised manuscript.

***m50)  L210-217: Changes in storm tracks could explain the increased precipitation in LGM in southern Europe, but it can't explain the model-proxy disagreement that this paragraph started with.  Another important part of the puzzle is the circulation in the GCM. What do the circulation patterns in CCSM look like?  Storm tracks should be easy enough to calculate, or at least a map of mslp.  You don't offer any descriptions of the climate in the driving GCM. Another reason for the different precipitation patterns in LGM is reduced evaporation from the cold and largely ice covered Atlantic (Strandberg et al.,2011).***

**RESPONSE**:

Yes, we agree that storm track changes cannot explain model proxy disagreement. We clarified this in the paragraph. Note also that the GCM results were presented in several publications (e.g., Hofer et al. 2012a, 2012b, Merz et al. 2015). We think that a detailed discussion of this is beyond the scope of this paper as the focus is here on regional modelling rather than repeating figures from the GCM analysis. Though, we agree that some more information is needed. We decided to place this when we introduce a description of the main findings of the GCM simulation in section 2.1. See answer to the second major comment.

***m51)  L232: This section is not entirely about atmospheric sensitivity to land cover. Consider restructuring.***

**RESPONSE**:

We present a new version of this section in the revised manuscript. Please refer to the response to the sixth major comment.

*m52)   L235: What do you mean by "again" here. Consider deleting.*

**RESPONSE**:

It is deleted in the revised manuscript.

*m53)   L236:  The atmospheric response mentioned here is not response to changes in the surface, but rather the models response to different forcing (GHGs, orbital forcing, orography...).*

**RESPONSE**:

We agree. We hope this is clearer in the new version of this section. See response to comment m51 (related to line L232).

*m54)   L236: Comparing LGM and PD is not a way to estimate the atmospheric sensitivity to land cover.*

**RESPONSE**:

We agree. We have re-organised this section in the new version of the manuscript. See response to comment m51 (related to line L232).

*m55)   L238-248:  It starts with a "precipitation decrease" on line 238.  This is illustrated by a "temperature response" on line 241. It then goes back to a "decrease of precipitation" on line 243. Please discuss one variable at the time. This is really hard to follow as it is now.*
**RESPONSE**:

We agree. This has been better discussed in the new version of this section (second paragraph of Sect. 6). See response to comment m51 (related to line L232).

*m56)  L240-248:  It is not clear if this paragraph is only about southern Europe.  Please, be more precise with what regions you are discussing.*

**RESPONSE**:

We agree. We included a better discussion in the new version of this section (second paragraph of Sect. 6). See response to comment m51 (related to line L232).

*m57)  L240: "atmospheric response" To what? Vegetation, GHGs, orbital forcing, orography ...?*

**RESPONSE**:

This has been clarified in the new version of this section (second paragraph of Sect. 6). See response to comment m51 (related to line L232).

*m58)  L243: "decrease of precipitation" Between what? PD-LGM? LGM(LGM) - LGM(PD)?*

**RESPONSE**:

We added a better discussion in the new version of this section (second paragraph of Sect. 6). See response to comment m51 (related to line L232).

*m59)  L245: "winter wetter conditions". On line 248 you mention a "general dryness in winter". Which is it? Do you discuss different regions?*

**RESPONSE**:

We have clarified this in the new version of this section (second paragraph of Sect. 6). See response to comment m51 (related to line L232).

*m60)  L248:  Do you see a shift in storm tracks in you models?  If not it could hardly explain the precipitation patterns. It is not enough to just reference other studies.*

**RESPONSE**:

We see this shift in the driving GCM simulation (see e.g. Hofer et al. 2012a). By extending section 2.1, we hope it became clear. We specified that the driving GCM shows this change.

*m61)   L249: "atmospheric response to the LGM(LGM) with respect to the PD(PD)". What does this mean? I don't understand.*

**RESPONSE**:

We hope this has been clarified in the new version of this section (beginning of third paragraph of Sect. 6). See response to comment m51 (related to line L232).

*m62)   L254: "reduced by 43 % in DJF and enhanced by about 35 % in JJA". I have difficulties to see this in Table 3.  First of all the numbers in Table 3 are given in mm/day so it is difficult to know the percentages.  Second, for LGM(LGM) - LGM(PD), which I guess this is about, precipitation is reduce for both DJF and JJA. I don't understand how JJA could see enhanced precipitation.*

**RESPONSE**:

This has been better explained in the new version of this section (third paragraph of Sect. 6). See response to comment m51 (related to line L232).

*m63)   L256-259:  What is the significance of your results?  Is it enough to make conclusions on? Precipitation changes need to be quite large to be significant.*
**RESPONSE**:

In the revised manuscript, we included a statistical test to assess the significance (with a 2 % significance level) and concentrate our discussion on significant changes.

*m64) L259-264: This is a discussion, where are the results?*

**RESPONSE**:

We hope this has been clarified in the new version of this section. See response to comment m51 (related to line L232).

*m65) L259-264: Again, I would recommend you to take a look at S11 and see what is said there. In general, you shouldn't have to speculate about how vegetation interacts with climate. There are plenty of papers to read about that. Furthermore, you have your own simulations. Why don't do a proper study of how for example albedo and heat fluxes change in your simulations? If you want to have an example of how that could be done in a palaeo context you could e.g. look at Strandberg et al. (2014). For a more general analysis I can recommend Davin et al. (2019).*

**RESPONSE**:

We followed the suggestion of the reviewer. We extended our analysis on heat fluxes change in our simulations in the revised manuscript.

*m66) L262: "variability in land cover" Do you actually mean difference in land cover?*

**RESPONSE**:

Yes, this has been corrected in the revised manuscript.

*m67) L266: There are several better references to this than AR5, e.g. Strandberg & Kjellström, 2019, Davin et al. 2019, Jia et al., 2020.*

**RESPONSE**:

We changed this accordingly.

*m68)   L280: Why do you expect the coupling to be particularly strong here?*

**RESPONSE**:

Strandberg et al. (2014) have shown in their RCM experiments for the Holocene that Eastern Europe summer precipitation is sensitive to land cover because there is an evapotranspiration feedback (see Fig. 8 in that paper). Temperature shows a similar effect, with reduction in tree cover leading to warmer and drier summers. The feedback is localized and especially strong in Eastern Europe. Therefore, we expected the coupling strength to be strong here, relative to other parts of Europe. We clarified this in the revised manuscript as follows:

"….their spatial pattern strongly changes across Europe (Fig. 7). Important spatial changes are statistically significant over eastern Europe in July. Strandberg et al. (2011) and Kjellstrom et al. (2010), in similar coupling designs, compared glacial simulations using two land cover settings and found that the simulated regional climate patterns in parts of Europe are sensitive to feedbacks from large differences in vegetation. Particularly, Kjellstrom et al. (2010) found that glacial-like vegetation leads to warmer conditions over Eastern Europe compared to modern vegetation. Strandberg et al. (2014) showed in their RCM experiments for the Holocene that summer temperature and precipitation are sensitive to changes in land cover in eastern Europe due to evapotranspiration (in our results as latent heat) feedbacks (see Fig. 8 in Strandberg et al., 2014). They found that a reduction in tree cover leads to warmer and drier summers in eastern Europe, which is similar to our finding as we observe that a reduction of vegetation cover fraction is associated with a warmer and drier July in the same region. This suggests that the land-atmosphere coupling-strength may be stronger in eastern Europe compared to other parts of Europe, especially during summer. "

*m69)   L282-285:  How would you explain these results?  If you can't explain it with physical effects it might as well be random .*

**RESPONSE**:

We regret saying that we hardly understand what the reviewer refers to. If it refers to lines 279-280, this has been better presented in a new version of the manuscript.

***m70)   L293: Is "parkland" the right way to describe LGM vegetation? Parkland seems highly anthropogenic.***

**RESPONSE**:

We agree that this can lead to misunderstandings. Boreal parkland is a common term used especially to describe a unique vegetation formation that was prevalent at LGM (e.g., Prentice et al., 2011; Loehle, 2007). It is still used to describe the transition zone from grassland to forest in Canada. We have clarified this in the revised version as follows:

"…The $LGM_{LGM}$ land cover was characterised by tundra and sparse vegetation, although open forest parkland (transition from grass to forest during the LGM) may have been common in many parts of central Europe, which is supported by comparisons with pollen-based vegetation reconstructions."

***m71)   L296-298: I'm not sure if "illustrates" is the right word here. Shows?***

**RESPONSE**:

We changed this to "Comparing $LGM_{LGM}$, i.e. the complete LGM conditions, to $PD_{PD}$ shows not only a general cooling and drying, but also a seasonality in the atmospheric response."

***m72)   L297:  "may be related to fluctuations in circulation patterns".  In the model one might add.***

**RESPONSE**:

We have deleted this sentence and have reorganised this paragraph with some reformulations. We hope this is now clear in the revised manuscript.

*m73)   L302-305: How do you know this? You don't show it.*

**RESPONSE**:

This is clarified in the revised manuscript. To that end, we added an analysis about spatially averaged climatology and the spatial pattern of the modelled latent and sensible heat fluxes in Sect. 6.

*m74)   L304: "water fluxes" I guess you mean heat fluxes.*

**RESPONSE**:

Yes, it is a typo. It has been changed to "heat fluxes"

*m75)   L303-305: I don't understand this sentence. "LGM land cover led to /.../ when influenced by reduced vegetation fraction". So, the land cover is influenced by the vegetation fraction? Consider rephrasing.*

**RESPONSE**:

We agree the sentence is misleading. We rephrase it in the revised manuscript as follows:

"…Our results show that dry conditions in LGM are partially attributed to LGM land cover as a reduction in vegetation overall led to stronger dryness compared to PD land cover. This is particularly true for central and eastern Europe during summer. "

*m76)   L304:  Be careful with the use of parenthesis around "JJA". It don't play well with the other parentheses in this sentence.*

**RESPONSE**:

We fully agree and removed "(JJA)" to avoid any misunderstanding and reformulated these lines. Please refer to response to comment m75 (related to lines 303-305).

*m77)   Fig 3: You seem to use "land use" and "land cover" interchangeably. Choose one and stick to it.  I think land cover is the proper one since there were not much land use during the LGM. Land use is an anthropogenic thing.  Define "green fraction".  "Green vegetation cover" is not an explanation, just another way to say it.*

**RESPONSE**:

We have changed this to "land cover in WRF" in the revised manuscript. We also modified the term green fraction to vegetation cover fraction and include its definition in the text but not in the figure caption.

*m78)   Fig 4: The colour scale in a) and b) is not good. It's practically impossible to distinguish between colours in the range -24 - -4, and when I see a colour in the map I don't know where to place it in the colour scale.  Furthermore, it's very difficult to see the dots in the maps.  Find another way to plot them, perhaps with white circles.  It's also difficult to see the green and red rings. Think about if there is another way to plot significance.*

**RESPONSE**:

We adjusted the colour scale and plot the dots with bigger circles in the revised manuscript.

*Technical comments:*

*t1)      L139: "climatological" -> "climatology"*

**RESPONSE**:

It was changed as suggested.

*t2)      L144: There is no reference to Fig 2 prior to this reference to Fig 3. Consider reordering the figures.*

We reordered the figures in the revised manuscript.

*t3)     L179 "Fig. 4a and b" -> "Fig. 3a and b"*

**RESPONSE**:

This line was reformulated and the reference to these figures does not exist anymore.

**To Referee #2**

*1.      Regarding the results presented in Fig. 3, the statement "The LGMLGM climate agrees with the pollen-based paleoclimate reconstructions at most sites" (l.198) is not very convincing. I will not say that only "few locations show considerable differences" (l.199) since 5 out of 14 sites show temperatures reconstructed in July significantly different from the simulations and since 6 out of 14 sites show precipitation in July significantly different from the simulated ones.*

**RESPONSE**:

We thank you for this comment. We agree that the statement in the manuscript is not yet convincing. We have instead considered not only the site as sample, but also the two variables and the two months (14 x 2 x 2 samples). This means that 14 sites offer 56 samples, from which 15 samples do not agree with modelled climate (5 temperature and 6 precipitation samples in July, and 4 precipitation samples in January). To clarify this, we have explained it better and reformulated the beginning of Sect. 4:

"To evaluate the $LGM_{LGM}$ climate simulation, we compared temperature and precipitation to pollen-based reconstructions. Wu et al. (2007) provided reconstructions of temperature and precipitation for the coldest and warmest months of the LGM at 14 sites in Europe. Thus, we considered 56 samples (14 sites x 2 variables x 2 months) in this comparison."

*2.        The regional character of the January precipitation in Southern Europe during the LGM compared to PD (higher LGM precipitation) is not supported by the data, as noted by the authors. They discuss this point but the cited works are not well cited or at least the text as written is misleading for the reader. Roucoux et al. (2005) effectively suggest that the LGM is not the driest and coldest interval of the last ice age, but wetter than the periods before and after it. However, in Roucoux et al. (2005) these colder and dryer periods are the Heinrich events and not the recent period. The Estanya lake record in the NE Iberia (Morellon et al. 2009) is also cited as showing wetter LGM conditions. This is ok but unlike the simulations, these lake data show that the LGM is wetter than the H1 in the NE of the Iberian Peninsula but much drier than the Holocene and in particular the final Holocene. The same applies to the modelling work of Ludwig et al, 2018, showing that the LGM is wetter than H1 but drier than the pre-industrial period. Citing all these works for justifying that other data or modelling experiments show wetter conditions during the LGM but avoiding to say "wetter than what" is misleading for the reader. In any case, they cannot be used as a justification to explain that the simulations show wetter winter conditions at the LGM than at the PD. A comparison with a larger number of sites would be beneficial for the evaluation of the simulations.*

**RESPONSE**:

We thank the reviewer for pointing out that the discussion of the literature is misleading for the reader. We would like to mention that we used these publications to highlight the uncertainties related with past climates. To avoid any further misleading, we have double-checked the literature and reformulated this paragraph in the revised manuscript as follows:

"One important model-proxy disagreement is the precipitation anomaly over the Iberian Peninsula in January. Based on evidence for the presence of certain tree species in the northwestern part of the Iberian Peninsula, Roucoux et al. (2005) suggested that the LGM was not necessarily the period of the most severe, i.e., cold and dry, climatic conditions everywhere. Roucoux et al. (2005) and Ludwig et al., (2018) also suggested that this region during LGM sensu strictu was warmer and wetter than the end of Marine Isotope Stage 3 (MIS3, ca. 23 ka; Voelker, A. H. L. et al., 1997; Kreveld, S. et al., 2000) and the start of the Heinrich event 1 (H1, ca. 19 ka; Sanchez Goñi and Harrison, 2010; Álvarez-Solas et al., 2011; Stanford et al., 2011). This could be a hint that model-proxy comparison fails because the proxies refer to $21 \pm 2$ ka (Wu et al., 2007), i.e., either the end of MIS3 or beginning of H1. Compared to the pre-industrial period, Beghin et al. (2016) found evidence in a model-proxy comparison that the interior and northwestern Iberian Peninsula experience wetter conditions during the LGM. These wetter conditions can be explained by a southward shift in the North Atlantic storm track during LGM compared to PD as suggested by many studies (e.g., Hofer et al., 2012a; Luetscher et al., 2015; Merz et al., 2015;

Ludwig et al., 2016; Wang et al., 2018; Raible et al., 2020). Note further that we had only two pollen-based quantitative climate reconstructions from Iberia for the LGM; we therefore consider the model-proxy intercomparison in this region equivocal. "

***3.       It would be good to add sites whose reconstructions are available in the literature and not only those from a compilation made more than 14 years ago.***

RESPONSE:

We appreciate this suggestion. We also agree that adding more sites would be beneficial for the evaluation. For example, since the publications of Wu et al. (2007) and Bartlein et al. (2011), more than 70 well-dated pollen records from Europe that cover the LGM have become available (Kaplan et al., 2016). However, these data have not been transformed into paleoclimate reconstructions to-date and such an effort would be beyond the scope of the current study.

***4.       A strong added value to the paper would be to estimate the temperature and precipitation (with a MAT or another method) over the 71 sites used in Figure 6. Doing a modeldata comparison on the basis of 71 sites instead of the 14 currently used would bring more robustness to the validation of the simulations by the data. Nevertheless, I would understand that it is a too much work for this paper.***

RESPONSE:

We would like to mention that we do carry out a model-data comparison using tree cover. We agree that further model-data comparisons using additional reconstructed information from these 71 sites would certainly add more value. However, this is surely an effort that is beyond the scope of this study.

***5.       The authors chose to compare simulated tree cover % to the available arboreal PFT % from pollen records to evaluate the model simulations. However, it would be great to take into account in the discussion that arboreal PFT % "is a relative rather than absolute metric of landscape openness" as stated by Davis et al. 2015. p. 6,***

**RESPONSE**:

We agree that it is currently not possible with any method to make reliable quantitative reconstructions of tree cover using LGM pollen assemblages. We clarified this in the last paragraph of Sect. 5 of the new version of the manuscript.

*6.      l. 179: "Fig. 3" instead of "Fig. 4".*

**RESPONSE**:

We thank the reviewer for pointing out the mismatch in the figure references. We have changed this as suggested.

We would like to thank the referees for the time invested in reviewing the manuscript so carefully. We are looking forward to meeting her/his expectations.

Best regards,

Patricio Velasquez (on behalf of the author team)

**References**

Álvarez-Solas, J., Montoya, M., Ritz, C., Ramstein, G., Charbit, S., Dumas, C., Nisancioglu, K., Dokken, T., Ganopolski, A., 2011. Heinrich event 1: an example of dynamical icesheet reaction to oceanic changes. Clim. Past 7, 1297–1306. https://doi.org/10.5194/cp-7-1297-2011.

Beluší c, D., Fuentes-Franco, R., Strandberg, G.and Jukimenko, A., 2019: Afforestation reduces cyclone intensity and precipitation extremes over Europe. Environ. Res.Lett. 14,https://doi.org/10.1088/1748-9326/ab23b2

Davin, E. L., Rechid, D., Breil, M.,Cardoso, R. M., Coppola, E., Hoffmann, P., Jach, L. L., Katragkou, E., de Noblet-Ducoudré, N., Radtke, K., Raffa, M., Soares, P. M. M., Sofiadis, G., Strada, S., Strand-berg, G., Tölle, M. H., Warrach-Sagi, K., and Wulfmeyer, V.: Biogeophysical impacts of forestation in Europe: first results from the LUCAS (Land Use and Climate Across Scales) regional climate model intercomparison, Earth Syst. Dynam., 11, 183–200, https://doi.org/10.5194/esd-11-183-2020, 2020.

Demory, M.-E., Berthou, S., Sørland, S. L., Roberts, M. J., Beyerle, U., Seddon, J.,Haarsma, R., Schär, C., Christensen, O. B., Fealy, R., Fernandez, J., Nikulin, G.,Peano, D., Putrasahan, D., Roberts, C. D., Steger, C., Teichmann, C., and Vautard,R.: Can high-resolution GCMs reach the level of information provided by 12–50 kmCORDEX RCMs in terms of daily precipitation distribution?, Geosci. Model Dev. Dis-cuss., https://doi.org/10.5194/gmd-2019-370, in review, 2020.

Di Luca, A., de Elía, R. and Laprise, R.: Potential for added value in precipitation simulated by high-resolution nested Regional Climate Models and observations, Clim. Dyn. 38, 1229–1247, https://doi.org/10.1007/s00382-011-1068-3, 2011.

Iles, C. E., Vautard, R., Strachan, J., Joussaume, S., Eggen, B. R., and Hewitt, C. D.: The benefits of increasing resolution in global and regional climate simulations for European climate extremes, Geoscientific Model Development Discussion, https://doi.org/10.5194/gmd-2019-253, 2019

Jia, G., E. Shevliakova, P. Artaxo, N. De Noblet-Ducoudré, R. Houghton, J. House, K.Kitajima, C. Lennard, A. Popp, A. Sirin, R. Sukumar, L. Verchot, 2019: Land–climate interactions. In: Climate Change and Land: an IPCC special report on climate change, desertification, land degradation, sustainable land management, food security, and greenhouse gas fluxes in terrestrial ecosystems [P.R. Shukla, J. Skea, E. Calvo Buendia, V. Masson-Delmotte, H.-O. Pörtner, D.C. Roberts, P. Zhai,

R. Slade, S. Connors, R.van Diemen, M. Ferrat, E. Haughey, S. Luz, S. Neogi, M. Pathak, J. Petzold, J. PortugalPereira, P. Vyas, E. Huntley, K. Kissick, M, Belkacemi, J. Malley, (eds.)]. In press.

Kreveld, S. van, Sarnthein, M., Erlenkeuser, H., Grootes, P., Jung, S., Nadeau, M. J., Pflaumann, U. and Voelker, A.: Potential links between surging ice sheets, circulation changes, and the Dansgaard-Oeschger Cycles in the Irminger Sea, 60–18 Kyr, Paleoceanography, 15(4), 425–442, https://doi.org/10.1029/1999PA000464, 2000.

Kjellström, E., Brandefelt, J., Näslund, J.-O., Smith, B., Strandberg, G., Voelker, A. H. L.& Wohlfarth, B. 2010: Simulated climate conditions in Europe during the Marine IsotopeStage 3 stadial. Boreas, 10.1111/j.1502-3885.2010.00143.x. ISSN 0300-9483.

Loehle, C. (2007). Predicting Pleistocene climate from vegetation in North America. Clim Past, 3(1), 109-118. doi:10.5194/cp-3-109-2007

Prentice, I. C., Harrison, S. P., & Bartlein, P. J. (2011). Global vegetation and terrestrial carbon cycle changes after the last ice age. New Phytol, 189(4), 988-998. doi:10.1111/j.1469-8137.2010.03620.x

Prein, A. F., Holland, G. J., Rasmussen, R. M., Done, J., Ikeda, K., Clark, M. P. and Liu, C. H.: Importance of Regional Climate Model Grid Spacing for the Simulation of Heavy Precipitation in the Colorado Headwaters. J. Climate, 26: 4848–4857, doi:10.1175/JCLI-D-12-00727.1, 2013.

Rauscher, S.A., Coppola, E., Piani and Giorgi F.: Resolution effects on regional climate model simulations of seasonal precipitation over Europe. Clim. Dyn. 35, 685–711, https://doi.org/10.1007/s00382-009-0607-7, 2010.

Sanchez Goñi, M.F., Harrison, S.P., 2010. Millennial-scale climate variability and vegetation changes during the last glacial: concepts and terminology. Quat. Sci. Rev. 29, 2823–2827. https://doi.org/10.1016/j.quascirev.2009.11.014.
Stanford, J. D., Rohling, E. J., Bacon, S., Roberts, A. P., Grousset, F. E. and Bolshaw, M.: A new concept for the paleoceanographic evolution of Heinrich event 1 in the North Atlantic, Quaternary Science Reviews, 30(9), 1047–1066, https://doi.org/10.1016/j.quascirev.2011.02.003, 2011.

Strandberg, G., Brandefelt, J., Kjellström, E. and Smith, B. 2011: High-resolution regional simulation of last glacial maximum climate over Europe. Tellus 63A, 107-125.DOI: 10.1111/j.1600-0870.2010.00485.x

Strandberg, G., Kjellström, E., Poska, A., Wagner, S., Gaillard, M.-J., Trondman, A.-K., Mauri, A., Davis, B. A. S., Kaplan, J. O., Birks, H. J. B., Bjune, A. E., Fyfe, R.,Giesecke, T., Kalnina, L., Kangur, M., van der Knaap, W. O., Kokfelt, U., Kuneš, P.,Latalowa, M., Marquer, L., Mazier, F., Nielsen, A. B., Smith, B., Seppä, H., and Sugita S.: Regional climate model simulations for Europe at 6 and 0.2 k BP: sensitivity to changes in anthropogenic deforestation, Clim. Past, 10, 661-680, doi:10.5194/cp-10-661-2014, 2014.

Sugita, S. (2007). Theory of quantitative reconstruction of vegetation I: pollen from large sites REVEALS regional vegetation composition. *The Holocene, 17*(2), 229-241. doi:10.1177/0959683607075837.

Voelker, A. H. L., Sarnthein, M., Grootes, P. M., Erlenkeuser, H., Laj, C., Mazaud, A., Nadeau, M.-J. and Schleicher, M.: Correlation of Marine 14C Ages from the Nordic Seas with the GISP2 Isotope Record: Implications for 14C Calibration Beyond 25 ka BP, Radiocarbon, 40(1), 517–534, https://doi.org/10.1017/S0033822200018397, 1997.

---

## Author Response (AR2)

**Response**

We greatly thank the reviewer for the careful and thorough reading of our revised manuscript. The technical corrections have been fully considered. We have corrected the three technical issues as suggested by the reviewer. We are looking forward to the published manuscript.

Best regards,

Patricio Velasquez (on behalf of the author team)